

# $G_2$-manifolds from 4d N=1 theories,
# part I: Domain walls

**Andreas P. Braun[1], Evyatar Sabag[2], Matteo Sacchi[2] and Sakura Schäfer-Nameki[2]**

**1** Department of Mathematical and Computing Sciences,
Durham University Upper Mountjoy Campus,
Stockton Rd, Durham DH1 3LE, UK
**2** Mathematical Institute, University of Oxford, Andrew-Wiles Building,
Woodstock Road, Oxford, OX2 6GG, UK

## Abstract

We propose new $G_2$-holonomy manifolds, which geometrize the Gaiotto-Kim 4d $\mathcal{N} = 1$ duality domain walls of 5d $\mathcal{N} = 1$ theories. These domain walls interpolate between different extended Coulomb branch phases of a given 5d superconformal field theory. Our starting point is the geometric realization of such a 5d superconformal field theory and its extended Coulomb branch in terms of M-theory on a non-compact singular Calabi-Yau three-fold and its Kähler cone. We construct the 7-manifold that realizes the domain wall in M-theory by fibering the Calabi-Yau three-fold over a real line, whilst varying its Kähler parameters as prescribed by the domain wall construction. In particular this requires the Calabi-Yau fiber to pass through a canonical singularity at the locus of the domain wall. Due to the 4d $\mathcal{N} = 1$ supersymmetry that is preserved on the domain wall, we expect the resulting 7-manifold to have holonomy $G_2$. Indeed, for simple domain wall theories, this construction results in 7-manifolds, which are known to admit torsion-free $G_2$-holonomy metrics. We develop several generalizations to new 7-manifolds, which realize domain walls in 5d SQCD theories and walls between 5d theories which are UV-dual.



# 1  Introduction

Geometric engineering is not only a means of translation between quantum field theory (QFT) and geometry, but can oftentimes be essential in making progress in either of these areas – for instance, the existence of 5d and 6d superconformal field theories [1, 2] hinges strongly on their robust realization in terms of geometric engineering. There is obviously no free lunch in this enterprise, and e.g. geometries associated to theories with less supersymmetry are usually also more difficult to analyze. The geometric challenges become particularly pronounced when constructing 4d minimally supersymmetric QFTs in M-theory on 7-manifolds with reduced holonomy $G_2$. The absence of simple criteria that imply the existence of Ricci-flat metrics, such as is the case for Calabi-Yau manifolds, makes these exceptional holonomy spaces notoriously difficult to construct.

An interesting avenue to pursue in order to construct spaces with (potentially) torsion-free $G_2$-holonomy metrics, is to use field theory results, which guarantee 4d $\mathcal{N} = 1$ super-symmetry, and to geometrize these. In this series of papers, we shall take precisely such an approach: the construction of certain 4d minimally supersymmetric theories – domain walls and super-conformal field theories (SCFTs) – will be systematically geometrized; thus, reverse-engineering $G_2$-manifolds.

Central to this endeavour is the construction of said 4d theories in terms of supersymmetric QFTs in 5d (or 6d), which in turn have a very precise geometric engineering characterization as M-theory on non-compact Calabi-Yau three-folds with canonical singularities (or F-theory on elliptically fibered Calabi-Yau three-folds). The 5d SCFTs and their extended Coulomb branch (which furnishes a weakly-coupled IR description) have a precise mathematical characterization in terms of the Kähler cone of the underlying Calabi-Yau geometry. Utilizing this allows us to geometrize the construction of 4d domain walls or SCFTs. This provides a systematic way to construct non-compact manifolds of real dimension seven, which – as long as the QFT results are sound and trustworthy – should admit torsion-free $G_2$-holonomy metrics.

**Field theory construction.** In this paper we focus on the duality domain walls of Gaiotto and Kim [3]. These are constructed in terms of two 5d theories along $\mathbb{R}^{1,4}$ with $x_4 > 0$ and $x_4 < 0$, respectively, with the domain wall separating them at $x_4 = 0$. The 5d theories are taken to be in different sub-chambers of the extended Coulomb branch (ECB) of the same SCFT, with a non-trivial identification of the ECB parameters at the location of the domain wall. This identification corresponds to a transformation in the Weyl group of the ultraviolet (UV) flavor symmetry. Concretely, this is realized by fibering the 5d theory along $x_4$ in such a way that we move along a path in its ECB from one sub-chamber to another. A crucial requirement for the existence of a non-trivial 4d domain wall is that $x_4 = 0$ is a locus in the ECB where the effective gauge coupling diverges.

One of the simplest such duality domain walls arises for the rank 1 $E_1$ Seiberg theory, which has a low energy $SU(2)_0$ gauge theory description. In this case we move in the direction of its ECB parameterized by switching on a non-trivial mass for the instanton particle. This is chosen to be positive for $x_4 < 0$, negative for $x_4 > 0$ and vanishing at the location of the domain wall $x_4 = 0$, which implies that the gauge coupling diverges at this point. Hence, on the two sides of the domain wall we have the same $SU(2)_0$ gauge theory, but with the instanton mass of opposite sign, which corresponds to a transformation in the Weyl group of the UV $SO(3)$ flavor symmetry group. This domain wall configuration is summarized in figure 1.

The construction in [3] that we will geometrize generalizes this example to the 5d SCFTs with a low energy effective gauge theory description in terms of $SU(N)_k + N_F \boldsymbol{F}$ with Chern–Simons level $k = N - \frac{N_F}{2}$ and $N_F \leq 2N$ flavors. Subsequently, in [4–6] it was shown that these constructions can be extended to a higher number of flavors – all the way up to the 5d KK-theories, obtained from 6d $(1,0)$ SCFTs compactified on a circle. Specifically, in the case of $N = 2$, one can extend the above constructions to include up to $N_F = 8$ flavors, which corresponds to the reduction of the rank 1 E-string 6d SCFT.

We will further extend this to domain walls that separate two 5d theories that are UV-dual, i.e. different 5d IR-descriptions that have a common SCFT origin. The simplest such UV-dual pair is $_\pi SU(2) - SU(2)_\pi$ and $SU(3)_0 + 2\boldsymbol{F}$. This opens up another avenue for constructing domain walls and associated $G_2$-holonomy spaces.

**Geometric construction.** The subject of this paper is the geometrization of the above-mentioned domain walls. What is by now very well understood is the construction of 5d SCFTs and their ECB from Calabi-Yau three-fold singularities. While 5d SCFTs correspond to the canonical singularities, moving onto the extended Coulomb branch corresponds to the res-

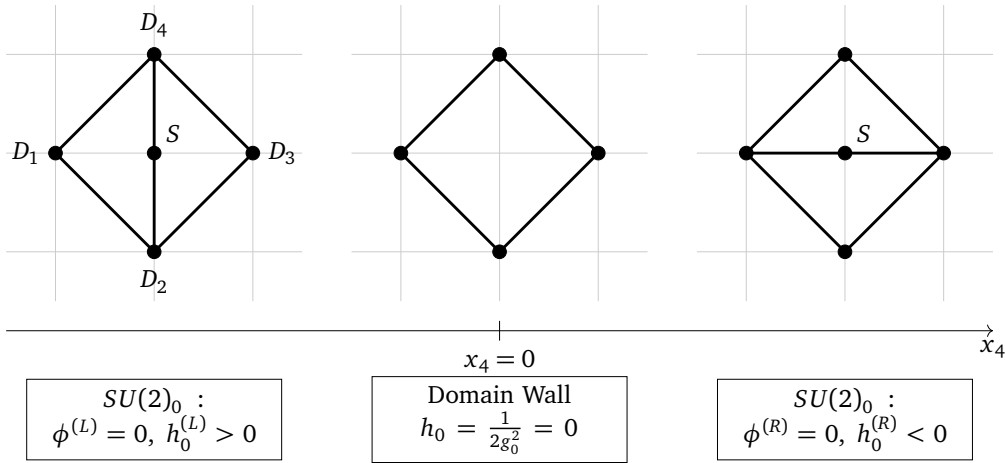

Figure 1: The domain wall for the 5d $E_1$ theory. The 5d theory is realized in terms of the Calabi-Yau manifold which is the complex cone over $\mathbb{F}_0 = \mathbb{P}^1 \times \mathbb{P}^1$. The top row shows the toric realization of the 5d geometry as it is fibered along $x_4$ ($D_i$ are non-compact divisors, and $S$ is the compact divisor). Below, we show the field theory description. On either side of the domain wall, which is located at $x_4 = 0$, we have a description in terms of an $SU(2)_0$ 5d gauge theory. The theories are parameterized by their extended Coulomb branch parameters: $\phi$ the Coulomb branch parameter and $h_0$ the mass of the instanton particle. The Coulomb branch parameters get identified by $\phi^{(L)} \to \phi^{(R)} = \phi^{(L)} - h_0^{(L)}/2$ and the instanton masses by $h_0^{(R)} = -h_0^{(L)}$. This change of extended Coulomb branch locus as we move from $x_4 < 0$ to $x_4 > 0$ is modelled by the distinct partial resolution of the toric geometry (which both realize $SU(2)_0$ gauge theories). The singular geometry associated to the SCFT is located at $x_4 = 0$. Fibering the Calabi-Yau geometry in this way along $x_4$ results in a $G_2$-holonomy manifold, which in this specific example is $\mathcal{M}_{E_1} = C(\mathbb{P}^3)/\mathbb{Z}_2$.

olution of the singularities (in a Calabi-Yau preserving, i.e. crepant, fashion). In this way, the Kähler cone maps precisely to the ECB of the 5d theory [7–14]. It is this precise dictionary in 5d that allows us to geometrize the domain wall construction: associated to each ECB phase of the 5d theory, we have a Calabi-Yau three-fold $\mathcal{X}$. The main idea is to fiber the Calabi-Yau threefold $\mathcal{X}(x_4)$ over the $x_4$-direction in space-time, whilst varying the Kähler parameters as dictated by the domain wall construction. This results in a 7d non-compact space $\mathcal{M}$, which is expected to admit a $G_2$-holonomy metric.

The domain wall is specified by an identification of 5d ECB parameters. In the geometry this corresponds to an identification of the Kähler parameters across different chambers in the Kähler cone. Crucially, the identification requires the Calabi-Yau fibers $\mathcal{X}(x_4)$ to develop canonical singularities at the location of the domain wall – often realizing the underlying SCFT.

**Geometric checks.** There are two immediate sniff-tests that we can apply to this construction. First of all, one could worry, whether the resulting space may be a 7-manifold that is a product of the Calabi-Yau with a real line. Indeed, the Cheeger-Gromoll theorem [15] implies that *complete* manifolds of non-negative Ricci curvature are necessarily an isometric product whenever they contain a geodesic line. Given a smooth $G_2$-manifold $\mathcal{M}$ that is a fibration of a smooth Calabi-Yau threefold $\mathcal{X}$ over the real line allows to construct such a geodesic and $\mathcal{M}$ is hence metrically a product manifold with holonomy $SU(3)$. However, existence of singularities which is essential for the domain-wall geometries turns $\mathcal{M}$ into a non-complete space, so that the Cheeger-Gromoll theorem does not obstruct $\mathcal{M}$ from having holonomy $G_2$.

Secondly, in the case that $\mathcal{M}$ is a conical $G_2$-manifold, its cross section $L$ must be a nearly Kähler-manifold. This in turn implies that there cannot be any calibrated complex surfaces inside $L$ that are non-trivial in homology.[1] Indeed, the field theory construction of the domain walls demands that all Coulomb branch parameters are set to zero,[2] which implies geometrically that all compact divisors of $\mathcal{X}$ are collapsed.

In fact, the simplest types of domain walls can be constructed by a (partial) unfolding of a hyper-Kähler singularity. For the case of the hyper-Kähler unfolding of an $A_1$ singularity, this results in a cone over $\mathbb{P}^3$, a space which famously allows a $G_2$ metric [16]. As anticipated from the description of this space as a hyper-Kähler unfolding, this metric can be described as a fibration of smoothed $A_1$ singularities [17]. Interestingly, the fibers do not carry a hyper-Kähler structure, but only a hypersymplectic one.

Using this approach we are not only able to connect field theories to known $G_2$-holonomy spaces, but, more importantly, to argue for a vast generalization of known methods of construction of $G_2$-spaces. Equipped with the huge collection of 5d SCFTs that are known (and potentially classified) and admit a geometric description in terms of Calabi-Yau three-folds, this approach provides a very exciting avenue to systematically construct vast classes of $G_2$-geometries that realize 4d $\mathcal{N} = 1$ domain wall theories.

One of the key challenges, which has a particular relevance for physics, is studying singularity resolutions in the context of $G_2$-holonomy spaces. For Calabi-Yau manifolds, the resolution (and deformation) of singularities often has a well-defined algebraic description, and there are simple criteria which guarantee that the resolved space continues to be Calabi-Yau. In contrast such theorems are rather sparse for singular $G_2$ spaces and limited to specific constructions such as the Joyce orbifolds (see e.g. [18]) and Calabi-Yau times circle orbifolds [19]. Furthermore, a systematic constructing large classes of $G_2$-manifolds is thus far limited to torus Joyce orbifolds or so-called twisted connected sums [20–25]. See also the construction of [26] where they find non-compact $G_2$ manifolds in a domain wall setup.

The truly exciting aspect of this proposal is its generality, as well as its close relation with the field theory construction: the geometric construction is completely dictated by the physics, and this in turn provides strong support for the existence of $G_2$-holonomy metrics on these spaces. Generically, whenever enhanced non-Abelian symmetries are present, these geometries will of course be singular. Although our focus here is primarily on topological considerations and we will not enter into the construction of torsion-free $G_2$-metrics, some examples that we will rediscover with this method are very well-known $G_2$-holonomy metrics [16, 27], thus providing strong support for this general expectation.

In Part II [28], we will use the domain wall constructions suggested in the current paper, and apply it to marginal 5d theories originating form 6d $(1, 0)$ SCFTs on a circle in order to construct 4d $\mathcal{N} = 1$ SCFTs. There has been an intense activity in recent years in the study of compactifications of 6d $(1, 0)$ SCFTs on Riemann surfaces with flux to give 4d $\mathcal{N} = 1$ SCFTs [4–6, 29–52] and our aim is to use these field theory results to systematically construct $G_2$ holonomy spaces. This requires stacking domain walls in a non-trivial manner from the field theory side, see [51]. This is expected to translate to non-trivial identifications of parameters on the gluing interfaces of the domain walls constructed here.

**Overview of results.** We start our analysis from the simple example of a massive free hyper-multiplet in 5d. The associated domain wall was studied thoroughly from the field theory point of view in [53]. The construction considers a free hyper-multiplet with varying mass $m(x_4)$

---

[1]On a nearly Kähler manifold we have $d\Omega = c J \wedge J$ so that $\int_S J \wedge J = 0$ for any closed surface $S$.

[2]More precisely, we mean that on the left of the domain wall $\phi_a^{(L)} = 0$ and on the right $\phi_a^{(R)} = 0$. The two are in general related by a non-trivial identification, see figure 1 for the $E_1$ case, so in particular $\phi^{(L)}$ will not vanish on the right side.

profile along the $x_4$ direction, which asymptotically behaves as $m(x_4 \to \pm\infty) = \pm c$ with $c \neq 0$ and correlated signs. It was shown that this results in a free chiral localized on the interface, where $m(x_4) = 0$. The 5d massive hyper-multiplet theory can be geometrically engineered in M-theory by the resolved conifold geometry. The varying mass $m(x_4)$ then instructs us to smoothly interpolate between two topologically inequivalent resolutions of the conifold, which can be thought of as fibering the conifold with varying Kähler parameter along the $x_4$ direction. At the interface, where $m(x_4) = 0$, we have the singular conifold geometry, and the transition between $x_4 < 0$ and $x_4 > 0$ is essentially a flop transition. The resulting geometry, should then allow a metric of $G_2$-holonomy and describe a free 4d chiral in the M-theory compactification. In this case this turns out to be a cone over $\mathbb{P}^3$, which is indeed known to have a $G_2$-holonomy metric. The boundary conditions of fixing the mass $m(x_4)$ to be positive on one side, and negative on the other, forces the continuously varying $m(x_4)$ to pass through zero such that there cannot be a deformation which gets rid of the domain wall. This will be an important feature in all the constructions considered in this work.

**Seiberg theories.** With the framework firmly established for the free hyper-multiplet, we move on to the 5d rank 1 $E_1$ SCFT. The construction of the $G_2$-manifold follows now a similar reasoning: the 5d $E_1$ theory is realized in M-theory in terms of a local Calabi-Yau-threefold constructed over the Hirzebruch $\mathbb{F}_0$ surface. This is the cone over the Sasaki-Einstein 5-manifold $Y^{2,0}$. Note that this is equivalently a $\mathbb{Z}_2$ orbifolding of the conifold. We fiber this Calabi-Yau along $x_4$, whilst varying the Kähler parameter as dictated by the domain wall construction, see figure 1. The two 5d theories are related in this case by an S-duality, which geometrically corresponds to two distinct partial resolutions, which realize $SU(2)_0$ gauge theories.

As is well-known, the $E_1$ theory has an alternative realization in terms of the Hirzebruch surface $\mathbb{F}_2$, whose geometry manifestly encodes the non-Abelian flavor symmetry enhancement in the UV. The corresponding Calabi-Yau is $C(Y^{2,2})$ and we show that this gives rise to the same $G_2$ geometry. This allows us to later generalize this construction to 5d $SU(N)_N$ gauge theories, given by placing M-theory on a partially resolved $C(Y^{N,N})$.

**SQCD.** Naturally, we can add flavor to this construction, which in the field theory construction of domain walls was considered in [3]. Within this class, we focus on the simplest duality domain walls, which preserve the maximal amount of flavor symmetry by setting all the masses of hyper-multiplets to be equal. The field theory analysis shows that these masses should not vanish in the proximity of the domain wall, where again the effective gauge coupling diverges. This makes it possible to engineer the 5d theories with manifest flavor symmetries from toric Calabi-Yau varieties. The fibration of these partially resolved Calabi-Yau three-folds results in 7-manifolds, which are closely related to those for the domain walls in pure gauge theories, however with some additional singularities, implied by the $\mathfrak{su}(N_F)$ sub-group of the flavor symmetry that is present along the fibration.

**UV-duality domain walls.** 5d SCFTs can admit different non-Abelian gauge theory descriptions along distinct walls of the extended Coulomb branch. Such gauge theories are then called UV-dual (see [54] for a large number of such UV dualities from field theory considerations). Geometrically, this means that there are distinct singular limits ("rulings") where complex surfaces collapse to curves of ADE-singularities (see [12] for an in-depth discussion). In toric models, these are detected very easily by the fact that the toric diagram allows for distinct IIA limits (see [11] for a detailed discussion). It is then natural to ask whether we can extend the above construction of domain walls to the case where the 5d theories on either side of the wall are in UV-dual frames. We answer this affirmatively and provide some examples for this using toric models. Clearly this opens up a vast new class of domain walls and constructions of $G_2$-spaces.

**Future directions.** The constructions that we proposed in this paper are largely exemplified using toric models (or toric phases for theories whose generic Coulomb branch description is not toric, such as the $E_{N_F+1}$ Seiberg theories with $N_F \geq 4$). We can clearly extract much more general lessons, which are applicable to any (geometrically realized) 5d SCFT – see [11,13,14,55–60] for examples of geometries with crepant resolutions, though not necessarily toric. Clearly it would be interesting to also incorporate those coming from elliptically fibered Calabi-Yau three-folds, which are not toric, but are vastly more general and also provide a complete description of the extended Coulomb branch in terms of the Kähler cone. Another extension is to generalized toric models [61], which in some instances have recently been shown to have a geometric description [62]. The condition of crepant resolvability of the singularity may also be dropped, as demonstrated with the conifold, and could be utilized to construct domain walls from 5d rank 0 theories [57,63,64].

**Plan of the paper.** The paper is organized as follows: in section 2 we construct $G_2$-holonomy manifolds from the domain wall in the hyper-multiplet theory, and discuss the geometry and metric of the resulting geometry. We also construct the geometries associated to stacking several domain walls. In section 3 we review 5d $\mathcal{N} = 1$ field theories, and discuss the general field theory construction of duality domain walls. In section 4 we construct $G_2$-holonomy geometries from the rank 1 $E_1$ Seiberg theory. We match the resulting geometry to the field theory expectations, and show two equivalent ways of constructing the geometry by fibering $\mathbb{F}_0$ or $\mathbb{F}_2$. In section 5 we generalize the $E_1$ theory case to the rank 1 $E_{N_F+1}$ SCFTs and the marginal rank 1 theory. In section 6 we expand the previous constructions to theories with $SU(N)_N$ low energy effective gauge description and the 5d SQCD theories $SU(N)_k + N_F \boldsymbol{F}$ with CS level $k = N - \frac{N_F}{2}$. In section 7 we propose a generalization of the field theory construction by considering UV-duality walls, i.e. walls in 5d SCFTs interpolating between different non-Abelian gauge theory descriptions, together with their geometric counterparts, focusing on the "beetle" and "millipede" SCFTs. The appendices contain an in depth discussion of the domain wall theory for the $E_3$ Seiberg theory, as well as the UV-dual domain wall for the $k = 4$ "millipede".

## 2 Domain wall geometries from conifolds

We begin with the simplest example of a domain wall in a 5d theory: the free hyper-multiplet, engineered by M-Theory on the resolved conifold, which undergoes a flop transition as we fiber this along the $x_4$ direction. We first describe this simplest of 5d theories in terms of the geometric realization as M-theory on the conifold geometry, and its complete moduli space. We assume the reader has some basic familiarity with toric geometry (for background see e.g. [65]).

### 2.1 The conifold and the 5d hyper-multiplet

A single 5d $\mathcal{N} = 1$ hyper-multiplet can be engineered by M-theory on the resolved conifold. The resolved conifold is a toric variety which has two small resolutions:

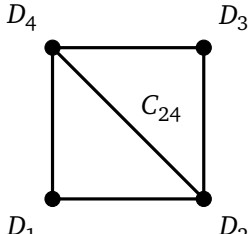 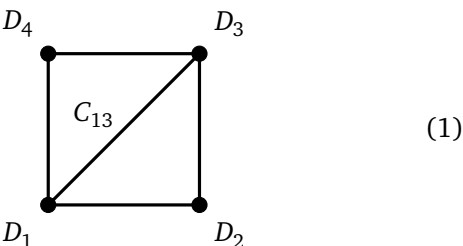 (1)

Here $D_i$ are non-compact toric divisors, and $C_{ij} = D_i \cdot D_j$ are rational curves (i.e. $\mathbb{P}^1$). The two distinct resolutions are related by a flop, i.e. blowing down the curve $C_{24}$ and resolving subsequently the curve $C_{13}$. The volume of the curves will be denoted by a Kähler parameter

$$\text{Vol}(C_{24}) = t = -\text{Vol}(C_{13}). \tag{2}$$

Compactifying M-theory on this geometry results in a mass $m = t$ hyper-multiplet in 5d.

For our purposes, it is useful to describe the resolved conifold as a symplectic quotient [66, 67], i.e. as the vacuum manifold of an $\mathcal{N} = 1$ gauged linear sigma-model (GLSM) with one $U(1)$ gauge group and four chiral fields with $U(1)$ charges

$$\begin{array}{c|cccc|c} & D_1 & D_2 & D_3 & D_4 & \text{FI} \\ \hline C_{24} & 1 & -1 & 1 & -1 & t \end{array}. \tag{3}$$

The resolved conifold is then given by

$$\mathcal{X}_C(t) = \{|z_1|^2 - |z_2|^2 + |z_3|^2 - |z_4|^2 = t;\ z_i \in \mathbb{C}\}/U(1), \tag{4}$$

where $z_i$ are the homogeneous coordinates associated with the divisors $D_i$ and the $U(1)$ acts with the charges in the above table.

The global symmetry of the theory (in addition to the $\mathfrak{so}(4,1)$ Lorentz symmetry) is $\mathfrak{su}(2)_R \oplus \mathfrak{su}(2)_F$, where the first part is the R-symmetry while the second is the flavor symmetry under which the two half-hyper-multiplets contained in the hyper-multiplet form a doublet. The mass parameter $m = t$ can be understood as the scalar component of a background vector multiplet for the $\mathfrak{su}(2)_F$ flavor symmetry.

## 2.2 The domain wall

We now discuss 4d $\mathcal{N} = 1$ walls located at $x_4 = 0$, which separate two hyper-multiplet theories in 5d. Such a set-up has been studied in detail in [53], where in particular it was shown that in order to have a non-trivial theory supported on the 4d interface one has to turn on a real mass $m(x_4)$ for the $\mathfrak{su}(2)_F$ symmetry with a non-trivial profile in the $x_4$ direction, such that it crosses zero at $x_4 = 0$. Hence, the mass is positive on one side, the left in our conventions for the entire paper, and negative on the other, changing its sign exactly at $x_4 = 0$.

This sign change corresponds to the $\mathbb{Z}_2$ Weyl-group action of the flavor symmetry algebra $\mathfrak{su}(2)_F$. This means that the two 5d theories on either side of the domain wall are equivalent, but they live in distinct Weyl chambers of the extended Coulomb branch, which in this simple case is parameterized by the mass $m$. This is the main feature of the construction of the domain wall, which generalizes: although the two 5d theories are equivalent, they are glued with a non-trivial identification by an element of the Weyl group of their UV global symmetry in $x_4 = 0$, which is the key ingredient to have 4d degrees of freedom living at this point.

The case of the free hyper-multiplet is particularly simple and, as shown in [53], the 4d degrees of freedom are just those of an $\mathcal{N} = 1$ chiral multiplet.

To implement the above field theory construction at the level of geometry, we only need to observe that the Kähler parameter $t$ corresponds in field theory to the mass of the free hyper, $m = t$. Hence, in order to realize the domain wall that we previously described field theoretically, we fiber the conifold geometry with $t = -x_4$ over $x_4 \in \mathbb{R}$ such that we obtain a 7-dimensional manifold

$$\mathcal{M}_C = \{\mathcal{X}_C(t) \mid t = -x_4 \in \mathbb{R}\}/U(1). \tag{5}$$

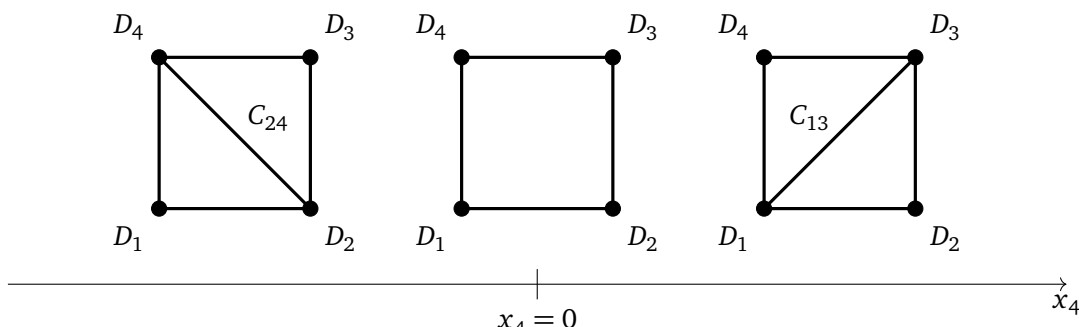

Figure 2: Fibration of the conifold: the coordinate $x_4$ is identified with the Coulomb branch parameter $t$. Effectively this fibers the resolved conifold, through its flop transition, with $x_4 = 0$ the singular conifold, over the Kähler moduli space. From the 5d perspective, this is a hyper-multiplet whose mass is now space-dependent.

Observe that on the right of the domain wall $t = -x_4 < 0$ so the volume of the curve $C_{24}$ becomes negative. This means that the geometry undergoes a flop-transition. In the flopped phase, the curve $C_{13}$ has a positive volume $-t = x_4 > 0$. At the point $x_4 = t = 0$, where the domain wall is located, we have the singular conifold. Hence, the overall picture is as shown in figure 2.

We can also describe $\mathcal{M}_C$ as a cone as follows. Note that (4) uniquely determines $x_4$ for every value of the $z_i$, so when constructing $\mathcal{M}_C$ as (5), the parameter $x_4$ can simply be dropped and we have

$$\mathcal{M}_C = \mathbb{C}^4/U(1). \tag{6}$$

To understand the topology of this space, let us introduce a new coordinate $r$ and write

$$\mathcal{M}_C = \left\{ \sum_i |z_i|^2 = r^2 \mid r \in \mathbb{R}_+ \right\} /U(1). \tag{7}$$

By considering $\bar{z}_3$ and $\bar{z}_4$ instead of $z_3$ and $z_4$ we can furthermore take the $U(1)$ above to act with charges $(1,1,1,1)$ on $\mathbb{C}^4$. For every value of $r$, the above then realizes $\mathbb{P}^3$ as a symplectic quotient. As $r$ ranges from 0 to $\infty$, we find that $\mathcal{M}_C$ can be written as a real cone over $\mathbb{P}^3$

$$\mathcal{M}_C = C(\mathbb{P}^3). \tag{8}$$

Let us examine how the two descriptions (5) and (7) of $\mathcal{M}_C$ are related. To see how $C(\mathbb{P}^3)$ is foliated by resolved conifolds, we reintroduce

$$-x_4 = |z_1|^2 - |z_2|^2 + |z_3|^2 - |z_4|^2. \tag{9}$$

Together with (7), $\mathcal{M}_C$ is then described by

$$\mathcal{M}_C = \left\{ \begin{array}{l} |z_1|^2 + |z_3|^2 = \frac{1}{2}\left(r^2 - x_4\right) \\ |z_2|^2 + |z_4|^2 = \frac{1}{2}\left(r^2 + x_4\right) \end{array} \right\} \Big/ U(1). \tag{10}$$

Fixing $r^2$ and $x_4$ such that $r^2 \pm x_4 > 0$, the above equations show the link of the conifold, $S^3 \times S^3/U(1) = S^2 \times S^3$. Whenever $r^2 \pm x_4 = 0$, one of the two $S^3$s has collapsed, leaving us with a single copy of $S^2$ sitting at the tip of the resolved conifold. For every $x_4$, we find a copy of the resolved conifold by letting $r^2$ range from $|x_4|$ to $\infty$, see figure 3.

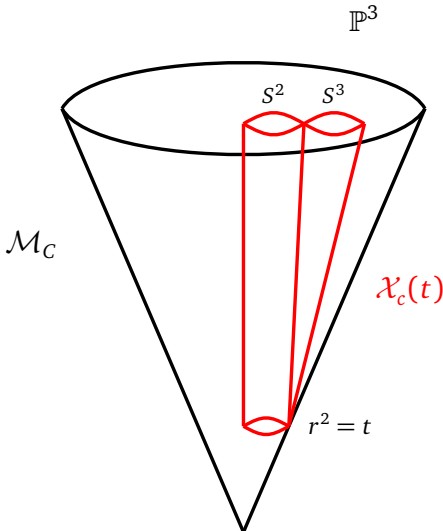

Figure 3: The cone $\mathcal{M}_C$ over $\mathbb{P}^3$, which is the $G_2$-manifold realizing the the domain wall of the conifold. The embedding of the copy of the resolved conifold $\mathcal{X}_c(t)$ is shown in red.

If we instead keep $r^2$ fixed and vary $x_4$, $S^3 \times S^3/U(1)$ fills out an entire $\mathbb{P}^3$: for fixed $r^2$, $x_4$ can only take values in the interval $|x_4| \leq r^2$ with one of the two $S^3$s collapsing at each end. This realizes a copy of $S^7$ with the $U(1)$ acting along the Hopf fiber, so that we find the space $S^7/U(1) = \mathbb{P}^3$ at fixed $r^2$, as expected.

## 2.3 The domain wall geometry as unfolding of $A_1$

The domain wall geometry has an equivalent description as an unfolding of a hyper-Kähler quotient, i.e. it appears as a special case of a construction of conical $G_2$ holonomy spaces proposed in [68] (see also the pedagogical discussion in [69]).

A hyper-Kähler quotient can be described in terms of the vacuum manifold of an $\mathcal{N} = 2$ GLSM, which is in turn given by setting to zero the triplet of D-terms (moment maps) modulo the action of the gauge group. Consider a theory with two hyper-multiplets $\Phi_1$ and $\Phi_2$, and a single $U(1)$ with charge assignments

$$\begin{array}{cc} \Phi_1 & \Phi_2 \\ \hline 1 & -1 \end{array}. \tag{11}$$

We can write $\Phi_1 = (z_1, z_2)$ and $\Phi_2 = (z_4, z_3)$ in terms of complex variables $z_i$ which have charges

$$\begin{array}{cccc} z_1 & z_2 & z_3 & z_4 \\ \hline 1 & -1 & 1 & -1 \end{array} \tag{12}$$

under the $U(1)$ action. The $\mathfrak{su}(2)_R$ triplet of D-term constraints is

$$\mu_i(\Phi) \equiv \Phi_1 \sigma_i \Phi_1 - \Phi_2 \sigma_i \Phi_2 = t_i, \tag{13}$$

where $\sigma_i$ are the Pauli matrices and the $t_i$ are a triplet of real FI parameters. In terms of the $z_i$ these equations are

$$z_1 z_2 - z_3 z_4 = t_1 + i t_2,$$
$$|z_1|^2 - |z_2|^2 + |z_3|^2 - |z_4|^2 = t_3. \tag{14}$$

For a fixed choice of the $t_i$ and after quotienting by the $U(1)$ action, this space is just a general smoothing of an $A_1$ singularity. It admits a Ricci-flat metric, the Eguchi-Hanson metric $g_{EH}$, where the $t_i$ are identified with the three parameters of $g_{EH}$.

The unfolding is now described by promoting the FI parameters $t_i$ to coordinates, so that we obtain a space of real dimension 7. As we can now simply solve (14) in terms of the $t_i$, this leaves the $z_i$ unconstrained and we find that the seven-manifold after unfolding is

$$\mathcal{M}_C = \mathbb{C}^4/U(1), \tag{15}$$

which is equal to a cone over $\mathbb{P}^3$ as we already saw above. Following the logic of [68] not only provides further evidence for $\mathcal{M}_C$ to admit a Ricci-flat $G_2$ metric, but furthermore implies that there is a single 4d $\mathcal{N}=1$ free chiral multiplet located at the singularity at the tip of the cone. This matches with the analysis of the 5d domain wall theory from field theory.

A further way to argue for the existence of this mode employs the IIA reduction of M-theory on $\mathcal{M}_C$. The presentation we have just given, (14), shows that we can think of $\mathcal{M}_C$ as a smoothed $A_1$ singularity fibered over $\mathbb{R}^3$ with coordinates $t_i$. This reduces in IIA to two D6-branes with four parallel directions. In the remaining six non-parallel directions, we can use $t_i$ as coordinates on one of the two D6-branes. Letting $n_i$ be the coordinates of the transverse directions, the second D6-brane is then displaced by $n_i = t_i$ over every point. These two D6-branes hence intersect along a copy of $\mathbb{R}^{1,3}$ at $t_1 = t_2 = t_3 = 0$ and consequently support a single 4d $\mathcal{N}=1$ chiral multiplet localized there. Using the general logic of M-theory on $G_2$-manifolds this also implies that there must be an isolated metric singularity at the tip of the cone $\mathcal{M}_C$.

We can also see this from the domain wall geometry directly. Reducing M-theory on $\mathcal{X}_C(t)$ to IIA by using the Hopf fiber on the $S^3$ contained in the cross section of the conifold tells us that the M-theory circle collapses over the $S^2$ at the tip of the resolved conifold. We hence find a single D6-brane with worldvolume $S^2$ of radius $|x_4|$. In the domain wall geometry where we fiber this over $x_4$, there are hence two D6-branes, each of which sits on a real cone over $S^2$. We hence recover the IIA picture of two D6-branes on $\mathbb{R}^3$ touching at a point.

## 2.4 The metric on the domain wall solution

So far we have discussed the domain wall geometry as a topological manifold $\mathcal{M}_C$ and found that it can be described as a real cone $C(\mathbb{P}^3)$ with a singularity at its apex. There is a known conical metric on $C(\mathbb{P}^3)$ which goes back to the seminal work by Bryant and Salamon [16]. As discussed in [27], this metric exactly captures the M-Theory lift of a configuration of two D6-branes on copies of $\mathbb{R}^3$ that are sitting inside $\mathbb{R}^6$, and are touching at a point. At the same time, $\mathcal{M}_C$ can be obtained as a fibration of a deformed $A_1$ singularity, which is topologically equivalent to $T^*S^2$, over $\mathbb{R}^3$. Over the origin of $\mathbb{R}^3$, the fiber degenerates by collapsing the section of $T^*S^2$. A description of the Bryant–Salamon metric in terms of this fibration has recently been given in [17]. Restricting the Bryant–Salamon metric to the coassociative $T^*S^2$ fibers does not imply the flat hyper-Kähler metric, but only a hypersymplectic structure (first introduced in [70]). This is closely related to Donaldson's work on coassociative Kovalev–Lefschetz fibrations [71] which suggests that this is a specific example of a more general story.

The Bryant–Salamon metric on $C(\mathbb{P}^3)$ is the singular limit of a smooth, asymptotically conical $G_2$ manifold which is topologically equal to the bundle of anti-self-dual 2-forms on $S^4$, $\Lambda^2_-(T^*S^4)$. In other words, $\Lambda^2_-(T^*S^4)$ carries a real one-parameter family of $G_2$ structure $\varphi_c$ for all $c \geq 0$. When $c = 0$, $\Lambda^2_-(T^*S^4)$ degenerates to $C(\mathbb{P}^3)$. We can hence deform our domain wall geometry to a smooth $G_2$ manifold. As detailed in [27], this is triggered by giving a vev to the scalar in the chiral multiplet, which can be written as

$$\Phi = c\, e^{i \int_F C_3}, \tag{16}$$

where $F$ is the homology class of the fibers of $\Lambda^2_-(T^*S^4)$ and $c$ measures the volume of $S^4$. As the singularity at the tip and the associated chiral $\Phi$ is due to two intersecting D6-branes, a vev of $\Phi$ signals their recombination into a single object, which was found to have the topology of $S^2 \times \mathbb{R}$. This can also be seen from the structure of the fibration by coassociatives. When $c > 0$, the locus of degeneration of the $T^*S^2$ fibers is no longer an isolated point but becomes a circle [17]. The compact $S^2$ in the fiber is associated with the Cartan of $A_1$ and measures the distance of the two D6-branes, so that the monodromy upon encircling the $S^1$ swaps the two D6-branes. We can hence describe the recombined D6-brane locus as a double cover of $\mathbb{R}^3$ branched over $S^1$, which is again equal to $S^2 \times \mathbb{R}$.

## 2.5 Stacking domain walls and transitions

We now consider a configuration in which we stack several of the previously discussed domain walls. Roughly speaking, what happens is that pairs of adjacent domain walls annihilate each other, such that we are left with either no domain wall if the number of stacked domain walls is even or a single domain wall if it is odd. In the absence of any domain wall, we expect to recover the 5d theory of one massive free hyper-multiplet sitting over a line, which we confirm geometrically.

More precisely, the field theory picture of the stacking of $n$ domain walls is that we have the 5d free hyper-multiplet fibered over $x_4$ with a mass $m(x_4)$ whose profile is such that it crosses zero at the $n$ points where the domain walls are located, going from positive to negative, and then back to positive again, and so on. Hence, if $n$ is even then $m$ has the same sign both at $x_4 = -\infty$ and $x_4 = +\infty$, so according to the analysis of [53] there is no non-trivial 4d interface and we just have the massive 5d hyper-multiplet fibered trivially over $x_4$. Field theoretically, this is understood as having two chirals with opposite charges under their $\mathfrak{u}(1)$ flavor symmetry, reconstructing a full massive hyper. On the contrary, if $n$ is odd then the sign of $m$ at $x_4 = -\infty$ and $x_4 = +\infty$ is opposite and we expect overall to get a single 4d $\mathcal{N} = 1$ chiral. Field theoretically, we have $n$ chirals that give mass to each other in pairs, leaving a single chiral since $n$ is odd.

This field theory expectation is nicely realized in our geometric set up as we are now going to explain. Intuitively, we would like to glue two (or more) copies of $\mathcal{M}_C$ and the way we do this is by using the fibration by $\mathcal{X}_C(t)$. What we want to do is to assume that $t$ does not approach infinity but a constant on one of the two sides for each of the domain walls, so that these can then be identified. This implies that we can think of the resulting space again as a fibration of the conifold over $\mathbb{R}$, but now we change twice from one phase into another, see figure 4.

This can be elegantly described as follows. Instead of choosing the Kähler parameter $t$ of the resolved conifold to be equal to the coordinate $x_4$ on $\mathbb{R}$, we can assume it is a non-trival function, e.g. we can set

$$t = x_4^2 - a\,, \tag{17}$$

and define

$$\mathcal{M}_{2C}(a) = \left\{ \mathcal{X}_C\left(x_4^2 - a\right) \mid x_4 \in \mathbb{R} \right\}. \tag{18}$$

For positive $a$, the Kähler parameter $t$ hence changes sign twice along $x_4$, each time in a different direction. Notice that this precisely mimics the behaviour of the mass parameter.

Explicitly, $\mathcal{M}_{2C}$ is described by

$$\mathcal{M}_{2C}(a) = \left\{ |z_1|^2 - |z_2|^2 + |z_3|^2 - |z_4|^2 = x_4^2 - a\,;\ x_4 \in \mathbb{R} \right\}/U(1). \tag{19}$$

Note that we now cannot simply solve for $x_4$ as before, contrary to the situation with just a single domain wall. The reason is that whereas $x_4 \in \mathbb{R}$, $x_4^2 - a$ is bounded below by $-a$, and for some values of the $z_i$ the equation above has no solution.

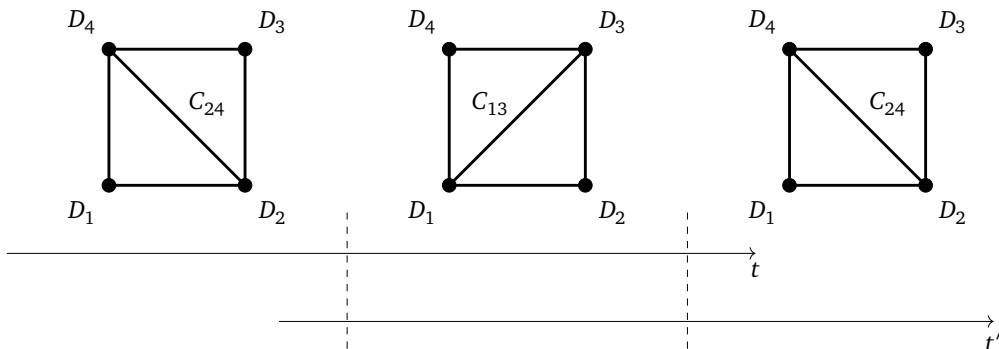

Figure 4: Stacking of domain walls: gluing the domain walls for the hyper-multiplet $\mathcal{M}_C(t)$ to a second copy $\mathcal{M}_C(t')$.

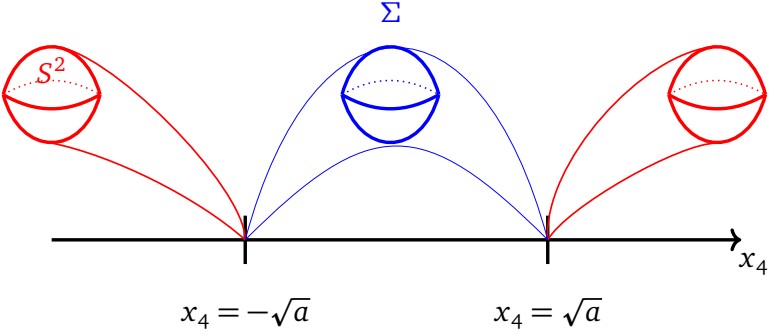

Figure 5: Geometry associated to stacking two domain walls: As the compact $S^2$ in $\mathcal{X}_C$ is fibered over $x_4$, it forms two non-compact thimbles when $x_4 > |a|$ and a compact $S^3$, $\Sigma$, when $x_4 < |a|$. A similar picture also holds for the rank 1 $E_1$ domain walls which we will discuss later on.

Decomposing $\mathcal{M}_{2C}$ in terms of $\mathcal{X}_C$, every fiber contains a $S^2$ with volume $|x_4^2 - a|$. Hence there is now a compact $S^3$ denoted by $\Sigma$ formed by fibering the compact $S^2$ in every $\mathcal{X}_C$ over $x_4 \in [-\sqrt{a}, \sqrt{a}]$. The $S^3$ is capped off by the two domain wall singularities located at $x_4 = \pm\sqrt{a}$. Locally, each of these singularities is (by construction) equivalent to the singularity found in $\mathcal{M}_C$. We can see this $S^3$ explicitly by noting that the $S^2$ in $\mathcal{X}_C$ for $x_4 \in [-\sqrt{a}, \sqrt{a}]$ is $z_1 = z_3 = 0$, i.e. it is given by

$$\Sigma = \left\{ |z_2|^2 + |z_4|^2 + (x_4)^2 = a \right\} / U(1). \tag{20}$$

Note that the $U(1)$ just acts on the coordinates $z_2$ and $z_4$ above. The two singularities of $\mathcal{M}_{2C}$ are sitting at the north and south-pole of $\Sigma$, $(z_2, z_4, x_4) = (0, 0, \pm\sqrt{a})$. The compact $S^2$ in $\mathcal{X}_C$ fibered over $x_4$ is sketched in figure 5.

As this stacked domain wall depends on the parameter $a$ controlling the distance of the two domain walls, it is natural to think about what happens when this parameter varies. Letting $a$ go to zero collides the two domain walls which then annihilate. From the 5d physics point of view this can be understood from the mass parameter of the theory never crossing zero. From the 4d physics point of view this can be thought of as probing longer distances, making the distance between the two domain walls be comparatively smaller, corresponding to moving towards the infra-red (IR).[3] From the geometry it can be seen as follows: when $a < 0$ the

---

[3]Notice, that $a$ has mass scale one, meaning it is an energy scale. When we write it in the expression $x_4^2 - a$ it

expression $x_4^2 - a$ is strictly positive for all $x_4$. We can hence rescale all of the coordinates $z_i$ by setting $z_i = z_i' \sqrt{x_4^2 - a}$ such that (19) becomes

$$\mathcal{M}_{2C}(a)|_{a<0} = \left\{ |z_1'|^2 - |z_2'|^2 + |z_3'|^2 - |z_4'|^2 = 1 \,\middle|\, x_4 \in \mathbb{R} \right\} / U(1), \tag{21}$$

after dividing by $x_4^2 - a$. This is just a product of $\mathcal{X}_C(1)$ with a copy of $\mathbb{R}$ with coordinate $x_4$.

A similar analysis can be done for the stacking of $n$ domain walls. In this case we let

$$\mathcal{M}_{nC}(P_n) = \left\{ \mathcal{X}_C \left( P_n(x_4) \right) \mid x_4 \in \mathbb{R} \right\}, \tag{22}$$

with $P_n(x_4) = \prod_{i=1}^{n}(x_4 - a_i)$. The domain walls are located at $x_4 = a_i$ for $i = 1, \cdots, n$. Deforming the polynomial $P_n(x_4)$ the domain walls can pairwise annihilate leaving either a single domain wall for $n$ odd or no domain wall for $n$ even.

# 3 Field theory construction of 4d $\mathcal{N} = 1$ domain walls

In this section we review some aspects of 5d $\mathcal{N} = 1$ theories, with a particular focus on the structure of their extended Coulomb branches. We will then discuss what are the key ingredients in the field theory construction of the 4d $\mathcal{N} = 1$ domain walls between 5d $\mathcal{N} = 1$ theories described in [3], which will then allow us to construct $G_2$ geometries in the next sections.

## 3.1 5d SCFTs and extended Coulomb branches

Our starting point are SCFTs in 5d with $\mathcal{N} = 1$ supersymmetry and superconformal algebra $\mathfrak{f}_4$ [72]. The bosonic subgroup is $\mathfrak{so}(5,2) \oplus \mathfrak{su}(2)_R$, where the first part is the conformal group in 5d and the second part is the R-symmetry. 5d SCFTs have a moduli space of vacua that consists of two main branches: the Higgs branch (HB) where $\mathfrak{su}(2)_R$ is completely broken, and the Coulomb branch (CB) where $\mathfrak{su}(2)_R$ is preserved.

In 5d an SCFT cannot be realized as the low energy limit of the renormalization group (RG) flow of a Lagrangian gauge theory since the latter is IR free. Nevertheless, starting from the pioneering work [1] it has been understood that 5d SCFTs do exist, but because of their intrinsic strongly coupled nature an embedding in string theory is paramount to substantiating their existence. As we have already seen in the previous section, in the present paper we are particularly interested in their realization in M-theory on canonical Calabi–Yau three-fold singularities, as originally explained in [8].

5d SCFTs admit mass deformations that can make them flow either to another SCFT or to a low energy gauge theory phase. This latter fact will be of particular interest to us. 5d $\mathcal{N} = 1$ gauge theories also admit a HB and a CB in their moduli spaces of vacua. The latter, which is what we will focus on, is in particular parameterized by the real scalars inside the vector multiplets.

Suppose that the 5d SCFT admits a gauge theory phase with gauge algebra $\mathfrak{g}$ of rank $r$ which we assume for simplicity to contain only one simple factor.[4] The CB of the theory is parameterized by the vacuum expectation value (vev) for the scalars in the vector multiplets for the Cartan of the gauge group, which is isomorphic to

$$\mathcal{C} = \mathbb{R}^r / W_{\mathfrak{g}}, \tag{23}$$

where $W_{\mathfrak{g}}$ is the Weyl group of $\mathfrak{g}$.

---

now appears to have units of length square or one over energy square. This is just because there should be a mass scale $M$ inserted as $x_4^2 - \frac{a}{M^3}$ that we ignore.

[4]We will mostly focus on these cases in the present paper, except for the last section where we will also consider a quiver gauge theory. Nevertheless, the discussion including the construction of the domain wall can be generalized to any gauge theory with gauge algebra containing several factors, which can also be Abelian.

On a generic point of the CB, the gauge group is broken by the vector multiplet scalar vevs to its Cartan subgroup. The effective low energy theory is then described by a $U(1)^r$ Abelian gauge theory with vector multiplet scalars $\phi_a$ for $a = 1, \cdots, r$, whose dynamics is controlled by a *prepotential* [1, 8]

$$\mathcal{F}_{\text{IMS}} = h_0 h_{ab} \phi^a \phi^b + \frac{c_{\text{cl}}}{6} d_{abc} \phi^a \phi^b \phi^c + \frac{1}{12} \left( \sum_{\vec{\alpha}} |\vec{\alpha} \cdot \vec{\phi}|^3 - \sum_i \sum_{\vec{w}_i \in \mathcal{R}_i} |\vec{w} \cdot \vec{\phi} + m_i|^3 \right), \quad (24)$$

where the subscript IMS indicates that we are using the conventions of [8]. In the previous expression, we have defined

$$h_{ab} = \text{Tr}\, T_a T_b, \qquad d_{abc} = \frac{1}{2} \text{Tr}\, T_a (T_b T_c + T_c T_b), \quad (25)$$

with $T_a$ the generators of the gauge algebra $\mathfrak{g}$, $h_0$ is related to the bare gauge coupling $g_0$ as

$$h_0 = \frac{1}{2 g_0^2}. \quad (26)$$

$c_{\text{cl}}$ is the classical Chern–Simons (CS) level which is only relevant for $SU(N)$ with $N \geq 3$ since otherwise $d_{abc}$ is trivial, and $\vec{\alpha}$ are the roots of $\mathfrak{g}$. We also included matter fields coming from hyper-multiplets, labeled by $i$, that transform in representations $\mathcal{R}_i$ of the gauge algebra $\mathfrak{g}$, whose corresponding weights are $\vec{w}_i$. The parameters $m_i$ correspond to mass parameters for the hyper-multiplets, which can be thought of as scalars in background vector multiplets for the flavor symmetry that acts on the hypers.

The set of parameters $\{\phi_a, m_i\}$ parameterize what is known as the *extended Coulomb branch* (ECB), as opposed to the standard CB which is only parameterized by the $\phi_a$. The chamber structure of the ECB can be characterized as follows. First, it is customary to choose a Weyl chamber cone $\mathcal{C}$ such that

$$\mathcal{C} = \left\{ \vec{\phi} \in \mathbb{R}^r \mid \vec{\alpha} \cdot \vec{\phi} \geq 0 \right\}. \quad (27)$$

This gets then further partitioned into sub-chambers where each combination $\vec{w} \cdot \vec{\phi} + m_i$ appearing in (24) has a definite sign. Such a combination is interpreted as the mass of the $i$-th hyper-multiplet. We then see that on a generic point of the ECB the hyper-multiplets have a non-vanishing mass, but on special points that correspond to walls separating two different chambers some of them become massless. A complete systematic description of all ECB phases was developed in [9].

The parameter $h_0$ also has the interpretation of a mass, but it is associated with a $U(1)_I$ symmetry called *instantonic symmetry*, whose presence is thanks to the fact that in 5d we can always define a conserved current as

$$J^\mu = \epsilon^{\mu\nu\rho\sigma\kappa} \text{Tr}\, F_{\nu\rho} F_{\sigma\kappa}. \quad (28)$$

The particle that is charged under this symmetry and whose mass $m_0$ is related to $h_0 = m_0/2$ is the instanton. We can then further refine the ECB by $h_0$, that is we now parameterize it by $\{\phi_a, h_0, m_i\}$.

One can reach an SCFT point by going to the origin of the ECB where all the parameters are set to zero. This can be seen from the fact that the effective gauge coupling $g_{\text{eff}}$, which can be computed from the prepotential (24) as

$$\frac{h_{ab}}{g_{\text{eff}}^2} = \frac{\partial^2 \mathcal{F}}{\partial \phi^a \partial \phi^b}, \quad (29)$$

diverges at this point, signaling that the low energy gauge theory description is no longer valid. Notice in particular that at the SCFT point the instanton also becomes massless. We point out that in order for $g_{\text{eff}}$ to diverge it is not strictly necessary for all the parameters to vanish, as this can also happen by tuning the parameters to specific values as we will see in explicit examples later on. We should emphasize that the the prepotential as written in (24) is only valid for $\frac{1}{g_{\text{eff}}^2} \geq 0$. Thus, when shifting some of the parameter values such that $\frac{1}{g_{\text{eff}}^2} < 0$, one will need to shift to a different expression corresponding to a different Weyl sub-chamber, as we discuss in the next subsection.

## 3.2 Domain walls

With the above ingredients, we can now characterize the general field theory construction of the domain walls between 5d theories of [3].

We take two copies of the same 5d theory, one of which lives in the region $x_4 < 0$ denoted by $(L)$ while the other lives in the region $x_4 > 0$ denoted by $(R)$, separated by the domain wall at $x_4 = 0$. The 5d theory in consideration is often the gauge theory phase of some UV SCFT, with all of the scalar vevs that parameterize the CB set to zero on both sides, $\phi_a^{(I)} = 0$ with $I = L, R$, such that we have the full, typically non-Abelian, gauge symmetry.

There is an interface located at $x_4 = 0$ which separates the two 5d theories, and some 4d degrees of freedom might reside on it. This interface is non-trivial only if the two 5d theories are glued in a non-trivial way across the interface, that is with some mapping between their parameters. In such a case we call the interface a *domain wall*. The parameters in question are the masses we introduced in the previous subsection, where we define them usually on the left side and then express the right side parameters in the vicinity of the domain wall in terms of the left parameters[5]

$$
\begin{aligned}
h_0^{(R)}(x_4 > 0) &= f_0\left(h_0^{(L)}(-x_4), m_i^{(L)}(-x_4)\right), \\
m_i^{(R)}(x_4 > 0) &= f_i\left(h_0^{(L)}(-x_4), m_i^{(L)}(-x_4)\right).
\end{aligned}
\tag{30}
$$

The crucial point of the construction of [3] is that such a transformation should be part of the symmetry of the UV SCFT, which is not manifest in the gauge theory description. As we will see in many explicit examples later on, this will be a $\mathbb{Z}_2$ transformation that is part of the Weyl group of the UV symmetry.

This $\mathbb{Z}_2$ transformation that is used when juxtaposing the two 5d theories will typically act as a sign change for one combination of the mass parameters that is related to the effective gauge coupling of the gauge theory on one side of the interface. We will fix this combination to be the one related to left side effective gauge coupling such that[6]

$$
m_\lambda^{(L)}(x_4 < 0) = \frac{1}{2\left(g_{\text{eff}}^{(L)}\right)^2}.
\tag{31}
$$

More precisely, $m_\lambda$ will be mapped across the interface as

$$
m_\lambda^{(R)}(x_4 > 0) = -m_\lambda^{(L)}(-x_4),
\tag{32}
$$

which should then be supplemented by the transformation of the hyper-multiplet masses $m_i$ to have a Weyl symmetry transformation.

---

[5]In this section we write explicitly the dependence on $x_4$ of these parameter mappings, but these will become implicit in the following sections to make the notation lighter.

[6]Note that this combination will not correspond to $\frac{1}{2g_{\text{eff}}^2}$ on the other side as this cannot be negative. We will comment more on this momentarily.

**Gluing condition.** Given that $m_\lambda$ is related to the effective gauge coupling squared, it might seem strange at first sight that the transformation changes its sign. The reason for this is that the $\mathbb{Z}_2$ transformation sends us into a different Weyl chamber of the ECB where in particular the standard form of the IMS prepotential (24) is not valid. Hence, in order to have a sensible prepotential on the right side of the wall we also need to use different CB parameters $\phi_a$ with some mapping across the interface

$$\phi_a^{(R)}(x_4 > 0) = f_{\phi_a}\left(\phi_a^{(L)}(-x_4), h_0^{(L)}(-x_4), m_i^{(L)}(-x_4)\right). \tag{33}$$

In other words, after the transformation (30) we can still get a sensible prepotential describing a gauge theory provided that we employ the new Coulomb branch variables $\phi_a^{(R)}$. To be more explicit, by using the transformation above one should find that

$$\mathcal{F}_{\text{IMS}}^{(L)}\left(h_0^{(L)}, m_i^{(L)}, \phi_a^{(L)}\right) = \mathcal{F}_{\text{IMS}}^{(R)}\left(h_0^{(R)}, m_i^{(R)}, \phi_a^{(R)}\right), \tag{34}$$

up to irrelevant constant terms. This however does not mean that $h_0^{(R)}$ and $m_i^{(R)}$ are the physical instanton and masses of hyper-multiplets on the right, since the prepotential will be reorganized in a way that the picture is completely mirrored across the domain wall. For example, the physical masses of hyper-multiplets on the right side will simply be $m_i^{(L)}(-x_4)$ and so on.

In a similar manner, this also implies that the definition of the effective gauge coupling will be different on the right side of the wall compared to the left side and in particular one finds

$$x_4 < 0: \quad \frac{1}{\left(g_{\text{eff}}^{(L)}(x_4)\right)^2} = m_\lambda^{(L)}(x_4) > 0, \qquad x_4 > 0: \quad \frac{1}{\left(g_{\text{eff}}^{(R)}(x_4)\right)^2} = m_\lambda^{(L)}(-x_4) > 0. \tag{35}$$

Notice that this issue is related to the fact we previously pointed out that the IMS prepotential is not valid on the entire ECB, which is ultimately due to the fact that it is not invariant under the full Weyl group of the UV symmetry. An alternative form of the prepotential that is manifestly Weyl invariant exists as discussed in [73] and we will encounter it in sections 4 and 6.

Similarly to what we did in section 2 when discussing the domain wall associated with the conifold, it is useful to realize the configuration that we just described as the 5d gauge theory fibered over the direction $x_4$ with a non-trivial profile for the mass parameters $h_0(x_4)$, $m_i(x_4)$ in such a way that their values at $x_4 < 0$ and $x_4 > 0$ are related by the Weyl transformation (30). In particular from (32) we see that the profile for $m_\lambda(x_4)$ is such that it has opposite sign between the two sides of the domain wall and it vanishes at its location

$$m_\lambda(x_4 < 0) > 0, \qquad m_\lambda(x_4 = 0) = 0, \qquad m_\lambda(x_4 > 0) < 0. \tag{36}$$

This can be understood as moving in the ECB as $x_4$ varies, in such a way that at $x_4 = 0$ we cross a wall which separates two different sub-chambers that are related by the $\mathbb{Z}_2$ Weyl transformation. Remembering (31), the above profile of $m_\lambda$ also implies that the effective gauge coupling $g_{\text{eff}}$ diverges on the domain wall, indicating a strong coupling behaviour at that point which is the origin for it hosting some non-trivial 4d dynamics.

The profile for the mass parameters along $x_4$ has the effect of breaking supersymmetry by a half, so that on the domain wall we have a 4d $\mathcal{N} = 1$ theory. In particular, the $\mathfrak{su}(2)_R$ R-symmetry of the original 5d theory is broken to its $\mathfrak{u}(1)_R$ Cartan. This is not necessarily the IR superconformal R-symmetry as in 4d $\mathcal{N} = 1$ theories it can mix with all the Abelian global symmetries of the model. The exact superconformal R-symmetry is determine by $a$-maximization [74].

**UV duals.** In general one can also consider a domain wall between two different gauge theories that are UV completed by the same 5d SCFT, so-called UV duals. In such a case there will be some mapping of parameters across the domain wall that will transform between the two theories prepotential. The crucial point will be that if we choose

$$m_\lambda(x_4 < 0) = m_\lambda^{(L)} = \frac{1}{2\left(g_{\text{eff}}^{(L)}\right)^2}\,, \tag{37}$$

then when $m_\lambda$ crosses to negative values it will be proportional to $-1/\left(g_{\text{eff}}^{(R)}\right)^2$. We will discuss such examples in section 7.

# 4  $G_2$-geometries from the $E_1$ Seiberg theory

We will now construct geometries, which realize in M-theory, the domain walls for the simplest possible 5d $\mathcal{N} = 1$ gauge theory, that is $SU(2)_0$, which is UV completed by the rank 1 $E_1$ Seiberg theory.

## 4.1  The field theory

The 5d rank 1 $E_1$ SCFT admits a mass deformation to an $SU(2)_0$ gauge theory with no flavors, where the subscript indicates that the theta angle is set to zero. The IMS prepotential is

$$\mathcal{F}_{SU(2)_0} = h_0\phi^2 + \frac{4}{3}\phi^3\,. \tag{38}$$

Here

$$h_0 = \frac{1}{2g_0^2}\,, \tag{39}$$

is related to the mass of the instanton particle. This is charged under the $U(1)_I$ instanton symmetry, which gets enhanced to the flavor symmetry $SO(3)$ at the SCFT point [75–77], with $g_0$ the bare gauge coupling. As we already commented in the previous section, (38) only makes sense for $h_0 > 0$ and so it is not invariant under the full Weyl group of the UV flavor symmetry $SO(3)$. Instead, $\phi$ is the single CB parameter in the Cartan of the $SU(2)$ gauge symmetry.

The mass parameter $m_\lambda$ that we introduced in subsection 3.2 is identified with

$$m_\lambda = h_0\,. \tag{40}$$

Hence, the $\mathbb{Z}_2$ transformation that relates the 5d theories on the two sides of the domain wall is just[7]

$$h_0^{(L)} \to h_0^{(R)} = -h_0^{(L)}\,. \tag{41}$$

This corresponds to the non-trivial transformation of the UV $SO(3)$ Weyl group. Equivalently, we can think that we have the 5d $SU(2)_0$ gauge theory with the instanton mass $h_0(x_4)$ turned on with a profile in the $x_4$ direction such that it crosses zero at $x_4 = 0$, being positive on the left region and negative on the right.

As we pointed out in the general discussion of subsection 3.2, if we apply such a transformation to the IMS prepotential (38) we get one for the theory on the right side of the domain wall that does not make sense as the prepotential of an $SU(2)_0$ gauge theory. This is once again due to the fact that (38) only holds for positive instanton mass, while on the right this

---

[7]Compared to the general expression (30) we are dropping the $x_4$ dependence to make the notation lighter.

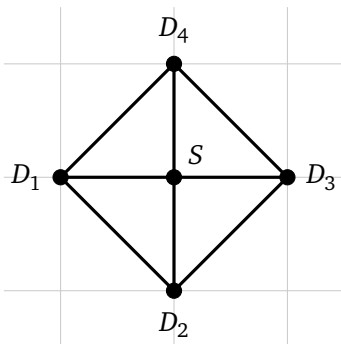

Figure 6: Fully resolved toric polygon for $\mathbb{F}_0$, which realizes the $SU(2)_0$ theory. $D_i$ denote non-compact divisors and $S$ is the compact divisor, given by $\mathbb{F}_0$.

becomes negative due to (41). Hence, in order to describe this side we need to introduce a new CB parameter

$$\phi^{(L)} \to \phi^{(R)} = \phi^{(L)} - \frac{h_0^{(L)}}{2}\,. \tag{42}$$

With this further identification, also on the right side we have a sensible prepotential for the $SU(2)_0$ gauge theory. Nevertheless, we will in the following describe both sides in terms of the same parameter $\phi^{(L)}$, which leads to it having an apparently weird behaviour when moving to the right side. Specifically, while on the left we need to set $\phi^{(L)} = 0$ to have the $SU(2)$ gauge theory, on the right the proper parameter that we should set to zero to still have a sensible $SU(2)$ gauge theory is $\phi^{(R)} = 0$, which in terms of $\phi^{(L)}$ means $\phi^{(L)} = \frac{1}{2}h_0^{(L)}$.

## 4.2 The geometry of the rank 1 $E_1$ theory

The 5d rank 1 $E_1$ SCFT is realized in M-theory by the complex cone over the Hirzebruch surface $\mathbb{F}_0 = \mathbb{P}^1 \times \mathbb{P}^1$

$$\mathcal{X}_{E_1} = K_{\mathbb{P}^1 \times \mathbb{P}^1}\,, \tag{43}$$

whose full resolution is described by the toric diagram shown in figure 6.[8]

There are two linearly independent non-compact divisors $D_i$, $i = 1, 2$ and one compact divisor $S$. We can describe this geometry in terms of an $\mathcal{N} = 1$ GLSM with two $U(1)$'s and the charge assignments

| | $D_1$ | $D_2$ | $D_3$ | $D_4$ | $S$ | FI |
|---|---|---|---|---|---|---|
| $C_{2S}$ | 1 | 0 | 1 | 0 | $-2$ | $t_1$ |
| $C_{1S}$ | 0 | 1 | 0 | 1 | $-2$ | $t_2$ |

$$\tag{44}$$

The vacuum equations are

$$\mathcal{X}_{E_1}(t_1, t_2) = \left\{ \begin{array}{l} |z_1|^2 + |z_3|^2 - 2|z_0|^2 = t_1 \\ |z_2|^2 + |z_4|^2 - 2|z_0|^2 = t_2 \end{array} \right\} \Big/ U(1) \times U(1), \tag{45}$$

where $z_i$ are the homogeneous coordinates associated to $D_i$, $z_0$ is the one associated to $S$, and the two $U(1)$'s act on them as in the table above. The FI parameters $t_1$, $t_2$ encode the volumes of the curves $C_{2S}$ and $C_{1S}$, respectively.

---

[8]We will not show the full 3d toric diagram, but only the $z = 1$ cross-section – all points lie on this plane, thanks to the Calabi-Yau condition.

There are equivalent GLSM's that describe the same geometry, but we choose this one in order to follow the conventions of [11], where the precise mapping between the FI parameters of the GLSM and the ECB parameters of the 5d theory has been worked out

$$h_0 = t_1 - t_2, \qquad \phi = \frac{t_2}{2}. \tag{46}$$

To obtain this mapping, one compares the IMS prepotential (38) with the one from the geometry $\mathcal{F}_{\text{geom}} = -\frac{1}{6}J^3$, finding the Kähler form[9]

$$J = h_0 D_1 - \phi S. \tag{47}$$

Using this Kähler form, we can work out the volume of the curves $C_{iS} = D_i \cdot S$ by intersecting with the Kähler form (more precisely the dual divisor) $\text{Vol}(C_{iS}) = D_i \cdot S \cdot J$,

$$
\begin{aligned}
t_1 &= \text{Vol}(C_{2S}) = h_0 + 2\phi, \\
t_2 &= \text{Vol}(C_{1S}) = 2\phi.
\end{aligned}
\tag{48}
$$

## 4.3 The domain wall geometry

Following our previous field theory discussion, we need the various parameters to have the following non-trivial profile along $x_4 \in \mathbb{R}$:

$$h_0 = -x_4, \qquad \phi = \frac{|h_0| - h_0}{4}. \tag{49}$$

Using the above identifications, this means in terms of the geometric parameters

$$t_1 - t_2 = h_0 = -x_4, \qquad t_2 = 2\phi = \frac{|x_4| + x_4}{2}. \tag{50}$$

We will denote the geometry $\mathcal{X}_{E_1}(t_1, t_2)$ with these specializations for fixed $x_4$ by $\mathcal{X}_{E_1}(-x_4)$. It is useful to re-write its defining equations (45) as

$$
\begin{aligned}
|z_1|^2 - |z_2|^2 + |z_3|^2 - |z_4|^2 &= t_1 - t_2 = h_0, \\
|z_2|^2 + |z_4|^2 &= 2|z_0|^2 + 2\phi,
\end{aligned}
\tag{51}
$$

where the first equation is now given by the combination $C_{2S} - C_{1S}$. For $x_4 < 0$, this implies that we have blown down the divisor $S$ together with the curves $C_{1S}$ and $C_{3S}$. The second equation in (51) uniquely determines the modulus of $z_0$ in terms of the other coordinates, and we can use its phase to gauge fix[10] the $U(1)$ associated with the FI parameter $t_2$. This leaves a residual $\mathbb{Z}_2$ as the weight of $z_0$ is $-2$, and we end up with the defining equation of the resolved conifold $\mathcal{X}_C(-x_4)/\mathbb{Z}_2$ subject to a $\mathbb{Z}_2$ acting as $(z_2, z_4) \to (-z_2, -z_4)$. When $x_4 > 0$, we are describing the flopped phase of $\mathcal{X}_C(-x_4)/\mathbb{Z}_2$. In this case, $C_{2S}$ and $C_{4S}$ are collapsed and the residual $\mathbb{Z}_2$ acts as $(z_1, z_3) \to (-z_1, -z_3)$.

For every value of $x_4$, we hence find the space $\mathcal{X}_{E_1}(-x_4) = \mathcal{X}_C(-x_4)/\mathbb{Z}_2$, such that the domain wall is hence described as

$$\mathcal{M}_{E_1} = \frac{\left\{|z_1|^2 - |z_2|^2 + |z_3|^2 - |z_4|^2 = -x_4 \mid x_4 \in \mathbb{R}\right\}}{U(1) \times \mathbb{Z}_2} = \frac{\mathcal{M}_C}{\mathbb{Z}_2}. \tag{52}$$

---

[9]An example of this computation in the case of the $E_3$ theory using the gdP$_3$ geometry is discussed in details in appendix A.

[10]When $|z_2|^2 + |z_4|^2 = 0$ such that $z_0 = 0$, this $U(1)$ does not act on anything and we can simply drop it.

For every fiber $\mathcal{X}_C(-x_4)$ with $x_4 \neq 0$, the fixed locus of the involution is the exceptional $\mathbb{P}^1$ of the resolved conifold $\mathcal{X}_C(-x_4)$. At $x_4 = 0$, it is the single point $z_1 = z_2 = z_3 = z_4 = 0$. This is shown in figure 1.

As $\mathcal{M}_C$ is equal to a cone over $\mathbb{P}^3$, we can also think of $\mathcal{M}_{E_1}$ as

$$\mathcal{M}_{E_1} = C(\mathbb{P}^3)/\mathbb{Z}_2 \,, \tag{53}$$

where the $\mathbb{Z}_2$ acts by inverting two of the four homogeneous coordinates. This implies that there are two $A_1$ singularities sitting in every cross section of the cone. At the tip of the cone, the two $A_1$ singularities meet.

**Domain wall theory.** The geometry described in (51) hence captures the domain wall theory of [3], which is

$$\boxed{2}\overset{S}{\underset{Q}{\rule{0pt}{0pt}\!\times\!}}\boxed{2} \tag{54}$$

The boxes indicate $\mathfrak{su}(2)$ symmetries, which from the perspective of the 4d $\mathcal{N} = 1$ domain wall theory are flavor symmetries, whereas from the 5d perspective they are still gauged. The line connecting them is a bifundamental chiral $Q$ in the $(\mathbf{2}, \mathbf{2})$ and the cross is a single chiral $S$ in the singlet representation $(\mathbf{1}, \mathbf{1})$ that is usually called the *flipping field*. The name was chosen since it couples to $Q$ with the superpotential

$$\mathcal{W} = S \det Q = S\, \epsilon_{ab}\epsilon^{ij}Q_i^a Q_j^b \,, \tag{55}$$

where $a, b$ and $i, j$ are the flavor indices of the two $\mathfrak{su}(2)$'s. Because of this interaction, the F-term equation of $S$ sets $\det Q = 0$, which is what we mean when we say that $S$ *flips* the operator $\det Q$. We stress here that from the 4d field theory perspective, the superpotential (55) is defined in the UV. Since there is no 4d gauge symmetry in the model, this interaction is actually irrelevant in the IR. Notice that the absence of the superpotential (55) at low energies also gives the emergence of a new $\mathfrak{u}(1)$ global symmetry that acts on the singlet $S$ alone, since this is now decoupled from $Q$.

Field theoretically, the presence of the flipping field $S$ is important for several reasons. First, as we will discuss in subsection 4.5 and as it was explained in [3], stacking two of the above domain walls should result in them annihilating each other giving back the original 5d theory. In field theory this is understood as a Seiberg duality and it is crucial to have the flipping field in order to eventually get just the 5d theory without any extra 4d chirals. Second, when adding $N_F = 8$ flavors to the 5d theory, which we will do in section 5, we get a similar 4d domain wall theory but with some extra chirals, and this has to correspond to the 6d rank 1 E-string theory compactified on a tube with flux for the flavor symmetry (which is implemented by the variable mass along $x_4$ [51]). This origin of the 4d domain wall theory requires it to have several properties, for example its anomalies should match those obtained by compactifying the 6d anomaly polynomial [78] on the tube with flux, and as shown in [4] this only happens if we add the flipping field. Finally, the domain wall theory was mainly justified in [3] by some supersymmetric partition function computations, where the flipping field is required.

To understand in more detail how this matter content arises, consider the construction from the Higgsing of

$$\begin{aligned} SU(4) &\to SU(2) \times SU(2) \times U(1)\,, \\ \mathbf{15} &\to (\mathbf{3}, \mathbf{1})_0 \oplus (\mathbf{1}, \mathbf{3})_0 \oplus (\mathbf{1}, \mathbf{1})_0 \oplus (\mathbf{2}, \mathbf{2})_2 \oplus (\mathbf{2}, \mathbf{2})_{-2}\,, \end{aligned} \tag{56}$$

where there is a singlet $(\mathbf{1}, \mathbf{1})_0$ (corresponding to the flipping field) and one bifundamental chiral $(\mathbf{2}, \mathbf{2})_2$ (modulo convention of the charge). Note that this non-trivial chiral index comes

from the fact that in a local Higgs bundle description this ALE-fibration is modeled by a Morse function $f(\mathbf{x}) = \sum_{i=1}^{3} x_i^2$, and thus $\phi = df$ has one chiral zero-mode. See [79–82] for a detailed discussion of this Higgs bundle picture, i.e. local ALE-fibration, for $G_2$-compactifications.

## 4.4 The domain wall geometry as unfolding of an $A_3$ singularity

The geometry $\mathcal{M}_{E_1}$ can again be realized as an unfolding of a hyper-Kähler quotient along the lines of [68] by starting with an $\mathcal{N} = 2$ GLSM with four hyper-multiplets $\Phi_i$ with charges

$$
\begin{array}{cccc|c}
\Phi_1 & \Phi_2 & \Phi_3 & \Phi_4 & \\
\hline
1 & -1 & 0 & 0 & U(1)_1 \\
0 & 1 & -1 & 0 & U(1)_2 \\
0 & 0 & 1 & -1 & U(1)_3
\end{array}
\tag{57}
$$

Here, the unfolding is done by only setting the D-term triplet of $U(1)_2$ equal to three real FI parameters $(t_1, t_2, t_3)$ while keeping the others equal to zero. Following [68], this results in two $A_1$ singularities sitting at $z_1 = z_3 = 0$ and $z_2 = z_4 = 0$ in every $\mathbb{P}^3$ cross section of $\mathcal{M}_{E_1} = C(\mathbb{P}^3)/\mathbb{Z}_2$, recovering the description found above.

The above also allows us to make contact with a description in terms of D6-branes in IIA. From this perspective, this setup is a deformation of a stack of four D6-branes which have been rotated in pairs such that they intersect along $\mathbb{R}^{1,3}$. Again, we conclude that the massless spectrum consists of two $\mathfrak{su}(2)$'s and a bifundamental chiral multiplet.

Reducing M-theory on $\mathcal{X}_{E1}(t_1, 0)$ to IIA results in two D6-branes wrapped on the curves $C_{24}$ or $C_{13}$, depending on the sign of $t_1$. In the IIA reduction of the domain wall geometry we hence find stacks of D6-branes on two real cones over $S^2$. This again shows that we have two stacks of D6-branes on $\mathbb{R}^{1,3} \times \mathbb{R}^3$ touching along $\mathbb{R}^{1,3}$.

## 4.5 Stacking domain walls

Gluing two copies of $\mathcal{M}_{E_1}$ can now be achieved analogously to our treatment of gluing copies of $\mathcal{M}_C$ as the extra $\mathbb{Z}_2$ acts fiberwise on $\mathcal{X}_C(t)$ only. This results in the geometry

$$
\mathcal{M}_{2E_1}(a) = \frac{\left\{ |z_1|^2 - |z_2|^2 + |z_3|^2 - |z_4|^2 = x_4^2 - a \mid x_4 \in \mathbb{R} \right\}}{U(1) \times \mathbb{Z}_2} = \mathcal{M}_{2C}/\mathbb{Z}_2 \,.
\tag{58}
$$

For $a > 0$, there are two domain walls located at $x_4 = \pm\sqrt{a}$.

For $x_4 > \sqrt{a}$ and $x_4 < -\sqrt{a}$ there is a $A_1$ sitting over the $\mathbb{P}^1$ with homogeneous coordinates $[z_1 : z_3]$, i.e. at $z_2 = z_4 = 0$. These give rise to an $\mathfrak{su}(2) \oplus \mathfrak{su}(2)$ flavor symmetry, as the intervals in $x_4$ they sit over are non-compact. For the finite interval $|x_4| < \sqrt{a}$ there is a third $A_1$ singularity sitting over the $\mathbb{P}^1$ with coordinates $[z_2 : z_4]$, i.e. at $z_1 = z_3 = 0$. This $\mathbb{P}^1$ collapses over the points $x_4 = \pm\sqrt{a}$ forming a compact three-cycle $\Sigma$ as before. This gives rise to a gauge $\mathfrak{su}(2)$ located at $\Sigma$. At the two points $x_4 = \pm\sqrt{a}$ this gauge $\mathfrak{su}(2)$ intersects the two flavor $\mathfrak{su}(2)$'s, giving rise to two bifundamental chiral multiplets. This is similar to figure 5.

The field theory associated to two consecutive domain walls described geometrically by $\mathcal{M}_{2E_1}(a)$ for $a > 0$ is then captured by

$$
\boxed{2} \quad \times \quad \bigcirc\!\!\!\!2 \quad \times \quad \boxed{2}
\tag{59}
$$

where now the circle indicates a 4d $\mathcal{N} = 1$ $\mathfrak{su}(2)$ gauge symmetry, which arises by putting the 5d gauge symmetry on the interval between the two walls. This 4d $\mathfrak{su}(2)$ gauge group has overall 4 chirals in its fundamental representation, which is a theory with a quantum deformed moduli space of vacua [83]. This means that at the quantum level the mesonic

operator constructed with the chiral on the left multiplied by the chiral on the right, which is in the bifundamental representation of the $\mathfrak{su}(2) \oplus \mathfrak{su}(2)$ flavor symmetry, acquires a non-vanishing vev that Higgses the $\mathfrak{su}(2) \oplus \mathfrak{su}(2)$ down to its diagonal $\mathfrak{su}(2)$ subgroup. In other words, the theory in (59) is Seiberg dual to the 5d $SU(2)_0$ pure gauge theory on a line:

$$\boxed{2}\!-\!\!\times\!\!-\!\bigcirc\!\!2\!\!-\!\!\times\!\!-\!\boxed{2} \qquad \overset{\text{Seiberg}}{\longleftrightarrow} \qquad \boxed{2} \tag{60}$$

The two flipping fields are removed in the dualization.

This can also be seen by colliding the two domain wall geometric realizations, i.e. letting $a$ become negative. This implies that $x_4^2 - a$ is strictly positive so that redefining $z_i = z_i' \sqrt{x_4^2 - a}$ we can write

$$\left.\mathcal{M}_{2E_1}(a)\right|_{a<0} = \frac{\left\{ |z_1'|^2 - |z_2'|^2 + |z_3'|^2 - |z_4'|^2 = 1 \ \middle|\ x_4 \in \mathbb{R} \right\}}{U(1) \times \mathbb{Z}_2} \,. \tag{61}$$

This shows that topologically

$$\left.\mathcal{M}_{2E_1}(a)\right|_{a<0} \cong \mathcal{X}_{E_1} \times \mathbb{R}_{x_4} \,. \tag{62}$$

Equivalently, we may observe that there are no more domain walls left in this case. There is only a single $A_1$ singularity sitting over the $\mathbb{P}^1$ at $z_2' = z_4' = 0$ for every $x_4$, giving rise to a single $\mathfrak{su}(2)$ flavor symmetry. We then see that the transition in the space of the parameter $a$ is the geometric version of the Seiberg duality. The geometry for $a > 0$ describes the field theory on the left of (60), while $a < 0$ corresponds to the theory on right of (60). Because of this, we can interpret the parameter $a$ as related to the energy scale. The limit $a \to 0$ corresponds to the low energy limit where the two field theories become equivalent.

As before, this stacking of domain walls can be extended to $n$ copies by setting

$$\mathcal{M}_{nE_1}(P_n) = \left\{ \mathcal{X}_{E_1}\!\left( P_n(x_4), 0 \right) \mid x_4 \in \mathbb{R} \right\}, \tag{63}$$

with $P_n(x_4) = \prod_{i=1}^{n}(x_4 - a_i)$. We now find $n-1$ $\mathfrak{su}(2)$ gauge factors, an $\mathfrak{su}(2)^2$ flavor symmetry and $n$ charged bifundamental chirals. By continuously changing the polynomial $P_n(x_4)$, we can make pairs of adjacent domain walls collide and annihilate each other, which is again understood as Seiberg duality in field theory, and as a transition in the geometry. If $n$ is even we are eventually left with no domain walls, meaning that we just have the 5d $SU(2)_0$ gauge theory with the mass deformation $h_0$ over the line that is the geometry $\mathcal{X}_{E_1} \times \mathbb{R}$. Instead, if $n$ is odd we are left with a single domain wall, which corresponds to the geometry $\mathcal{M}_{E_1}$ and the associated field theory in (54).

$$\boxed{2}\!-\!\!\times\!\!-\!\underbrace{\bigcirc\!\!2\!-\!\!\times\!\!-\cdots-\!\!\times\!\!-\!\bigcirc\!\!2}_{n-1}\!-\!\!\times\!\!-\!\boxed{2} \quad \overset{(\text{Seiberg})^{n-1}}{\longleftrightarrow} \quad \begin{cases} \boxed{2} & n \text{ even,} \\[4pt] \boxed{2}\!-\!\!\times\!\!-\!\boxed{2} & n \text{ odd.} \end{cases} \tag{64}$$

## 4.6 Equivalence of the $\mathbb{F}_0$ and $\mathbb{F}_2$ constructions

The 5d rank 1 $E_1$ SCFT has an equivalent realization as a collapsed $\mathbb{F}_2$, instead of $\mathbb{F}_0 = \mathbb{P}^1 \times \mathbb{P}^1$. Here, the gauge theory phase is described by the non-compact Calabi-Yau threefold

$$X_{\mathbb{F}_2} = K_{\mathbb{F}_2} = C(Y^{2,2}) \,. \tag{65}$$

The toric polygon is shown in figure 7.

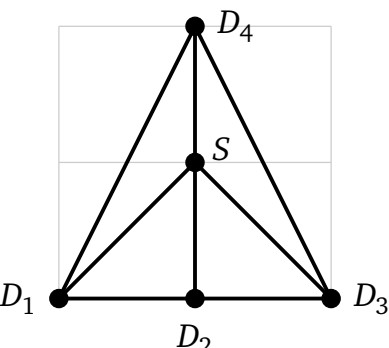

Figure 7: Fully resolved toric polygon for $\mathbb{F}_2$, which realizes the $SU(2)_0$ theory. $D_i$ denotes non-compact divisors and $S$ is the compact divisor, given by $\mathbb{F}_2$. We have also shown the underlying lattice for clarity.

The GLSM that describes this geometry is

$$
\begin{array}{c|ccccc|c}
 & D_1 & D_2 & D_3 & D_4 & S & \text{FI} \\
\hline
\mathcal{C}_{2S} & 1 & -2 & 1 & 0 & 0 & t_1 \\
\mathcal{C}_{1S} & 0 & 1 & 0 & 1 & -2 & t_2
\end{array}
\tag{66}
$$

and the associated D-term equations are

$$
\begin{aligned}
|z_1|^2 - 2|z_2|^2 + |z_3|^2 &= t_1 , \\
|z_2|^2 + |z_4|^2 - 2|z_0|^2 &= t_2 ,
\end{aligned}
\tag{67}
$$

where $z_i$ are the coordinates associated with the $D_i$ and $z_0$ is associated with $S$.

The crucial difference between these two presentations is that the $SO(3)$ flavor symmetry group (see [75, 77] for proof of this global form of the flavor symmetry group) of the $E_1$ theory is manifest in the $\mathbb{F}_2$ geometry, namely there is a curve $C_{2S}$, which is a $(-2, 0)$ curve that corresponds to the flavor W-bosons. At the level of the prepotential, this implies that it is better to use the one of [73] which is invariant under the Weyl group of $SO(3)$ and hence is valid on the entire ECB, so for this reason it is also called the *complete prepotential*, instead of the IMS one

$$
\mathcal{F}_{\text{complete}} = -\frac{|h_0|^2}{4}\varphi + \frac{4}{3}\varphi^3 ,
\tag{68}
$$

which is obtained from (38) by replacing $\varphi = \phi + \frac{|h_0|}{4}$. The Kähler form is[11]

$$
J = |h_0|D_1 - \phi S ,
\tag{69}
$$

and the volumes of curves are

$$
\begin{aligned}
\text{Vol}(C_{3S}) &= \text{Vol}(C_{1S}) = 2\phi , \\
\text{Vol}(C_{2S}) &= |h_0| , \\
\text{Vol}(C_{4S}) &= 4\phi + |h_0| .
\end{aligned}
\tag{70}
$$

---

[11] The mapping of the Kähler parameters appearing in $J$ and the field theory mass parameters can be obtained by comparing the field theory prepotential (68) with the geometric one

$$
\mathcal{F}_{\text{geom}} = -\frac{1}{6}J^3 .
$$

See appendix A for the details of a similar computation in the case of the $E_3$ theory using the gdP$_3$ geometry.

Hence we identify

$$t_1 = |h_0|, \qquad t_2 = \phi. \tag{71}$$

The manifest flavor symmetry comes at the price that both $h_0$ and $-h_0$ correspond to identical points in the Kähler cone. Dropping the modulus in (68), we cannot have $h_0 < 0$ as this implies that the curves $C_{2S}$ and $C_{4S}$ have negative volume. This is no reason for concern physically, as $\mathbb{F}_0$ and $\mathbb{F}_2$ are isomorphic as real manifolds and only differ in their complex structure [84]. Setting $\phi = 0$ to construct the domain wall geometries for both geometries, we can also see their equivalence as follows. In both cases we can write the non-compact Calabi-Yau as the total space of a sum of line bundles over $\mathbb{P}^1$ modulo $\mathbb{Z}_2$

$$K_{\mathbb{F}_0}|_{\phi=0} = \mathcal{O}(-1) \oplus \mathcal{O}(-1)/\mathbb{Z}_2, \qquad K_{\mathbb{F}_2}|_{\phi=0} = \mathcal{O}(-2) \oplus \mathcal{O}(0)/\mathbb{Z}_2. \tag{72}$$

In both cases, the $\mathbb{Z}_2$ acts on the fibers by inverting all coordinates and leaving only the zero section of the bundle fixed. As real spaces, both of the above are real vector bundles of rank 4 over $S^2$. Such bundles are classified by $\pi_1(GL(4, \mathbb{R})) = \mathbb{Z}_2$. There are hence only two such bundles, one of which is the trivial bundle. Both of the bundles above are clearly non-trivial as they are sums of non-trivial vector bundles, which then implies they must be isomorphic as real vector bundles. Finally, the quotient only acts on the fibers in an identical way, such that as real manifolds

$$K_{\mathbb{F}_0}|_{\phi=0} \cong K_{\mathbb{F}_2}|_{\phi=0}. \tag{73}$$

We can use the identification between ECB parameters to associate points in the Kähler moduli space of $K_{\mathbb{F}_0}$ with points in the Kähler moduli space of $K_{\mathbb{F}_2}$, and doing so we can map the domain wall constructed using $\mathbb{F}_0$ to a domain wall obtained from $K_{\mathbb{F}_2}$. The Kähler cone of $\mathcal{K}_{\mathbb{F}_0}$ and the path we are interest in for the construction of a domain wall are

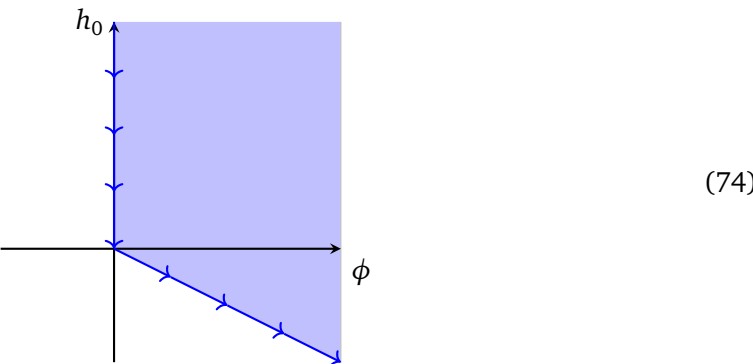

$$\tag{74}$$

Note that the Kähler cone $\mathcal{K}_{\mathbb{F}_0}$ has the symmetry $2\phi \leftrightarrow 2\phi + h_0$ which corresponds to swapping the two rulings on $\mathbb{F}_0$. A different choice of ruling corresponds to a different gauge theory interpretation. For the $\mathbb{F}_2$ description, the only change is that the identification of $(h_0, \phi)$ with the Kähler parameters changes slightly according to the above rules, but the path remains the same.

We then find the following equivalent characterization of the domain wall geometry $\mathcal{M}_{E_1}$ associated to the $E_1$ theory

$$\mathcal{M}_{E_1} = \frac{\left\{ |z_1|^2 - 2|z_2|^2 + |z_3|^2 = |x_4|, \, z_4 \in \mathbb{C} \mid x_4 \in \mathbb{R} \right\}}{U(1) \times \mathbb{Z}_2}, \tag{75}$$

where the $U(1)$ acts as in the first row of (66), while the $\mathbb{Z}_2$ acts as $(z_2, z_4) \to (-z_2, -z_4)$.

# 5   $G_2$-geometries for rank 1 Seiberg theories

In this section we extend the analysis of the previous section by adding flavors, that is we consider the domain walls in the rank 1 $E_{N_F+1}$ Seiberg theories and the 5d KK-theory coming from the circle reduction of the 6d rank 1 E-string theory. These admit an $SU(2) + N_F \boldsymbol{F}$ gauge theory description with $N_F \leq 8$, whose domain walls were studied field theoretically in [3, 4]. As before, we will discuss the corresponding $G_2$ geometries derived from these constructions.[12]

## 5.1   The field-theoretic description

The 5d rank 1 $E_{N_F+1}$ SCFTs and also the marginal rank 1 theory coming from the circle reduction of the 6d rank 1 E-string have a mass deformation to an $SU(2)$ gauge theory with $N_F \leq 8$ flavors. The IMS prepotential for these theories is

$$\mathcal{F}_{SU(2),N_F} = h_0 \phi^2 + \frac{4}{3}\phi^3 - \frac{1}{12}\sum_{i=1}^{N_F}\sum_{\pm} |\pm\phi + m_i|^3, \tag{76}$$

where as before $h_0 = \frac{1}{2g_0^2}$ is related to the instanton mass, $g_0$ is the bare gauge coupling, and $m_i$ are the masses for the $\mathfrak{so}(2N_F)$ flavor symmetry. Together the parameters $\{\phi, h_0, m_i\}$ parameterize the ECB.

One can choose the masses to be generally positive or negative, but for the sake of making things less convoluted we will choose $\pm\phi + m_i > 0$. The mass parameter $m_\lambda$ from subsection 3.2 is given by

$$\frac{1}{2g_{\text{eff}}^2} = m_\lambda = h_0 - \frac{1}{2}\sum_{i=1}^{N_F} m_i, \tag{77}$$

where we took the Coulomb parameter to vanish $\phi = 0$ so $m_i > 0$, and we require $h_0 > \frac{1}{2}\sum_{i=1}^{N_F} m_i$ such that $m_\lambda$ is positive and the gauge description makes sense. One can see that this prepotential is not invariant under the full Weyl group of the UV $E_{N_F+1}$ symmetry.[13] One can now generate a domain wall by using a $\mathbb{Z}_2$ transformation relating the parameters on the two sides as

$$m_\lambda^{(L)} \to m_\lambda^{(R)} = -m_\lambda^{(L)}, \qquad m_i^{(L)} \to m_i^{(R)} = m_i^{(L)} + \frac{m_\lambda^{(L)}}{2}, \qquad i = 1, \cdots, N_F, \tag{78}$$

where the left side $(L)$ is the side of the domain wall where we fixed our parameters above and where the prepotential correctly describes the $SU(2)$ gauge theory. Note that the $\mathbb{Z}_2$ transformation above is a part of the Weyl group of the UV theory. In addition, as we mentioned in section 3, $m_i^{(R)}$ are not the physical masses of hyper-multiplets on the right side but only indicate the parameter transformation that will allow to identify the prepotential on both sides. The physical masses of the hyper-multiplets are

$$m_{i,\text{phys}}^{(I)} = m_i^{(I)} - \phi^{(L)}, \qquad I = L, R, \tag{79}$$

and we will see below that $\phi^{(L)}$ is shifted on the right side.

As we already mentioned, the IMS prepotential is not invariant under such Weyl transformations. Specifically, the IMS prepotential is valid only in a specific Weyl sub-chamber of the ECB, while the two theories involved in the construction of the domain wall live in different

---

[12]The field theories for the higher rank versions of these domain walls, specifically for the 5d KK-theory coming from the higher rank E-string and the 5d SCFTs obtained from it after decoupling flavors, have appeared in [41,85].

[13]For $N_F = 8$ the UV symmetry is $E_8 \times U(1)_{\text{KK}}$ where the $U(1)_{\text{KK}}$ is associated with the circle of the 6d theory reduction.

sub-chambers. For this reason, if we apply (78) to (76) we would apparently end up with a prepotential that does not suitably describe an $SU(2)$ gauge theory anymore, but this is just an artifact of trying to describe the theory in a new Weyl sub-chamber on the right side while still using a CB $\phi$ that was only valid in the original Weyl sub-chamber on the left. The solution is to just move to a different Coulomb parameter on the other side of the wall

$$\phi^{(L)} \to \phi^{(R)} = \phi^{(L)} - \frac{m_\lambda^{(L)}}{2} \,. \tag{80}$$

With this further identification we will find the same prepotential on the right side describing the $SU(2) + N_F \boldsymbol{F}$ gauge theory. Nevertheless, we will describe both sides in terms of the same parameter $\phi^{(L)}$. This will prove to be beneficial later, when we construct the geometry in terms of continuously varying parameters. The price to pay is that this leads to a seemingly strange behaviour when moving to the right side. Specifically, while on the left we need to set $\phi^{(L)} = 0$ to have the $SU(2)$ gauge theory, on the right the proper parameter that we should set to zero to still have a sensible $SU(2)$ gauge theory is $\phi^{(R)} = 0$ which in terms of $\phi^{(L)}$ means $\phi^{(L)} = \frac{1}{2} m_\lambda^{(L)}$. Accordingly, then the physical masses are as follows:

$$m_{i,\text{phys}}^{(L)}(x_4 < 0) = m_i^{(L)}(x_4)\,, \qquad m_{i,\text{phys}}^{(R)}(x_4 > 0) = m_i^{(L)}(-x_4)\,. \tag{81}$$

Hence, if we want to describe both sides with the same CB parameter we can do it in a way that one side is manifestly $SU(2) + N_F \boldsymbol{F}$ using the IMS convention for the prepotential, while the other side would not manifest the $SU(2) + N_F \boldsymbol{F}$ gauge theory. Specifically we will get a non-vanishing $\phi$ parameter.

Going back to the domain wall construction of [3], one can see that it preserves an $\mathfrak{su}(N_F)$ flavor symmetry for the rank 1 theories. In order to preserve this symmetry we need to require that all the hyper-multiplet masses are equal; thus, we will set

$$m_i^{(I)} = m^{(I)} > 0\,, \qquad I = L, R\,, \tag{82}$$

for all $i = 1, \ldots, N_F$. Since all the masses transform in the same way in (78), $m^{(L)}$ will transform uniformly to a single $m^{(R)}$. Re-expressing the transformation in terms of $m^{(I)}$ we find

$$m_\lambda^{(L)} \to m_\lambda^{(R)} = -m_\lambda^{(L)}\,, \qquad m^{(L)} \to m^{(R)} = m^{(L)} + \frac{m_\lambda^{(L)}}{2}\,, \tag{83}$$

with

$$m_\lambda^{(I)} = h_0^{(I)} - \frac{N_F}{2} m^{(I)}\,, \qquad I = L, R\,. \tag{84}$$

We already pointed out that $m_\lambda$ is the mass parameter that changes its sign across the domain wall. It will be useful to define an orthogonal combination of $h_0$ and $m$ that remains constant across the domain wall as

$$m_c^{(I)} = h_0^{(I)} - \frac{N_F - 8}{2} m^{(I)} \quad \Rightarrow \quad m_c^{(L)} \to m_c^{(R)} = m_c^{(L)} \equiv m_c\,. \tag{85}$$

This means that $m_c$ should be taken to have a constant profile. We can also invert the previous relations to write both the hyper-multiplet masses $m^{(I)}$ and $h_0^{(I)}$ in terms of $m_\lambda^{(I)}$ and $m_c$ as

$$m^{(I)} = \frac{m_c - m_\lambda^{(I)}}{4}\,, \qquad h_0^{(I)} = \frac{N_F}{8} m_c + \frac{8 - N_F}{8} m_\lambda^{(I)}\,. \tag{86}$$

The domain wall will be located on the interface where

$$m^{(I)} = \frac{m_c}{4}\,, \tag{87}$$

setting $m_\lambda^{(I)} = 0$. We have summarized the notations and relations among the parameters in table 1.

Table 1: The notations and relations of the right parameters ($x_4 > 0$) and the physical parameter for any $x_4$ in terms of the parameters given on the left side denoted by the superscript ($L$).

| Param | Right side parameter ($x_4 > 0$) | Physical parameter |
|---|---|---|
| $h_0$ | $h_0^{(R)} = h_0^{(L)}(-x_4) - \frac{8-N_F}{4} m_\lambda^{(L)}(-x_4)$ | $h_0^{\text{phys}}(x_4) = \frac{1}{2g_0^2} = \begin{cases} h_0^{(L)}(x_4), & x_4 \le 0, \\ h_0^{(L)}(-x_4), & x_4 > 0 \end{cases}$ |
| $m$ | $m^{(R)}(x_4) = m^{(L)}(-x_4) + \frac{m_\lambda^{(L)}(-x_4)}{2}$ | $m_{\text{phys}}(x_4) = \begin{cases} m^{(L)}(x_4), & x_4 \le 0, \\ m^{(L)}(-x_4), & x_4 > 0 \end{cases}$ |
| $\phi$ | $\phi^{(R)}(x_4) = \phi^{(L)}(-x_4) - \frac{m_\lambda^{(L)}(-x_4)}{2}$ | $\phi_{\text{phys}}(x_4) = 0$ |
| $m_\lambda$ | $m_\lambda^{(R)}(x_4) = -m_\lambda^{(L)}(-x_4)$ | $m_\lambda^{\text{phys}}(x_4) = \frac{1}{2g_{\text{eff}}^2} = \begin{cases} m_\lambda^{(L)}(x_4), & x_4 \le 0, \\ m_\lambda^{(L)}(-x_4), & x_4 > 0 \end{cases}$ |
| $m_c$ | $m_c^{(R)}(x_4) = m_c^{(L)}(-x_4)$ | No physical meaning |

## 5.2 The geometry of 5d rank 1 SCFTs

Let us now move to the discussion of the geometry of the $SU(2) + N_F \mathbf{F}$ theories. As is well-known, the rank 1 SCFTs are realized by the complex cone over generalized del Pezzo surfaces $\text{gdP}_{N_F+1}$, which do not all have a toric description. Note that we will require the generalized del Pezzo description (see e.g. [12] for detailed analysis of these geometries), as this manifests the flavor symmetry.

The absence of a toric description of the SCFT is however not detrimental: we will only require a description of the theories in a particular subspace of the extended Coulomb branch, where a toric description is in fact available. The toric polygon, with the triangulation required is shown in figure 8. We cannot remove the curve $D_1 \cdot D_4$, as the resulting singular polygon would not be convex. However in the extended Coulomb branch phase, where this curve (and associated M2-brane wrapped state) has finite mass, this is a perfectly fine toric geometry (the union of two convex polygons).

The non-compact divisors $D_i$ for $i \ge 5$ model the addition of flavors, and in figure 8 we show the case $N_F = 8$. The models with less flavors are obtained by lowering the connection from $D_4 - D_{12}$ to $D_4 - D_{N_F+4}$ in the toric polygon of figure 8.

One can in a similar manner to the $N_F = 0$ case of the previous section describe the geometry in terms of an $\mathcal{N} = 1$ GLSM. Since this analysis is very similar and does not give any added value we will not repeat it here, and only refer to appendix A, where we write the full analysis for the case of the rank 1 $E_3$ SCFT. In the general case we find the following curve volumes:

$$
\begin{aligned}
\text{Vol}(C_{1S}) &= \text{Vol}(C_{3S}) = 2\phi, \\
\text{Vol}(C_{2S}) &= \text{Vol}(C_{4S}) = 2\phi + m_\lambda, \\
\text{Vol}(C_{14}) &= m_1 - \phi, \\
\text{Vol}(C_{(4+i)(5+i)}) &= m_{i+1} - m_i, \qquad i = 1,..,N_F - 1,
\end{aligned}
\tag{88}
$$

$$
\begin{aligned}
\text{Vol}(C_{1S}) &= \text{Vol}(C_{3S}) = 2\phi, \\
\text{Vol}(C_{2S}) &= \text{Vol}(C_{4S}) = 2\phi + m_\lambda, \\
\text{Vol}(C_{14}) &= m_1 - \phi, \\
\text{Vol}(C_{4(4+i)}) &= m_{i+1} - m_i, \qquad i = 1,..,N_F - 1.
\end{aligned}
\tag{89}
$$

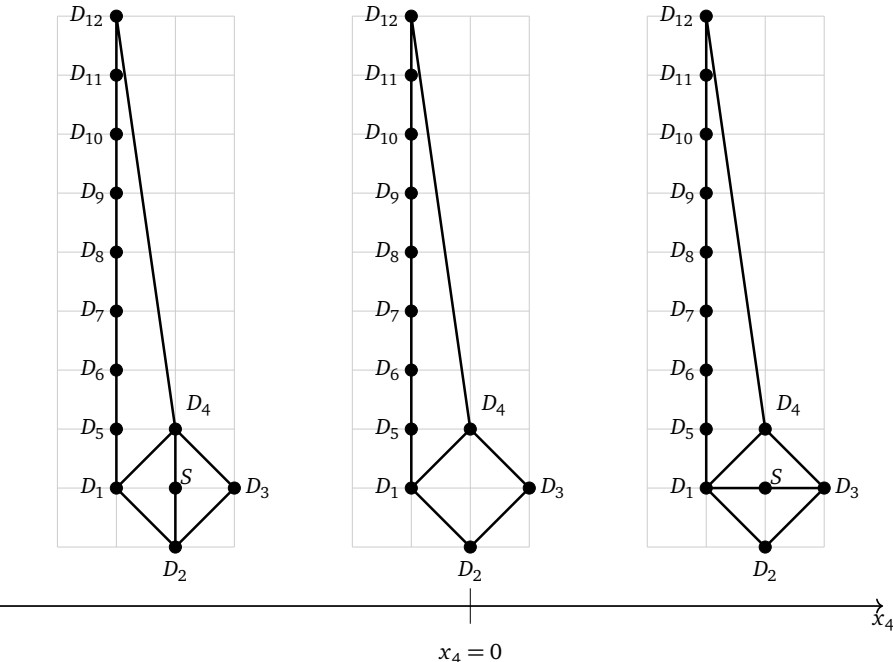

$x_4 = 0$

Figure 8: Toric polygon for the partially resolved geometry that realizes the $SU(2) + N_F F$ theory. Again $S$ is the compact divisor, which is now a generalized del Pezzo, and for $N_F$ flavors, the line between $D_4$ and $D_{N_F+4}$ is included. The figure shows the $N_F = 8$ case. Here we show already the three phases relevant for the domain wall solution: on the left and right hand-sides the partial resolutions, and at $x_4 = 0$ the singular geometry. Note that the curve $C_{14} = D_1 \cdot D_4$ cannot be collapsed while retaining the toric description.

One can see from the last line that setting $m_i = m$ for all $i = 1, \ldots, N_F$ explicitly shows the existence of a global $\mathfrak{su}(N_F)$ symmetry. In the domain wall construction $\mathrm{Vol}(C_{14}) = m - \phi = m_{\mathrm{phys}} > 0$ for $x_4$ in the proximity of the domain wall, with $m_{\mathrm{phys}}$ being the physical mass of the hyper-multiplet; thus, the toric description is valid. In addition, we see as in the $E_1$ case that on the left $\phi = 0$ and $m_\lambda > 0$ blowing down the curves $C_{1S}$ and $C_{3S}$, while on the right hand side $\phi = -m_\lambda/2$ and $m_\lambda < 0$ blowing down $C_{2S}$ and $C_{4S}$.

## 5.3 The $G_2$ geometry of domain walls

The geometry for general $N_F > 0$ can be described in terms of an $\mathcal{N} = 1$ GLSM with $N_F + 2$ $U(1)$'s and the charge assignments

|          | $D_1$ | $D_2$ | $D_3$ | $D_4$ | $D_{n+3}$       | $D_{n+4}$      | $D_{n+5}$ | $S$  | FI                    |
|----------|-------|-------|-------|-------|-----------------|----------------|-----------|------|-----------------------|
| $C_{2S}$ | 1     | 0     | 1     | 0     | 0               | 0              | 0         | $-2$ | $t_1 = m_\lambda + 2\phi$ |
| $C_{1S}$ | 0     | 1     | 0     | 1     | 0               | 0              | 0         | $-2$ | $t_2 = 2\phi$         |
| $C_{14}$ | $-1$  | 0     | 0     | $-1$  | $\delta_{n,2}$  | 0              | 0         | 1    | $t_3 = m - \phi$      |
| $C_{45}$ | 1     | 0     | 0     | 0     | $-2\delta_{n,2}$ | $\delta_{n,2}$ | 0         | 1    | $t_4 = 0$             |
| $C_{4(n+4)}$ | 0 | 0     | 0     | 0     | 1               | $-2$           | 1         | 0    | $t_{n+3} = 0$         |

(90)

where $n = 2, \cdots, N_F - 1$ and we have already set the FI's to the values we are interested in for constructing the domain wall, as in (88) and figure 8.

The D-term vacuum equations are

$$|z_1|^2 - |z_2|^2 + |z_3|^2 - |z_4|^2 = t_1 - t_2 = m_\lambda \,,$$
$$|z_2|^2 + |z_4|^2 = 2|z_0|^2 + 2\phi \,,$$
$$-|z_1|^2 - |z_4|^2 + |z_5|^2 + |z_0|^2 = t_3 = m - \phi = \frac{m_c - m_\lambda}{4} - \phi \,, \tag{91}$$
$$|z_1|^2 + |z_6|^2 = 2|z_5|^2 \,,$$
$$|z_{n+3}|^2 + |z_{n+5}|^2 = 2|z_{n+4}|^2 \,, \qquad n = 2, \ldots, N_F - 1 \,,$$

where for convenience we replaced the equation for $C_{2S}$ with the combination $C_{2S} - C_{1S}$ giving the first equations and we introduced $m_c = m_\lambda + 4m$ which is the parameter that is trivially identified across the domain wall.

One can solve all of the above equations except for the first and the third one in terms of $|z_0|$ and $|z_{n+4}|$ for $n = 1, \ldots, N_F - 1$ and use the corresponding $U(1)$'s to also gauge fix their arguments, up to an overall $\mathbb{Z}_{N_F}$ that acts as

$$(z_1, z_{N_F+4}) \to (\omega z_1, \omega^{N_F-1} z_{N_F+4}) \,, \qquad \omega^{N_F} = 1 \,, \tag{92}$$

which is obtained by taking the combination of $U(1)$'s appearing in the GLSM given by

$$\mathbb{Z}_{N_F} : \quad \sum_{n=1}^{N_F-1} n \, C_{4(n+4)} \,. \tag{93}$$

Now, we can find the domain wall geometry by fibering the GLSM vacuum equations over $x_4 \in \mathbb{R}$ taking

$$m_\lambda = -x_4 \,, \qquad m = \frac{m_c + x_4}{4} \,, \qquad \phi = \frac{x_4 + |x_4|}{4} \,, \tag{94}$$

with $m_c$ being a real constant. We point out that using this parameterization the volume of the curve $C_{14}$ is

$$t_3 = \text{Vol}\big(C_{14}\big) = m - \phi = \frac{m_c - |m_\lambda|}{4} = \frac{m_c - |x_4|}{4} \,, \tag{95}$$

which corresponds to the physical mass $m_{\text{phys}}$ of the hyper-multiplets.

Explicitly, this can be written as

$$\mathcal{M}_{E_{N_F+1}}(m_c) = \frac{\left\{ \begin{array}{ll} |z_1|^2 - |z_2|^2 + |z_3|^2 - |z_4|^2 = -x_4 & | \quad x_4 \in \mathbb{R} \\ \left(1 - \dfrac{4}{N_F}\right)|z_1|^2 + |z_2|^2 + |z_3|^2 - 3|z_4|^2 + \dfrac{4}{N_F}|z_{N_F+4}|^2 = m_c \end{array} \right\}}{U(1)^2 \times \mathbb{Z}_2 \times \mathbb{Z}_{N_F}} \,. \tag{96}$$

For odd $N_F$ the action of $\mathbb{Z}_{N_F}$ is already contained in the $U(1)$ action, and for even $N_F = 2k$, not a multiple of 4, a $\mathbb{Z}_k$ is already contained in the $U(1)$.

According to [3], the 4d $\mathcal{N} = 1$ field theory living on the domain wall that we just realized geometrically is

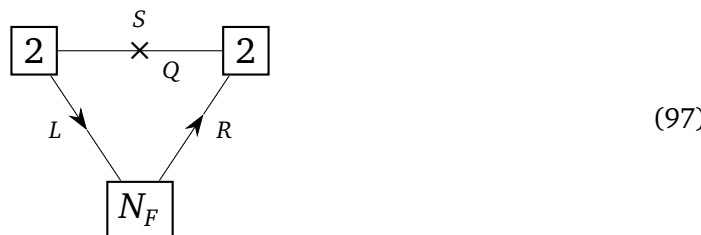

$$\tag{97}$$

As in the $E_1$ example, the two boxes on the sides represent $\mathfrak{su}(2)$ symmetries that are flavor from the 4d perspective, while they are gauged from the 5d one. The bottom box is instead an $\mathfrak{su}(N_F)$ symmetry that is flavor from both the 4d and the 5d points of view. The matter content consists of the $\mathfrak{su}(2) \oplus \mathfrak{su}(2)$ bifundamental chiral $Q$ and the flipping field $S$ as in the $E_1$ case, but now we also have two extra sets of chirals $L$ and $R$. $L$ is in the fundamental of the left $\mathfrak{su}(2)$ and the fundamental of $\mathfrak{su}(N_F)$, while $R$ is in the fundamental of the right $\mathfrak{su}(2)$ and the anti-fundamental of $\mathfrak{su}(N_F)$, where the fact that they are in complex conjugate representations of $\mathfrak{su}(N_F)$ is encoded in the different orientation of the arrows in the quiver. These chiral fields interact with the superpotential

$$\mathcal{W} = S \det Q + L Q R = S \, \epsilon_{ab} \epsilon^{ij} Q_i^a Q_j^b + L_a^\alpha Q_i^a R_\alpha^j \,, \tag{98}$$

where $\alpha$ is an $\mathfrak{su}(N_F)$ flavor index, while $a, b$ and $i, j$ are the indices of the two $\mathfrak{su}(2)$'s as before. Similarly to what we said for the $E_1$ theory, this superpotential is irrelevant in the IR since there is no 4d gauge symmetry.

The origin of the chirals $L$, $R$ is different from that of $Q$, $S$. The latter are genuine 4d fields that live only on the domain wall at $x_4 = 0$. The former are instead coming from certain components of the 5d hyper-multiplets in the bulk that are given suitable boundary conditions. More precisely, if we denote by $q_L, \tilde{q}_L$ the scalar components of the 5d hyper-multiplets living in the region $x_4 < 0$ and similarly by $q_R, \tilde{q}_R$ those for $x_4 > 0$, then we give $\frac{1}{2}$-BPS boundary conditions which are Dirichlet for, say, $\tilde{q}_L, q_R$ and Neumann for the others

$$\begin{aligned}
\tilde{q}_L|_{x_4=0} &= 0 \,, & \partial_4 q_L|_{x_4=0} &= 0 \,, \\
q_R|_{x_4=0} &= 0 \,, & \partial_4 \tilde{q}_R|_{x_4=0} &= 0 \,.
\end{aligned} \tag{99}$$

The components $q_L$, $\tilde{q}_R$ that are given Neumann boundary conditions are the only ones that remain dynamical on the domain wall and become the scalar components of the 4d chirals $L$, $R$ respectively, which transform in conjugate representations under the flavor symmetry. In the 5d bulk we also have a mass term for the hyper-multiplets, which schematically is

$$\mathcal{S} \supset \int_{x_4<0} m_{\text{phys}}^{(L)} q_L \tilde{q}_L + \int_{x_4>0} m_{\text{phys}}^{(R)} q_R \tilde{q}_R \,, \tag{100}$$

where $m_{\text{phys}}^{(I)} = \frac{m_c - |m_\lambda^{(I)}|}{4}$ are the physical masses of hyper-multiplets, which are equal to the $C_{14}$ curve volume, and are not the same as the parameters $m^{(I)}$. Due to the boundary conditions (99), the mass terms do not survive on the domain wall at $x_4 = 0$, so the 4d chirals $L$ and $R$ localized there are actually massless.

From the geometric perspective we can recover the field theory analysis as follows. As for the $E_1$ theory, there are two $A_1$ singularities located on copies of $\mathbb{R}^3$ in $\mathcal{M}_{E_{N_F+1}}$ that give rise to two $\mathfrak{su}(2)$ flavor symmetries, as well as a bifundamental $Q$ and a singlet $S$ at their intersection. On top of this, there is now also a further $A_{N_F-1}$ singularity for every $x_4$, encoding the $\mathfrak{su}(N_f)$ flavor symmetry, see figure 9.

This singularity is separated from the $\mathfrak{su}(2)$'s by the curve $C_{14}$, which has finite volume in the vicinity of the domain wall. From the 5d perspective, the volume of $C_{14}$ measures the mass of the hyper. At $x_4 = \pm m_c$, this curve collapses,[14] so that the loci of the $A_{N_F-1}$ singularity and one $A_1$ (for each side) intersect. Correspondingly, we find two chiral multiplets $L$ and $R$ charged under $\mathfrak{su}(N_F)$ and one of the two $\mathfrak{su}(2)$ each. Equivalently, there is the mass of a hyper-multiplet going from positive to negative at this point, so just as in the case of the free

---

[14]This cannot be realized within the Kähler cone of the simple toric variety used here, but only in a more complicated setup.

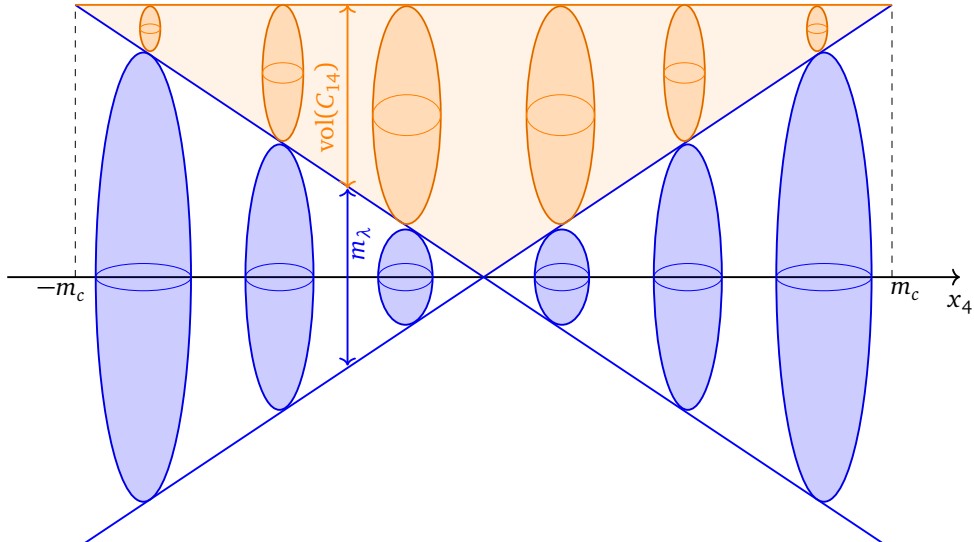

Figure 9: Depiction of the geometry of the domain wall theory for $SU(2) + N_F \boldsymbol{F}$. The 2-sphere $C_{14}$ is fibered, with the height of the orange segment corresponding to its volume. This sweeps out a three-cycle, (orange shaded triangle), which generates a superpotential coupling. This associative three-cycle connects three codimension 7 loci: two of them are the $L$ and $R$ chirals (at $x_4 = \pm m_c$), while the third is the bifundamental chiral $Q$ (at $x_4 = 0$). Transverse to this picture there are the $A_1$ singularity that is fibered over the blue spheres (whose radius sets $m_\lambda$) and above each of the $C_{14}$ (the orange spheres), there is an $A_{N_F - 1}$ singularity.

hyper-multiplet we should expect a chiral to be localized at the point of transition, and to have a codimension 7 singularity in our geometry. Note that this will make our prepotential analysis invalid as we initially considered all the masses to be positive.

Finally, the collapse of the curve $C_{14}$ at $x_4 = \pm m_c$ allows to form a compact three-cycle, which generates a superpotential between the fields $Q$, $L$ and $R$. Once we pass the singular points at $x_4 = \pm m_c$ we get a flop transition that cannot be described by the toric diagrams shown in figure 8. The new curves that emerge after the flop of $C_{14}$ on both sides will correspond to masses for hyper-multiplets in the two 5d theories on both sides of this construction. These masses profile will go to some constant values at $x_4 \to \pm\infty$. Likewise, the value of $m_\lambda$ will asymptote to a constant value in such a way that no additional codimension 7 singularities are formed. Far away from the domain wall the theory will be that of a 5d $SU(2) + N_F \boldsymbol{F}$ with massive hypers.

## 5.4 Stacked domain walls geometry

As was discussed in subsection 4.5 for the $E_1$ domain wall, we can stack two of the rank 1 domain walls above, which annihilate each other leaving just the 5d $SU(2) + N_F \boldsymbol{F}$ theory on a line. From the field theory perspective this is once again understood as a Seiberg duality

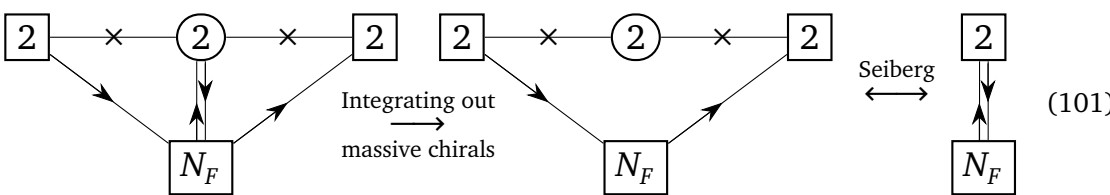 (101)

In the quiver on the left and the middle, as in the $E_1$ case, the $\mathfrak{su}(2)$'s at the two sides are flavor symmetries from the perspective of the 4d domain wall theory while being gauged in the 5d bulk. The $\mathfrak{su}(2)$ in the middle is a 4d gauge symmetry and the $\mathfrak{su}(N_F)$ on the bottom is a flavor symmetry both from the 4d and the 5d perspective. On the right quiver, we have two $\mathfrak{su}(2) \oplus \mathfrak{su}(N_F)$ chirals that recombine to form a hyper-multiplet.

We can understand this geometrically exactly in the same way as in the $E_1$ theory, since from the defining equations (96) of $\mathcal{M}_{E_{N_F}}(m_c)$ we see that the only equation with a Kähler parameter that has a non-trivial profile along $x_4$ is the first one which is the same as the one that appeared in $\mathcal{M}_{E_1}$ for the domain wall of the $E_1$ theory and in $\mathcal{M}_C$ for the free hyper, see eqs. (75) and (5), respectively.

We define the geometry of two stacked domain walls as

$$
\mathcal{M}_{2E_{N_F}}(m_c) = \frac{\left\{ \begin{array}{cc} |z_1|^2 - |z_2|^2 + |z_3|^2 - |z_4|^2 = x_4^2 - a & | \quad x_4 \in \mathbb{R} \\ \left(1 - \dfrac{4}{N_F}\right)|z_1|^2 + |z_2|^2 + |z_3|^2 - 3|z_4|^2 + \dfrac{4}{N_F}|z_{N_F+4}|^2 = m_c \end{array} \right\}}{U(1)^2 \times \mathbb{Z}_2 \times \mathbb{Z}_{N_F}} \, . \tag{102}
$$

As in the $E_1$ theory, for $a > 0$ there are three $A_1$ singularities, two sit on non-compact cycles for $|x_4| > \sqrt{a}$, and one sitting on a compact cycle for $|x_4| < \sqrt{a}$. The cross section of these cycles is a $\mathbb{P}^1$; thus, we get the $\mathfrak{su}(2)$'s on the two sides of the quiver, and the gauge $\mathfrak{su}(2)$ in the middle of the quiver on the left and the middle of (101). The picture is then similar to the one of figure 5 for $E_1$, but this is further decorated by the curve $C_{14}$ and the non-compact $A_{N_F-1}$ singularity in a similar fashion to figure 9.

- For $a > m_c$ we actually find two chirals with a superpotential coupling between them, transforming in the bifundamental of the middle $\mathfrak{su}(2)$ and the $\mathfrak{su}(N_F)$.

- For $0 < a < m_c$ the flop in $C_{14}$ does not happen before the second domain wall is reached so that the two massive chirals in the middle are absent (see middle figure of (101)). Letting $a$ becomes smaller corresponds to flowing to the IR, so that we can think about these two massive chirals as being integrated out.

- For $a < 0$ we have a transition to the trivial geometry $\mathcal{X}_{E_{N_F+1}} \times \mathbb{R}$ as we move further towards the IR.

This can again be generalized to the stacking of $n$ domain walls exactly as in the $E_1$ theory.

# 6 $G_2$-geometries for 5d SQCD domain walls

The domain wall construction we discussed in the previous sections can be applied to any 5d SCFT admitting a low energy gauge theory description where some UV $\mathbb{Z}_2$ Weyl transformation is not manifest. In this section we consider the higher rank example of the $SU(N)_N$ gauge theories, where the subscript is the Chern–Simons level, and subsequently also add fundamental flavor. These domain wall theories were discussed from the field theory point of view in [3].

## 6.1 The Coulomb branch description of $SU(N)_N$

We start from the IMS prepotential of the $SU(N)_N$ theory

$$\mathcal{F}_{\text{IMS}} = \frac{h_0}{2} \sum_{a,b=1}^{N-1} h^{ab} \phi_a \phi_b + \frac{N}{2} \sum_{a=1}^{N-1} \left( \phi_{a-1}^2 \phi_a - \phi_a \phi_{a+1}^2 \right)$$
$$+ \frac{4}{3} \sum_{a=1}^{N-1} \phi_a^3 + \frac{1}{2} \sum_{a=1}^{N-1} (N-2a) \left( \phi_{a-1}^2 \phi_a - \phi_a \phi_{a+1}^2 \right), \tag{103}$$

where $\phi_0 = \phi_N = 0$. The two terms in the first line are the classical contribution from the kinetic and from the Chern–Simons (CS) terms, respectively, while the second line is the one-loop contribution. We denote by $\phi_a$ with $a = 1, \cdots, N-1$ the CB vevs in the Cartan of the $SU(N)$ gauge group. We choose a parameterization such that the Cartan matrix $h^{ab}$ is

$$h^{ab} = 2\delta^{a,b} - \delta^{a,b+1} - \delta^{a+1,b}. \tag{104}$$

We remind the reader that $h_0$ is the mass associated with the instantonic symmetry. When $\phi_a = 0$, $h_0$ is related to the effective gauge coupling as

$$\frac{1}{2g_{\text{eff}}^2} = h_0 \equiv m_\lambda. \tag{105}$$

Following our general discussion in subsection 3.2, we now want to construct a domain wall such that the sign of $m_\lambda = h_0$ changes between the two sides. This $\mathbb{Z}_2$ transformation, similarly to the rank one case of the $E_1$ theory, just corresponds to the Weyl group of the UV symmetry, which for all of these models is $SO(3)$, which is enhanced from the $U(1)_I$ instantonic symmetry of the gauge theory. As we will explain in detail in the next subsection, the $C(Y^{N,N})$ geometry of these theories makes the $SO(3)$ symmetry manifest. Hence, similarly to what we did in subsection 4.6 when discussing the $E_1$ example from the point of view of the $K_{\mathbb{F}_2} = C(Y^{2,2})$ geometry, it is more convenient to work with an alternative form of the prepotential which is manifestly invariant under such $\mathbb{Z}_2$ Weyl transformation, in the same spirit of [73].

In particular, in [73] it was shown that from the IMS prepotential for the rank one $E_1$ theory with $h_0 \geq 0$

$$\mathcal{F}_{\text{IMS}} = h_0 \phi^2 + \frac{4}{3} \phi^3, \tag{106}$$

one can get a prepotential that is invariant under the $\mathbb{Z}_2$ Weyl group of the $SO(3)$ flavor symmetry by redefining

$$\varphi = \phi + \frac{h_0}{4}. \tag{107}$$

This gives, up to irrelevant constant terms, the new form of the prepotential

$$\mathcal{F}_{\text{complete}} = -\frac{h_0^2}{4} \varphi + \frac{4}{3} \varphi^3, \tag{108}$$

which is manifestly invariant under the $h_0$ sign change and hence holds on the entire ECB. The effective gauge coupling can be recovered as

$$\frac{1}{2g_{\text{eff}}^2} = \frac{1}{2} \frac{\partial^2}{\partial \varphi^2} \mathcal{F}_{\text{complete}} = 4\varphi = 4\phi + h_0. \tag{109}$$

The generalization to the higher rank theories with $SO(3)$ flavor symmetry and pure $SU(N)_N$ gauge theory description works as follows. Starting from the IMS prepotential (103) we redefine

$$\varphi_a = \phi_a + \frac{a}{2N} h_0, \qquad a = 1, \cdots, N-1,$$

(110)

so to obtain, up to irrelevant $\varphi_a$-independent terms

$$
\begin{aligned}
\mathcal{F}_{\text{complete}} &= -\frac{h_0^2}{4} \varphi_{N-1} + \frac{N}{2} \sum_{a=1}^{N-1} \left( \varphi_{a-1}^2 \varphi_a - \varphi_a \varphi_{a+1}^2 \right) \\
&\quad + \frac{4}{3} \sum_{a=1}^{N-1} \varphi_a^3 + \frac{1}{2} \sum_{a=1}^{N-1} (N-2a) \left( \varphi_{a-1}^2 \varphi_a - \varphi_a \varphi_{a+1}^2 \right) \\
&= -\frac{h_0^2}{4} \varphi_{N-1} + \frac{4}{3} \sum_{a=1}^{N-1} \varphi_a^3 + \sum_{a=1}^{N-1} (N-a) \left( \varphi_{a-1}^2 \varphi_a - \varphi_a \varphi_{a+1}^2 \right).
\end{aligned}
$$

This prepotential is now invariant under the $\mathbb{Z}_2$ Weyl group of the $SO(3)$ flavor which transforms $h_0 \to -h_0$. The case of the $SU(N)_N$ theory was not discussed in [73], so (111) is a new result about the complete prepotential of the 5d SCFTs that UV complete the $SU(N)_N$ theories.

When constructing the domain wall, we take as usual two copies of this theory related by the Weyl transformation

$$h_0 \to -h_0,$$

(111)

that is the instanton mass $h_0(x_4)$ is taken to have a profile such that it is positive for $x_4 < 0$ and negative for $x_4 > 0$, vanishing at the location of the domain wall $x_4 = 0$.

## 6.2 The geometric realization of $SU(N)_N$

The geometry that describes the 5d higher rank SCFTs that UV complete the $SU(N)_N$ gauge theories is $C(Y^{N,N})$. The toric polygon for the geometry complete resolution is drawn figure 10. The SCFT, is obtained by blowing down all the compact divisors $S_a$, and its UV flavor symmetry group $SO(3)$ [77] is manifest in the geometry, due to the non-compact divisor $D_2$.

The three relations among the divisors

$$D_1 = D_3, \qquad N D_4 + S_0 + \sum_{a=1}^{N-1} a S_a = 0, \qquad D_1 + D_2 + D_3 + D_4 + \sum_{a=1}^{N-1} S_a = 0,$$

(112)

can be solved in terms of e.g. $D_1$ and $S_a$ for $a = 1, \cdots, N-1$. We thus parametrize the Kähler form as

$$J = \mu D_1 + \sum_{a=1}^{N-1} \nu_a S_a.$$

(113)

The mapping between the Kähler parameters and the field theory parameters is achieved by comparing the complete prepotential (111) with the geometric one, which after computing the triple intersection numbers we find to be

$$\mathcal{F}_{\text{geom}} = \mu \sum_{a=1}^{N-1} (\nu_a^2 - \nu_a \nu_{a+1}) - \frac{4}{3} \sum_{a=1}^{N-1} \nu_a^3 + \sum_{a=1}^{N-1} \left( \nu_{a-1}^2 \nu_a - \nu_a \nu_{a+1}^2 \right),$$

(114)

where $\nu_0 = \nu_N = 0$. This implies the mapping

$$\mu = |h_0|, \qquad \nu_a = -\varphi_{N-a} + \frac{N-a}{2N} |h_0|, \qquad a = 1, \cdots, N-1,$$

(115)

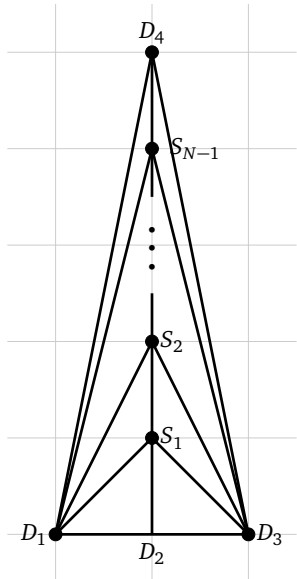

Figure 10: Fully resolved toric polygon for $C(Y^{N,N})$, which realizes the $SU(N)_N$ theory. $D_i$ denote the non-compact divisors, while $S_a$ denote the compact divisors.

and the curve volumes

$$\text{Vol}\left(C_{1S_a}\right) = \text{Vol}\left(C_{3S_a}\right) = \begin{cases} v_2 - 2v_1 = 2\varphi_{N-1} - \varphi_{N-2} - \dfrac{|h_0|}{2}, & a = 1, \\ v_{a+1} + v_{a-1} - 2v_a = 2\varphi_{N-a} - \varphi_{N-a-1} - \varphi_{N-a+1}, & a = 2, \cdots, N-2, \\ v_{N-2} - 2v_{N-1} = 2\varphi_1 - \varphi_2, & a = N-1, \end{cases}$$

$$\text{Vol}\left(C_{2S_1}\right) = \mu = |h_0|,$$

$$\begin{aligned} \text{Vol}\left(C_{S_a S_{a+1}}\right) &= \mu - 2(a+1)v_a + a\,v_{a+1} \\ &= 2(a+1)\varphi_{N-a} - 2a\varphi_{N-a-1}, \qquad a = 1, \cdots, N-2, \end{aligned}$$

$$\text{Vol}\left(C_{4S_{N-1}}\right) = \mu - 2N v_{N-1} = 2N\varphi_1. \tag{116}$$

The GLSM that describes the $C(Y^{N,N})$ geometry is

| | $D_1$ | $D_2$ | $D_3$ | $D_4$ | $S_1$ | $S_2$ | $\cdots$ | $S_{a-1}$ | $S_a$ | $S_{a+1}$ | $\cdots$ | $S_{N-2}$ | $S_{N-1}$ | FI |
|---|---|---|---|---|---|---|---|---|---|---|---|---|---|---|
| $C_{2S_1}$ | 1 | $-2$ | 1 | 0 | 0 | 0 | $\cdots$ | 0 | 0 | 0 | $\cdots$ | 0 | 0 | $t_1$ |
| $C_{1S_1}$ | 0 | 1 | 0 | 0 | $-2$ | 1 | $\cdots$ | 0 | 0 | 0 | $\cdots$ | 0 | 0 | $t_2$ |
| $\vdots$ | $\vdots$ | $\vdots$ | $\vdots$ | $\vdots$ | $\vdots$ | $\vdots$ | $\ddots$ | $\vdots$ | $\vdots$ | $\vdots$ | $\ddots$ | $\vdots$ | $\vdots$ | $\vdots$ |
| $C_{1S_a}$ | 0 | 0 | 0 | 0 | 0 | 0 | $\cdots$ | 1 | $-2$ | 1 | $\cdots$ | 0 | 0 | $t_{a+1}$ |
| $\vdots$ | $\vdots$ | $\vdots$ | $\vdots$ | $\vdots$ | $\vdots$ | $\vdots$ | $\ddots$ | $\vdots$ | $\vdots$ | $\vdots$ | $\ddots$ | $\vdots$ | $\vdots$ | $\vdots$ |
| $C_{1S_{N-1}}$ | 0 | 0 | 0 | 1 | 0 | 0 | $\cdots$ | 0 | 0 | 0 | $\cdots$ | 1 | $-2$ | $t_N$ |

$$\tag{117}$$

From the associated D-term equations we find that $C(Y^{N,N})$ can be described as a symplectic quotient as follows:

$$\mathcal{X}_{SU(N)_N}(t) = \left\{ \begin{aligned} |z_1|^2 - 2|z_2|^2 + |z_3|^2 &= t_1 \\ |z_2|^2 - 2|y_1|^2 + |y_2|^2 &= t_2 \\ |y_{a-1}|^2 - 2|y_a|^2 + |y_{a+1}|^2 &= t_{a+1}, \qquad a = 2, \cdots, N-2 \\ |z_4|^2 + |y_{N-2}|^2 - 2|y_{N-1}|^2 &= t_N \end{aligned} \right\} \Big/ U(1)^N, \tag{118}$$

where $z_i$ are the coordinates associated to the non-compact divisors $D_i$ and $y_a$ those for the compact divisors $S_a$, on which the $U(1)^N$ acts as in the table above.

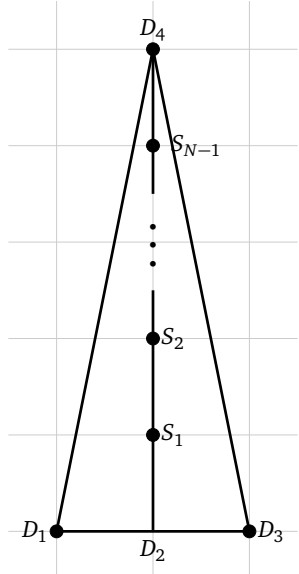

Figure 11: Ruling of the toric polygon for $C(Y^{N,N})$ that is used in the construction of the domain wall.

## 6.3 The $G_2$ geometry for domain walls

We now want to construct the domain wall. As before we want to be in a phase where we get the non-Abelian $SU(N)_N$ gauge theory. This requires the CB parameters to be switched off $\phi_a = 0$ or in terms of the parameters of the complete prepotential, recalling (110)[15]

$$\varphi_a = \frac{a}{2N}|h_0|\,. \tag{119}$$

This implies that we set the FI parameters to

$$t_1 = |h_0|\,, \qquad t_a = 0\,, \qquad a = 2, \cdots, N\,, \tag{120}$$

which also results in the curve volumes being

$$\begin{aligned}
\text{Vol}\left(C_{1S_a}\right) &= \text{Vol}\left(C_{3S_a}\right) = 0\,, & a &= 1, \cdots, N-1\,, \\
\text{Vol}\left(C_{2S_1}\right) &= \text{Vol}\left(C_{S_a S_{a+1}}\right) = \text{Vol}\left(C_{4S_{N-1}}\right) = |h_0|\,, & a &= 1, \cdots, N-1\,.
\end{aligned} \tag{121}$$

This gives the partial resolution of $C(Y^{N,N})$ shown in figure 11. Looking at the specialized D-term equations of the partially resolved geometry $\mathcal{X}_{SU(N)_N}(t_1 = |h_0|, t_2 = \cdots = t_N = 0)$ one can gauge fix $y_1, \cdots, y_{N-1}$, namely we can fix their modulus using all the equations except the first one and their argument by using the corresponding $U(1)$'s. This leaves behind a residual $\mathbb{Z}_N$ gauge symmetry acting as

$$(z_2, z_4) \to (\omega z_2, \omega^{N-1} z_4)\,, \qquad \omega^N = 1\,, \tag{122}$$

which can be found looking at the combination of the $U(1)$'s in the GLSM given by

$$\mathbb{Z}_N: \quad \sum_{a=1}^{N-1} a\, C_{1S_a}\,. \tag{123}$$

---

[15]For $N = 2$ this becomes $\varphi = \frac{1}{4}|h_0|$, which remembering that $\varphi = \phi + \frac{1}{4}h_0$ becomes equivalent to what we had in (49) for the $E_1$ case.

Finally, we obtain the $G_2$-geometry for the domain wall by fibering $\mathcal{X}_{SU(N)_N}(|h_0|, 0, \cdots, 0)$ over $x_4 \in \mathbb{R}$ with the profile $h_0 = -x_4$, i.e.

$$\mathcal{M}_{SU(N)_N} = \frac{\left\{ |z_1|^2 - 2|z_2|^2 + |z_3|^2 = |x_4| , z_4 \in \mathbb{C} \mid x_4 \in \mathbb{R} \right\}}{U(1) \times \mathbb{Z}_N} , \tag{124}$$

where the $U(1)$ acts as in the first row of (117), while the $\mathbb{Z}_N$ acts as (122).

By a similar analysis to the one done for the $N = 2$ case of $\mathcal{M}_{E_1}$ in section 4, we can write this space as

$$\mathcal{M}_{SU(N)_N} = \left( \mathcal{O}_{\mathbb{P}^1}(-2) \oplus \mathcal{O}_{\mathbb{P}^1}(0) \right) / \mathbb{Z}_N \rtimes \mathbb{R} \cong \left( \mathcal{O}_{\mathbb{P}^1}(-1) \oplus \mathcal{O}_{\mathbb{P}^1}(-1) \right) / \mathbb{Z}_N \rtimes \mathbb{R} , \tag{125}$$

with the $\mathbb{Z}_N$ acting on the fibers with the origin being the only fixed point.

According to [3], the resulting 4d $\mathcal{N} = 1$ theory that lives on the domain wall we just realized is given by an $\mathfrak{su}(N) \oplus \mathfrak{su}(N)$ bifundamental $Q$ plus the flipping field $S$

$$\boxed{N} \xrightarrow[\ Q\ ]{\ S\ } \times \longleftarrow \boxed{N} \tag{126}$$

where an ingoing arrow means fundamental representation, while an outgoing one means anti-fundamental.

The two $\mathfrak{su}(N)$ symmetries come from the $A_{N-1}$ singularity associated with the divisors $S_a$ so they are flavor symmetries from the 4d perspective but they are gauged in the 5d bulk. As these two singularities meet at $x_4 = 0$, there is furthermore a bifundamental chiral $Q$, as well as the singlet chiral $S$ in the spectrum. In this case the operator that is flipped by $S$ is the baryon constructed from the bifundamental, namely we have the superpotential interaction

$$\mathcal{W} = S \det Q = S \, \epsilon_{a_1 \cdots a_N} \epsilon^{i_1 \cdots i_N} Q^{a_1}_{i_1} \cdots Q^{a_N}_{i_N} , \tag{127}$$

but this is again irrelevant in the IR since the model has no 4d gauge symmetry.

## 6.4 Adding flavors: $G_2$-manifolds from 5d SQCD

A straight-forward generalization is adding flavors to the $SU(N)_N$ theories, while also changing appropriately the CS levels. The generalized toric polygon for these SCFTs is generically not toric (see e.g. [86,87]), however we again go into the particular Kähler cone locus where there is a gauge theory description in terms of an $SU(N)_k + N_F \boldsymbol{F}$ gauge theory with a particular constraint on the masses of the flavors, so that the polygon is actually toric.

We will in particular focus on the case where the CS level is $k = N - \frac{N_F}{2}$. The toric polygon for these theories in the partial resolution we are interested in is shown in figure 12. As in the case with no flavors, the non-compact divisor $D_2$ makes manifest an $SO(3)$ symmetry of the SCFTs. The domain wall is then constructed using its $\mathbb{Z}_2$ Weyl transformation. This acts as before as a sign change this $SO(3)$ symmetry mass

$$m_\lambda \to m_\lambda = -m_\lambda , \tag{128}$$

where similarly to the case $N = 2$ with flavors, $m_\lambda$ is identified with the following combination of the instantonic and the baryonic symmetries of the gauge theory:

$$m_\lambda = h_0 - \frac{N_F}{2} m . \tag{129}$$

Here we are once again taking $\phi_a = 0$ and all the hyper-multiplet masses to be equal and positive $m_i = m > 0$. They transform under the domain wall action as [3]

$$m \to m + \frac{m_\lambda}{N} . \tag{130}$$

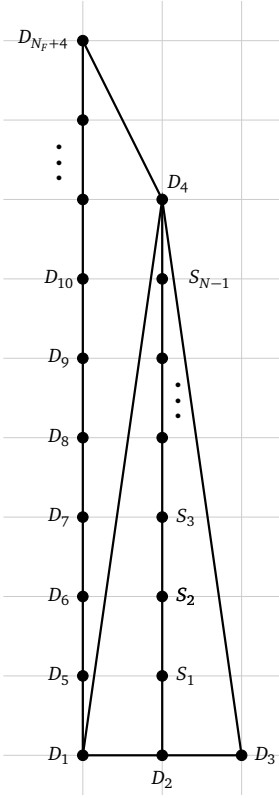

Figure 12: Partially resolved toric polygon for $SU(N)_k + N_F \boldsymbol{F}$ with CS level $k = N - \frac{N_F}{2}$. Again the curve $C_{14} = D_1 \cdot D_4$ cannot be blowndown, and corresponds to one of the non-vanishing ECB parameters. The resulting polygon is perfectly well defined as a toric variety.

The above transformations imply that while $m_\lambda$ is the combination of $h_0$ and $m$ that changes its sign, the orthogonal combination that is trivially identified is

$$m_c = m_\lambda + 2Nm. \tag{131}$$

All of the curve volumes in the partial resolution of figure 12 depend on only two Kähler parameters and they can be expressed in terms of the only two independent field theory mass parameters that we have in our setup

$$\text{Vol}\big(C_{2S_1}\big) = \text{Vol}\big(C_{S_a S_{a+1}}\big) = \text{Vol}\big(C_{4S_{N-1}}\big) = |m_\lambda|, \qquad a = 1, \cdots, N-1,$$
$$\text{Vol}\big(C_{14}\big) = \frac{m_c - |m_\lambda|}{2N}, \tag{132}$$

while all the other curve volumes are zero. The seven-dimensional domain wall geometry is then obtained once again by fibering over $x_4 \in \mathbb{R}$ with

$$m_\lambda = -x_4, \tag{133}$$

and $m_c$ constant along $x_4$.

The 4d $\mathcal{N} = 1$ domain wall theory was found in [3] to be

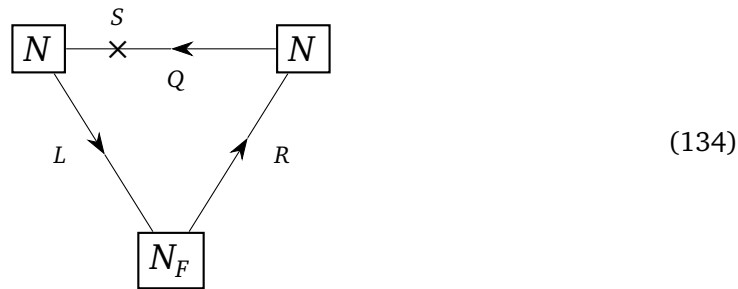

(134)

As in the previous examples, the two $\mathfrak{su}(N)$ symmetries come from the $A_{N-1}$ singularity associated with the divisors $S_a$ such that they are flavor symmetries from the 4d perspective while being gauged in the 5d bulk. The $\mathfrak{su}(N_F)$ symmetry comes from the $A_{N_F-1}$ singularity associated with the non-compact divisors $D_5, \cdots, D_{N_F+4}$, such that it is a flavor symmetry both from the 4d and the 5d points of view. This quiver is supplemented with the superpotential

$$\mathcal{W} = S \det Q + L Q R. \tag{135}$$

The field theory description can be understood from the geometrical perspective in a similar manner to the $N = 2$ cases discussed earlier. In addition, one can again stack such domain walls.

# 7 Gluing across a 5d UV duality: The beetle and the millipede

In this section we consider duality domain walls that do not relate two copies of the same gauge theory with a $\mathbb{Z}_2$ Weyl identification, but rather two genuinely different gauge theories that are UV dual, that is they are UV completed by the same SCFT. An example of this construction was also studied from the field theory in [3], but here we consider as the main example the case of the UV duality between the $SU(k)_0 + (2k-4)\boldsymbol{F}$ theory and a linear quiver with $k-1$ $SU(2)$ gauge nodes connected by one bifundamental hyper-multiplet and comment on the resulting domain wall geometry.

## 7.1 The beetle domain wall between $SU(2)_\pi \times SU(2)_\pi$ and $SU(3)_0 + 2\mathbf{F}$

Consider the duality domain wall between the gauge theories $SU(2)_\pi \times SU(2)_\pi$ with one bifundamental hyper-multiplet and $SU(3)_0 + 2\boldsymbol{F}$. These are UV completed by the same rank 2 SCFT, whose singular toric polygon we draw in figure 13, called the "beetle".

The IMS prepotentials of the two gauge theories are

$$
\begin{aligned}
\mathcal{F}_{\text{IMS}}^{SU(2)_\pi \times SU(2)_\pi} = {}& h_1 \phi_1^2 + h_2 \phi_2^2 + \frac{4}{3}\left(\phi_1^3 + \phi_2^3\right) \\
& - \frac{1}{12}\left(|\phi_1 + \phi_2 + m|^3 + |\phi_1 - \phi_2 + m|^3 + |-\phi_1 + \phi_2 + m|^3 + |-\phi_1 - \phi_2 + m|^3\right), \\
\mathcal{F}_{\text{IMS}}^{SU(3)_0 + 2\boldsymbol{F}} = {}& \widetilde{h}_0\left(\widetilde{\phi}_1^2 + \widetilde{\phi}_2^2 - \widetilde{\phi}_1\widetilde{\phi}_2\right) + \frac{4}{3}\left(\widetilde{\phi}_1^3 + \widetilde{\phi}_2^3\right) - \frac{1}{2}\left(\widetilde{\phi}_1^2\widetilde{\phi}_2 + \widetilde{\phi}_1\widetilde{\phi}_2^2\right) \\
& - \frac{1}{12}\sum_{i=1}^{2}\left(|\widetilde{\phi}_1 + \widetilde{m}_i|^3 + |-\widetilde{\phi}_1 + \widetilde{\phi}_2 + \widetilde{m}_i|^3 + |-\widetilde{\phi}_2 + \widetilde{m}_i|^3\right).
\end{aligned}
\tag{136}
$$

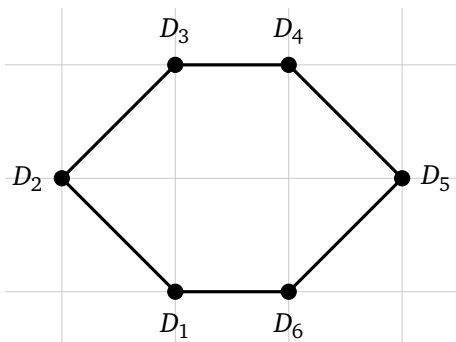

Figure 13: The singular toric polygon of the "beetle" SCFT that UV completes both the $SU(3)_0 + 2F$ and the $SU(2)_\pi \times SU(2)_\pi$ theories.

The mapping between the parameters of the two theories has been worked out in [11][16]

$$
\begin{aligned}
\widetilde{\phi}_1 &= \phi_1 + \frac{2h_1 + h_2}{3}, & \widetilde{\phi}_2 &= \phi_2 + \frac{h_1 + 2h_2}{3}, & \widetilde{h}_0 &= -h_1 - h_2, \\
\widetilde{m}_1 &= \frac{h_1 - h_2}{3} - m, & \widetilde{m}_2 &= \frac{h_1 - h_2}{3} + m.
\end{aligned}
\tag{137}
$$

Notice that this mapping tells us that the diagonal combination of the two instantonic symmetries on the quiver side gets mapped to the instantonic symmetry of the $SU(3)_0 + 2F$ theory and an off-diagonal combination $\frac{1}{3}(h_1 - h_2)$ of them gets instead mapped to the $\mathfrak{u}(1)_B$ baryonic symmetry defined by writing the flavor symmetry as $\mathfrak{u}(2) \cong \mathfrak{su}(2) \oplus \mathfrak{u}(1)_B$, while the $\mathfrak{su}(2)$ symmetries on both sides are directly identified.

The duality domain wall is constructed by taking all the CB parameters as well as the hyper-multiplet masses to zero and on the $SU(2)_\pi \times SU(2)_\pi$ side the two instanton masses are tuned to be equal

$$
\begin{aligned}
SU(2)_\pi \times SU(2)_\pi: & \quad \phi_1 = \phi_2 = m = 0, & h_1 = h_2, \\
SU(3)_0 + 2F: & \quad \widetilde{\phi}_1 = \widetilde{\phi}_2 = \widetilde{m}_1 = \widetilde{m}_2 = 0.
\end{aligned}
\tag{138}
$$

This in particular implies that the effective gauge couplings are given by

$$
\begin{aligned}
SU(2)_\pi \times SU(2)_\pi: & \quad \frac{1}{2\left(g_{\text{eff}}^{(1)}\right)^2} = \frac{1}{2\left(g_{\text{eff}}^{(2)}\right)^2} = h_1 = h_2, \\
SU(3)_0 + 2F: & \quad \frac{1}{2\widetilde{g}_{\text{eff}}^2} = \widetilde{h}_0,
\end{aligned}
\tag{139}
$$

where $g_{\text{eff}}^{(a)}$ for $a = 1, 2$ are the two $SU(2)$ effective gauge couplings, while $\widetilde{g}_{\text{eff}}$ is the $SU(3)$ one. We are then led to identify

$$
m_\lambda = h_1 = h_2 = -\frac{\widetilde{h}_0}{2},
\tag{140}
$$

---

[16]Compared to [11], whose parameters we denote with a CDS superscript, we redefined

$$
h_1 = h_1^{\text{CDS}} - m, \qquad h_2 = h_2^{\text{CDS}} - m, \qquad \widetilde{h}_0 = \widetilde{h}_0^{\text{CDS}} - \frac{\widetilde{m}_1 + \widetilde{m}_2}{2},
$$

to conform with the conventions of [8] which we are using. We also point out for the comparison that the bare CS level of [8] corresponds to the effective CS level of [11].

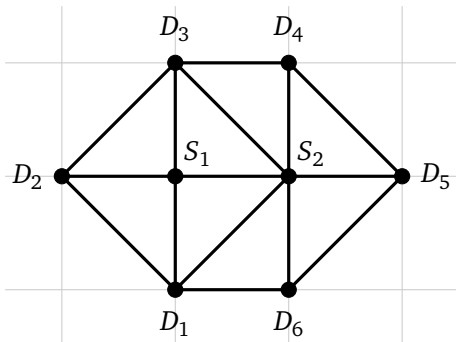

Figure 14: One of the full resolutions of the toric polygon for the rank 2 SCFT which realizes both the $SU(2)_\pi \times SU(2)_\pi$ theory in the ECB chamber with $\pm\phi_1 + \phi_2 + m > 0$, $\pm\phi_1 - \phi_2 + m < 0$ and the $SU(3)_0 + 2F$ theory in the ECB chamber with $\widetilde{\phi}_1 + \widetilde{m}_i > 0$, $-\widetilde{\phi}_1 + \widetilde{\phi}_2 + \widetilde{m}_i > 0$ and $-\widetilde{\phi}_2 + \widetilde{m}_i < 0$.

as the parameter that changes its sign across the domain wall, going from a positive value on the left $m_\lambda(x_4 < 0) > 0$ to a negative value on the right $m_\lambda(x_4 > 0) < 0$. The above identification then indicates that on the left the $SU(2)_\pi \times SU(2)_\pi$ description is valid since here $(g^1_{\text{eff}})^2 = (g^2_{\text{eff}})^2 > 0$, while on the right we should switch to the description in terms of $SU(3)_0 + 2F$ since $\widetilde{g}^2_{\text{eff}} > 0$, while $(g^1_{\text{eff}})^2 = (g^2_{\text{eff}})^2 < 0$.

We now want to translate these field theory considerations in terms of the geometry. We will start from the resolution of the toric diagram where $\pm\phi_1 + \phi_2 + m > 0$, $\pm\phi_1 - \phi_2 + m < 0$ or equivalently $\widetilde{\phi}_1 + \widetilde{m}_i > 0$, $-\widetilde{\phi}_1 + \widetilde{\phi}_2 + \widetilde{m}_i > 0$, $-\widetilde{\phi}_2 + \widetilde{m}_i < 0$, as shown in figure 14. We do not discuss explicitly all the details of the matching between the field theory and the geometry parameters, for which we refer the reader to [11], and instead provide only the final result for the curve volumes

$$
\begin{aligned}
\text{Vol}\big(C_{1S_1}\big) &= 2\phi_1 - \phi_2 + h_1 = 2\widetilde{\phi}_1 - \widetilde{\phi}_2 \,, \\
\text{Vol}\big(C_{1S_2}\big) &= -\phi_1 + \phi_2 + m = -\widetilde{\phi}_1 + \widetilde{\phi}_2 + \widetilde{m}_2 \,, \\
\text{Vol}\big(C_{2S_1}\big) &= 2\phi_1 = 2\widetilde{\phi}_1 + \widetilde{h}_0 - \frac{\widetilde{m}_1 + \widetilde{m}_2}{2} \,, \\
\text{Vol}\big(C_{3S_1}\big) &= 2\phi_1 - \phi_2 + h_1 = 2\widetilde{\phi}_1 - \widetilde{\phi}_2 \,, \\
\text{Vol}\big(C_{3S_2}\big) &= -\phi_1 + \phi_2 - m = -\widetilde{\phi}_1 + \widetilde{\phi}_2 + \widetilde{m}_1 \,, \\
\text{Vol}\big(C_{S_1 S_2}\big) &= 2\phi_1 = 2\widetilde{\phi}_1 + \widetilde{h}_0 - \frac{\widetilde{m}_1 + \widetilde{m}_2}{2} \,, \\
\text{Vol}\big(C_{4S_2}\big) &= \phi_2 + h_2 + m = \widetilde{\phi}_2 - \widetilde{m}_1 \,, \\
\text{Vol}\big(C_{5S_2}\big) &= 2\phi_2 = 2\widetilde{\phi}_2 + \widetilde{h}_0 + \frac{\widetilde{m}_1 + \widetilde{m}_2}{2} \,, \\
\text{Vol}\big(C_{6S_2}\big) &= \phi_2 + h_2 - m = \widetilde{\phi}_2 - \widetilde{m}_2 \,.
\end{aligned}
\tag{141}
$$

As usual, when translating to the geometry the specialization (138) of the field theory parameters that describes the domain wall, it is useful to translate it in terms of the parameters that we have on the left which we can extend to the entire $x_4 \in \mathbb{R}$ direction. Combining (137) and (138) we find

$$
m_\lambda = -x_4 \,, \qquad \phi_a = \frac{|m_\lambda| - m_\lambda}{2} \,.
\tag{142}
$$

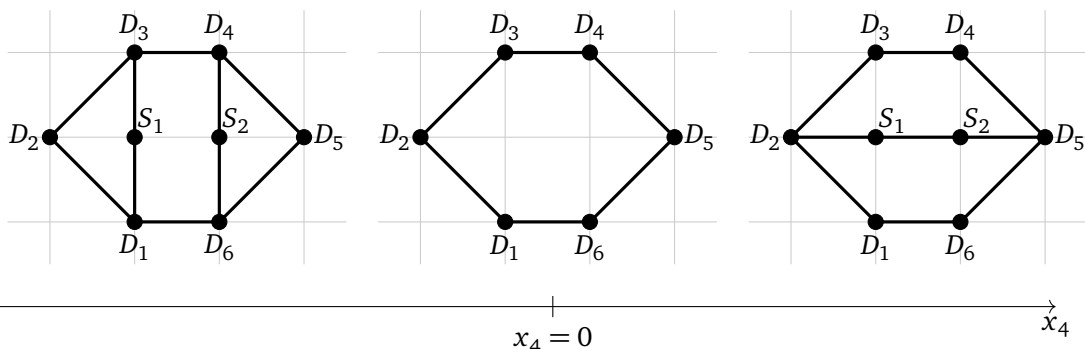

Figure 15: Geometry of the domain wall between the $SU(2)_\pi \times SU(2)_\pi$ and the $SU(3)_0 + 2F$ theory.

In this way, most of the curves shrink to zero, so that the only non-trivial curves on the two sides of the domain wall are

$$
\begin{aligned}
SU(2)_\pi \times SU(2)_\pi : \quad &\mathrm{Vol}\left(C_{1S_1}\right) = \mathrm{Vol}\left(C_{3S_1}\right) = \mathrm{Vol}\left(C_{4S_2}\right) = \mathrm{Vol}\left(C_{6S_2}\right) = m_\lambda , \\
SU(3)_0 + 2F : \quad &\mathrm{Vol}\left(C_{2S_1}\right) = \mathrm{Vol}\left(C_{S_1S_2}\right) = \mathrm{Vol}\left(C_{5S_2}\right) = -2m_\lambda ,
\end{aligned}
\tag{143}
$$

while all the curves that are not indicated on each side are absent. Notice in particular that all the surviving curves consistently have positive volume, since the $SU(2)_\pi \times SU(2)_\pi$ description holds for $x_4 < 0$ where $m_\lambda$ is positive while the $SU(3)_0 + 2F$ holds for $x_4 > 0$ where $m_\lambda < 0$. At the location of the domain wall $x_4 = 0$ instead all of the curves collapse since $m_\lambda = 0$ and we have the singular geometry shown in figure 13. The overall structure of the geometry of this domain wall is depicted in figure 15.

In order to characterize the domain wall geometry more explicitly, we can as usual use the GLSM description

|            | $D_1$ | $D_2$ | $D_3$ | $D_4$ | $D_5$ | $D_6$ | $S_1$ | $S_2$ | FI |
|------------|-------|-------|-------|-------|-------|-------|-------|-------|-----|
| $C_{3S_1}$ | 0 | 1 | 0 | 0 | 0 | 0 | $-2$ | 1 | $t_1 = 2\phi_1 - \phi_2 + m_\lambda$ |
| $C_{3S_2}$ | 0 | 0 | $-1$ | 1 | 0 | 0 | 1 | $-1$ | $t_2 = 0$ |
| $C_{6S_2}$ | 1 | 0 | 0 | 0 | 1 | $-1$ | 0 | $-1$ | $t_3 = \phi_2 + m_\lambda$ |
| $C_{1S_2}$ | $-1$ | 0 | 0 | 0 | 0 | 1 | 1 | $-1$ | $t_4 = 0$ |
| $C_{S_1S_2}$ | 1 | 0 | 1 | 0 | 0 | 0 | $-2$ | 0 | $t_5 = 2\phi_1$ |

$$\tag{144}$$

where we are expressing the FI parameters in terms of the field theory parameters on the left of the domain wall.

The associated D-term vacuum equations are

$$
\begin{aligned}
|z_2|^2 - 2|y_1|^2 + |y_2|^2 &= 2\phi_1 - \phi_2 + m_\lambda , \\
-|z_3|^2 + |z_4|^2 + |y_1|^2 - |y_2|^2 &= 0 , \\
|z_1|^2 + |z_5|^2 - |z_6|^2 - |y_2|^2 &= \phi_2 + m_\lambda , \\
-|z_1|^2 + |z_6|^2 + |y_1|^2 - |y_2|^2 &= 0 , \\
|z_1|^2 + |z_3|^2 - 2|y_1|^2 &= 2\phi_1 ,
\end{aligned}
\tag{145}
$$

where as usual we denote by $z_i$ the homogeneous coordinates associated with the non-compact divisors $D_i$ and with $y_a$ those associated with the compact divisors $S_a$. By taking suitable linear

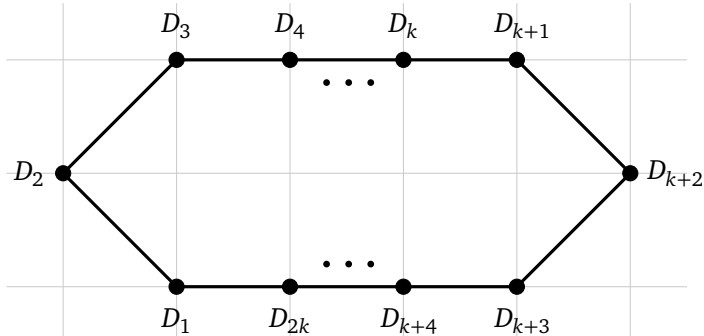

Figure 16: The singular toric polygon of the "millipede" SCFT that UV completes both the $SU(2)^{k-1}$ linear quiver theory and the $SU(k)_0 + (2k-4)F$ theory.

combinations and remembering that $\phi_1 = \phi_2$, we can rewrite this set of equations as

$$
\begin{aligned}
|z_2|^2 - |z_3|^3 + |z_5|^2 - |z_6|^2 &= 2m_\lambda, \\
|z_1|^2 - |z_3|^2 + |z_4|^2 - |z_6|^2 &= 0, \\
-2|z_1|^2 + |z_2|^2 - |z_3|^2 + |z_4|^2 - |z_5|^2 + 2|z_6|^2 &= 0, \\
|z_1|^2 + |z_3|^2 &= 2|y_1|^2 + 2\phi_1, \\
|z_4|^2 + |z_6|^2 &= 2|y_2|^2 + 2\phi_1.
\end{aligned}
\tag{146}
$$

Notice in particular that the last two equations allow us to gauge fix the coordinates $y_1$, $y_2$ for the blown-down compact divisors $S_1$, $S_2$ up to two independent $\mathbb{Z}_2$ transformations

$$
(z_1, z_3) \to (-z_1, -z_3), \qquad (z_4, z_6) \to (-z_4, -z_6).
\tag{147}
$$

By fibering this partially resolved geometry over $x_4 \in \mathbb{R}$ with profile of the parameters given in (142), we obtain the seven-dimensional domain wall geometry

$$
\mathcal{M}_{\text{Beetle}} = \left\{
\begin{aligned}
|z_2|^2 - |z_3|^3 + |z_5|^2 - |z_6|^2 &= -2x_4 \quad | \quad x_4 \in \mathbb{R} \\
|z_1|^2 - |z_3|^2 + |z_4|^2 - |z_6|^2 &= 0 \\
-2|z_1|^2 + |z_2|^2 - |z_3|^2 + |z_4|^2 - |z_5|^2 + 2|z_6|^2 &= 0
\end{aligned}
\right\} \Big/ U(1)^3 \times \mathbb{Z}_2^2,
\tag{148}
$$

where the three $U(1)$'s act with the charges appearing as coefficients in the equations and the two $\mathbb{Z}_2$'s were defined above.

## 7.2 Millipede domain walls between $SU(2)^{k-1}$ and $SU(k)_0 + (2k-4)F$

The previous UV-duality domain wall can be generalized to the case of the duality between the linear quiver with $k-1$ $SU(2)$ gauge nodes where adjacent nodes are connect by one bifundamental hyper-multiplet and the $SU(k)_0 + (2k-4)F$ theory for generic $k$. Notice that for $k = 2$ we have the same pure $SU(2)_0$ gauge theory on both sides and we indeed recover the $\mathbb{Z}_2$ Weyl domain wall for the rank 1 $E_1$ SCFT that we studied in detail in section 4. We discussed the case $k = 3$ in the previous subsection and we give more details about the case $k = 4$ in appendix B, while here we will only make some general comments for generic $k$.

The toric polygon that describes the singular geometry, corresponding to the common UV SCFT of both gauge theories, is depicted in figure 16 and (in reference to the beetle) we refer to as the "millipede". As before, we want to consider a particular partial resolution for the theories on the two sides. Specifically, on both sides we set all the CB parameters to zero and

we also turn off all the hyper-multiplet masses. Moreover, on the linear quiver side we also tune all the instanton masses of the leftmost and the rightmost gauge nodes to be equal, while we turn off the others. This leaves only one parameter that we define as $m_\lambda$

$$
\begin{aligned}
SU(2)^{k-1}: \quad &\phi_1 = \cdots = \phi_{k-1} = m_1 = \cdots = m_{k-2} = 0\,, \\
&h_1 = h_{k-1} = m_\lambda\,, \\
&h_2 = \cdots = h_{k-2} = 0\,, \\
SU(k)_0 + (2k-4)\boldsymbol{F}: \quad &\widetilde{\phi}_1 = \cdots = \phi_{k-1} = \widetilde{m}_1 = \cdots = \widetilde{m}_{2k-4} = 0\,, \\
&\widetilde{h}_0 = -2m_\lambda\,.
\end{aligned}
\tag{149}
$$

The reason for this choice is due to the symmetry mapping across the UV duality. Specifically, the diagonal combination of all the instantonic symmetries of the quiver gets mapped to the instantonic symmetry of the $SU(k)_0 + (2k-4)\boldsymbol{F}$ theory, while an off-diagonal combination of those of the two gauge nodes at the two ends of the quiver gets mapped to the $\mathfrak{u}(1)_B$ baryonic symmetry in the $\mathfrak{u}(2k-4) \cong \mathfrak{su}(2k-4) \oplus \mathfrak{u}(1)_B$ flavor symmetry. On the other hand, the remaining $k-3$ instantonic symmetries together with the $k-2$ $\mathfrak{su}(2)$ symmetries that act of the bifundamental hypers in the quiver theory gets enhanced to $\mathfrak{su}(2k-4)$ that is manifest as the flavor symmetry in the $SU(k)_0 + (2k-4)\boldsymbol{F}$ theory. This is due to the fact that the corresponding gauge nodes in the quiver are balanced, i.e. the number of flavors is twice the number of colors. Hence, turning on only $h_1 = h_{k-1} = m_\lambda$ on the quiver side corresponds to turning on only $\widetilde{h}_0 = -2m_\lambda$ on the $SU(k)_0 + (2k-4)\boldsymbol{F}$ side. We will confirm this matching of symmetries explicitly for $k = 4$ in appendix B, where we will also discuss an alternative possible domain wall set-up for that special case.

With this specialization we get the following effective gauge couplings

$$
\begin{aligned}
SU(2)^{k-1}: \quad &\frac{1}{2\left(g_{\text{eff}}^{(1)}\right)^2} = \frac{1}{2\left(g_{\text{eff}}^{(k-1)}\right)^2} = m_\lambda\,, \\
&\frac{1}{2\left(g_{\text{eff}}^{(2)}\right)^2} = \cdots = \frac{1}{2\left(g_{\text{eff}}^{(k-2)}\right)^2} = 0\,, \\
SU(k)_0 + (2k-4)\boldsymbol{F}: \quad &\frac{1}{2\widetilde{g}_{\text{eff}}^2} = -2m_\lambda\,.
\end{aligned}
\tag{150}
$$

Notice in particular that the couplings for all the nodes in the linear quiver except the first and the last diverge, meaning that we actually have a non-Lagrangian 5d theory on this side of the wall. When we construct the domain wall we take $m_\lambda$ to be positive for $x_4 < 0$ and negative for $x_4 > 0$. On the domain wall $x_4 = 0$, $m_\lambda$ vanishes, meaning all the effective gauge couplings diverge.

The picture of the resulting geometry is represented in figure 17, where one can check the curve volumes for the case $k = 4$ using the results of appendix B. Notice that on the left side of the domain wall we have the ruling with only two vertical curves, which we can think of as realizing an $SU(2) \times SU(2)$ gauging of the two $SU(2)$ symmetries of the SCFT that UV completes the $SU(k-2) + (2k-4)\boldsymbol{F}$ theory. This is a non-Lagrangian theory that arises from having set some of the gauge couplings in the linear quiver to infinite value, as we commented above, and it is encoded in the rectangle with no internal edges appearing in the polygon on the left of figure 17. On the right side instead we have the partial resolution with only one of the horizontal curves, corresponding to the $SU(k)_0 + (2k-4)\boldsymbol{F}$ theory with massless hypers. Finally, on the wall we have the singular geometry corresponding to the SCFT point.

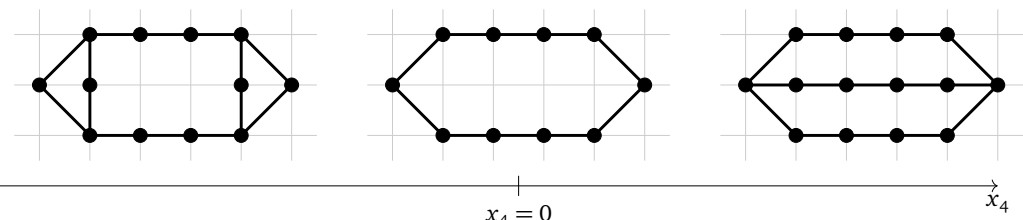

Figure 17: Geometry of the domain wall between the non-Lagrangian limit of the $SU(2)^{k-1}$ quiver and the $SU(k)_0 + (2k-4)\mathbf{F}$ theory. We show the case $k = 5$.

# Acknowledgments

We thank Bobby Acharya, Fabio Apruzzi, Antoine Bourget, Lorenzo Foscolo, Mark Haskins, Jason Lotay, and Shlomo Razamat for useful discussions. AB, ES and SSN thank the KITP for hospitality during the Simons Collaboration on Special Holonomy workshop 3/2023, where part of this work was completed.

**Funding information.** This work was supported by the ERC Consolidator grants 682608 (ES, SSN) and 864828 (MS), the "Simons Collaboration on Special Holonomy in Geometry, Analysis and Physics" (ES, SSN), the "Simons Collaboration for the Nonperturbative Bootstrap" under grant 494786 (MS), and the EPSRC Open Fellowship EP/X01276X/1 (SSN).

# A   The $G_2$-geometry for $E_3$ domain walls

In this appendix we present the details of the domain wall construction we discussed in section 5 for the $E_{N_F+1}$ SCFTs in the particular case of $N_F = 2$.

## A.1   The field theory analysis

The 5d rank 1 $E_3$ SCFT admits a mass deformation to the $SU(2)+2\mathbf{F}$ gauge theory, whose IMS prepotential reads

$$\mathcal{F}_{SU(2),N_F=2} = h_0 \phi^2 + \frac{4}{3}\phi^3 - \frac{1}{12}\sum_{i=1}^{2}\sum_{\pm} |\pm \phi + m_i|^3, \tag{A.1}$$

where as usual we are assuming that $h_0 > 0$.

When constructing the domain wall, we take the 5d theory to be in an ECB chamber where $\pm\phi + m_i > 0$. Moreover, on both sides of the wall we take all the hyper-multiplet masses to be identical and, on the left side, we set $\phi^{(L)} = 0$ such that we have the full $SU(2)$ gauge symmetry

$$\phi^{(L)} = 0, \qquad m_1^{(I)} = m_2^{(I)} = m^{(I)} > 0, \qquad I = L, R. \tag{A.2}$$

The domain wall is characterized by the following $\mathbb{Z}_2$ transformation that relates the theories on the two sides of the domain wall ($x_4 > 0$):

$$m_\lambda^{(R)}(x_4) = -m_\lambda^{(L)}(-x_4), \qquad m^{(R)}(x_4) = m^{(L)}(-x_4) + \frac{m_\lambda^{(L)}(-x_4)}{2}, \tag{A.3}$$

where

$$m_\lambda^{(I)} = h_0^{(I)} - m^{(I)}. \tag{A.4}$$

One can check that this transformation is part of the Weyl group of the $\mathfrak{e}_3 = \mathfrak{su}(3) \oplus \mathfrak{su}(2)$ symmetry of the UV SCFT. Notice also that the combination

$$m_c^{(I)} = h_0^{(I)} + 3m^{(I)}, \tag{A.5}$$

is instead trivially identified between the two sides. This can be modeled by having a globally defined $m_\lambda(x_4)$ such that it is positive for $x < 0$ and negative for $x_4 > 0$, while $m_c(x_4) = m_c$ is constant.

As we explained in subsection 3.2, after the above transformation of the mass parameters we should also redefine the CB parameter in order to have a sensible gauge theory description on the right side of the domain wall

$$\phi^{(L)} \to \phi^{(R)} = \phi^{(L)} - \frac{m_\lambda^{(L)}}{2}. \tag{A.6}$$

Because of this, the effective gauge coupling on the two sides of the domain wall is

$$\frac{1}{\left(g_{\text{eff}}^{(I)}\right)^2} = |m_\lambda| > 0, \tag{A.7}$$

and similarly the physical masses of the hypers are

$$m_{\text{phys}}^{(I)} = \frac{m_c - |m_\lambda|}{4}, \tag{A.8}$$

which will be measured by the volume of some specific curve in the geometry, as we will see next. In particular, we point out that these physical masses of the hypers $m_{\text{phys}}^{(I)}$ are not necessarily the $m^{(I)}$ defined above, which in particular is what happens on the right side.

## A.2 The rank 1 $E_3$ theory geometry

The $E_3$ theory, admitting the $SU(2) + 2F$ gauge theory phase, can be described by the gdP$_3$ geometry, whose toric polygon in the specific resolution where $\phi \pm m_i > 0$ is shown in figure 18. The linear relations are simply $\sum v_{x,y,z}^i D_i = 0$, i.e.

$$D_2 + 2D_3 + D_4 + S \cong 0, \qquad -D_2 + D_4 + D_5 + 2D_6 \cong 0, \qquad S + \sum_{i=1}^{6} D_i \cong 0. \tag{A.9}$$

They imply that e.g. we can replace $D_3$, $D_4$, $D_5$ in terms of the other divisors, and the Kähler form can be parameterized by

$$J = \mu_1 D_1 + \mu_2 D_2 + \mu_6 D_6 + \nu S. \tag{A.10}$$

In order to obtain the geometric prepotential $\mathcal{F}_{\text{geom}} = -\frac{1}{6}J^3$ we need to compute the triple intersection numbers, which we can do using (A.9) and the fact that $D_i \cdot D_j \cdot D_k = 1$ only if $D_i$, $D_j$, $D_k$ are the vertices of the same cone in the toric polygon while it is zero otherwise, and similarly for $D_i \cdot D_j \cdot S$. In this way we find the relevant[17] non-zero intersection numbers to be

$$\begin{aligned}
SD_1^2 &= -1, & SD_6^2 &= -1, & SD_1 D_2 &= 1, \\
S^2 D_1 &= -1, & S^2 D_2 &= -2, & S^2 D_6 &= -1, & S^3 &= 6.
\end{aligned} \tag{A.11}$$

[17]By relevant we mean those that contain the compact divisor $S$ at least once. This is because as we will see only the Kähler parameter $\nu$ is related to $\phi$, so that all the triple intersection numbers that do not involve $S$ would lead to terms that do not depend on $\phi$ on the prepotential.

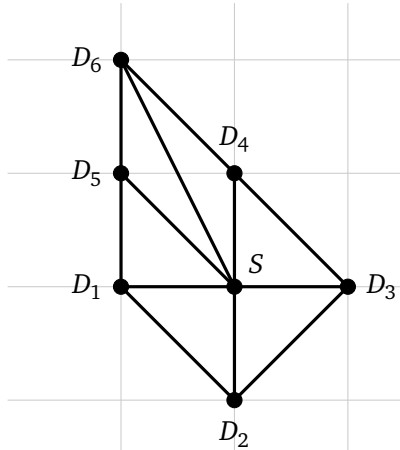

Figure 18: One of the full resolutions of the toric polygon for gdP$_3$, which realizes the $SU(2) + 2F$ theory in the ECB chamber with $\phi \pm m_i > 0$.

This implies that the geometric prepotential is (up to $\phi$-independent terms)

$$\mathcal{F}_{\text{geom}} = -\nu^3 + \nu^2 \left( \frac{\mu_1}{2} + \mu_2 + \frac{\mu_6}{2} \right) + \nu \left( \frac{\mu_1^2}{2} + \frac{\mu_6^2}{2} - \mu_1 \mu_2 \right). \tag{A.12}$$

We would like to find the mapping between the Kähler parameters of the geometry and the mass parameters of the field theory. In order to do this, we compare the geometric prepotential with the IMS prepotential (A.1), which for $\phi \pm m_i > 0$ reads

$$\mathcal{F}_{\text{IMS}} = \phi^3 + h_0 \phi^2 - \frac{1}{2} \phi \left( m_1^2 + m_2^2 \right). \tag{A.13}$$

To facilitate the comparison, we consider the ruling with fiber $C_{3S}$, which has volume

$$\text{Vol}(C_{3S}) = D_3 \cdot S \cdot J = \mu_2 - 2\nu. \tag{A.14}$$

We can then make the Ansatz

$$\text{Vol}(C_{3S}) = 2\phi \quad \Rightarrow \quad \nu = -\phi + \frac{\mu_2}{2}. \tag{A.15}$$

With this, the geometric prepotential becomes (up to $\phi$-independent terms)

$$\mathcal{F}_{\text{geom}} = \phi^3 + \phi^2 \left( \frac{\mu_1}{2} - \frac{\mu_2}{2} + \frac{\mu_6}{2} \right) - \frac{1}{2} \phi \left( \mu_1^2 - \mu_1 \mu_2 + \frac{\mu_2^2}{2} + \mu_2 \mu_6 + \mu_6^2 \right), \tag{A.16}$$

and comparing with (A.13) gives

$$h_0 = \frac{\mu_1}{2} - \frac{\mu_2}{2} + \frac{\mu_6}{2},$$

$$m_1^2 + m_2^2 = \mu_1^2 - \mu_1 \mu_2 + \frac{\mu_2^2}{2} + \mu_2 \mu_6 + \mu_6^2 = \left( \frac{\mu_2}{2} - \mu_1 \right)^2 + \left( \mu_6 + \frac{\mu_2}{2} \right)^2, \tag{A.17}$$

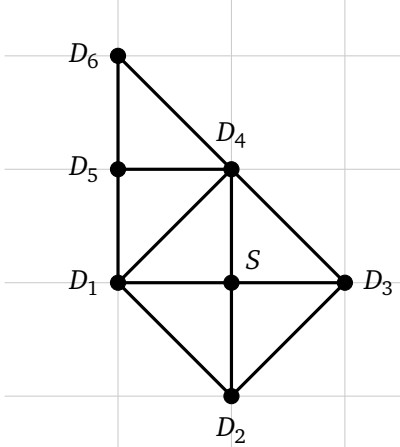

Figure 19: One of the full resolutions of the toric polygon for gdP$_3$, which realizes the $SU(2) + 2F$ theory in the ECB chamber with $\pm\phi + m_i > 0$.

such that we find agreement between the geometric and gauge theoretic prepotentials with the assignments[18]

$$
\begin{aligned}
\mu_1 &= -h_0 - \frac{3}{2}m_1 + \frac{1}{2}m_2, \\
\mu_2 &= -2h_0 - m_1 + m_2, \\
\mu_6 &= h_0 + \frac{1}{2}(m_1 + m_2), \\
\nu &= -\phi - h_0 - \frac{1}{2}(m_1 - m_2).
\end{aligned}
\tag{A.18}
$$

**Flopped phase.** Equipped with these identifications, we can now consider the phase of interest to us where $\pm\phi + m_i > 0$, which we draw in figure 19. This can be obtained from the resolution in figure 18 by flopping the two curves $C_{6S}$ and $C_{5S}$ into $C_{45}$ and $C_{14}$.

In this phase we find the following curve volumes:

$$
\begin{aligned}
\mathrm{Vol}(C_{1S}) &= \mathrm{Vol}(C_{3S}) = \mu_2 - 2\nu = 2\phi, \\
\mathrm{Vol}(C_{2S}) &= \mathrm{Vol}(C_{4S}) = \mu_1 - 2\nu = 2\phi + h_0 - \frac{m_1 + m_2}{2} = 2\phi + m_\lambda, \\
\mathrm{Vol}(C_{14}) &= \nu - \mu_1 = m_1 - \phi, \\
\mathrm{Vol}(C_{45}) &= \mu_1 + \mu_6 = m_2 - m_1,
\end{aligned}
\tag{A.19}
$$

where we used (A.18) and we introduced $m_\lambda = h_0 - \frac{1}{2}(m_1 + m_2)$.

The gdP$_3$ geometry we just reviewed can be realized by the following $\mathcal{N} = 1$ GLSM:

|          | $D_1$ | $D_2$ | $D_3$ | $D_4$ | $D_5$ | $D_6$ | $S$  | FI    |
|----------|-------|-------|-------|-------|-------|-------|------|-------|
| $C_{2S}$ | 1     | 0     | 1     | 0     | 0     | 0     | $-2$ | $t_1$ |
| $C_{1S}$ | 0     | 1     | 0     | 1     | 0     | 0     | $-2$ | $t_2$ |
| $C_{14}$ | $-1$  | 0     | 0     | $-1$  | 1     | 0     | 1    | $t_3$ |
| $C_{45}$ | 1     | 0     | 0     | 0     | $-2$  | 1     | 0    | $t_4$ |

(A.20)

---

[18]This mapping can also be found by comparing the field theory IMS prepotential with the geometric prepotential in several different resolutions, i.e. gauge theory phases. This justifies the Ansatz (A.15).

The vacuum equations are

$$\mathcal{X}_{E_3}(t_1, t_2, t_3, t_4) = \left\{ \begin{array}{c} |z_1|^2 + |z_3|^2 - 2|z_0|^2 = t_1, \quad |z_2|^2 + |z_4|^2 - 2|z_0|^2 = t_2 \\ -|z_1|^2 - |z_4|^2 + |z_5|^2 + |z_0|^2 = t_3, \quad |z_1|^2 - 2|z_5|^2 + |z_6|^2 = t_4 \end{array} \right\} \bigg/ U(1)^4, \tag{A.21}$$

where $z_i$ are the homogeneous coordinates associated to $D_i$ and $z_0$ is the one associated to $S$, and the four $U(1)$'s act on them as in the table above. The FI parameters $t_1$, $t_2$, $t_3$, $t_4$ encode the volumes of the curves $C_{2S}$, $C_{1S}$, $C_{14}$, $C_{45}$ and so following (A.19) they are related to the Kähler and mass parameters as

$$\begin{aligned}
t_1 &= \mu_1 - 2\nu = 2\phi + m_\lambda, \\
t_2 &= \mu_2 - 2\nu = 2\phi, \\
t_3 &= \nu - \mu_1 = m_1 - \phi, \\
t_4 &= \mu_1 + \mu_6 = m_2 - m_1.
\end{aligned} \tag{A.22}$$

## A.3  $G_2$ geometry for the $E_3$ domain wall

Combining the previous field theory and geometric considerations, we can finally construct the $G_2$ geometry that realizes the domain wall in the case of the $E_3$ theory.

Observe from figure 19 and the curve volumes (A.19) that the core structure of the resolution of $\mathrm{gdP}_3$ that we are interested in is similar to that of $\mathbb{F}_0$ for the $E_1$ theory that we used in section 4. Specifically, there are two $\mathbb{P}^1$s $C_{1S}$ and $C_{2S}$ connected to the compact divisor $S$ whose volumes are $2\phi$ and $2\phi + m_\lambda$, such that the transition of interest to us where $\phi = 0$ and $m_\lambda$ changes its sign corresponds to a flop. Compared to the $E_1$ case this structure is further decorated with additional divisors and curves that realize the two extra flavors. In particular, setting $m_1 = m_2 = m > 0$, which is what we are interested in, blows down $C_{45}$ generating an $A_1$ singularity, which corresponds to an $\mathfrak{su}(2)$ flavor symmetry that will show up in the 4d domain wall theory as we will see in what follows.

After the specializations $m_1 = m_2 = m > 0$, the volumes of the curves in (A.19) become

$$\begin{aligned}
\mathrm{Vol}(C_{1S}) &= \mathrm{Vol}(C_{3S}) = 2\phi, \\
\mathrm{Vol}(C_{2S}) &= \mathrm{Vol}(C_{4S}) = 2\phi + m_\lambda, \\
\mathrm{Vol}(C_{45}) &= 0, \\
\mathrm{Vol}(C_{14}) &= m - \phi,
\end{aligned} \tag{A.23}$$

and considering the profile of $\phi$ discussed before that is

$$\phi = \frac{|m_\lambda| - m_\lambda}{4}, \tag{A.24}$$

we find the rulings on both sides of the domain wall to be represented as shown in figure 20. The $\mathfrak{su}(2)$ flavor symmetry we mentioned is manifest in the non-compact divisor $D_5$.

In order to describe this ruling of the $\mathrm{gdP}_3$ geometry it is convenient to perform a redefinition of the $U(1)$'s appearing in the GLSM (A.20)

| | $D_1$ | $D_2$ | $D_3$ | $D_4$ | $D_5$ | $D_6$ | $S$ | FI |
|---|---|---|---|---|---|---|---|---|
| $C_{2S} - C_{1S}$ | 1 | $-1$ | 1 | $-1$ | 0 | 0 | 0 | $t = t_1 - t_2 = m_\lambda$ |
| $C_{1S}$ | 0 | 1 | 0 | 1 | 0 | 0 | $-2$ | $t_2 = 2\phi$ |
| $C_{14}$ | $-1$ | 0 | 0 | $-1$ | 1 | 0 | 1 | $t_3 = m - \phi$ |
| $C_{45}$ | 1 | 0 | 0 | 0 | $-2$ | 1 | 0 | $t_4 = 0$ |

$$\tag{A.25}$$

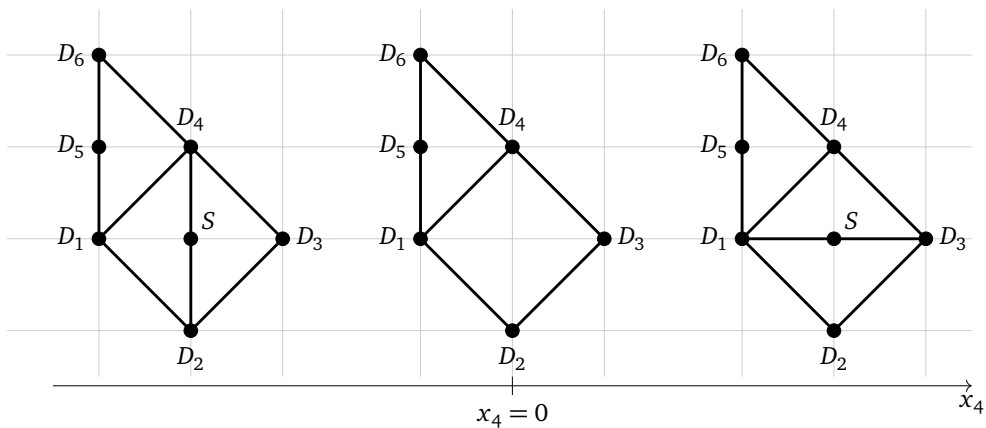

Figure 20: Domain wall geometry for the $E_3$ theory.

Recalling also from (A.5) that $m_c = m_\lambda + 4m$ is the parameter that is trivially identified between the two sides of the domain wall, the vacuum equations become

$$
\begin{aligned}
|z_1|^2 - |z_2|^2 + |z_3|^2 - |z_4|^2 &= m_\lambda, \\
|z_2|^2 + |z_4|^2 = 2|z_0|^2 + 2\phi &= 2|z_0|^2 + \frac{|m_\lambda| - m_\lambda}{2}, \\
|z_1|^2 + |z_6|^2 &= 2|z_5|^2, \\
-|z_1|^2 - |z_4|^2 + |z_5|^2 + |z_0|^2 = m - \phi &= \frac{m_c - |m_\lambda|}{4}.
\end{aligned}
\tag{A.26}
$$

Notice that we can solve the second and the third equation in terms of $|z_0|$ and $|z_5|$ and also use the second and the third $U(1)$'s in the GLSM (A.25) to fix their arguments. This means that we can use such $U(1)$'s to gauge fix the coordinates $z_0$ and $z_5$ up to two $\mathbb{Z}_2$ subgroups of these $U(1)$s due to the fact that $z_0$ and $z_5$ have charge 2. By using the $U(1)$ corresponding to the first line in (A.25), the action of the $\mathbb{Z}_2$ associated with gauge fixing $z_0$ can be written in two different ways. Depending on $x_4 < 0$ or $x_4 > 0$, writing it as

$$
\begin{aligned}
x_4 < 0 : \quad & (z_2, z_4) \rightarrow (-z_2, -z_4), \\
x_4 > 0 : \quad & (z_1, z_3) \rightarrow (-z_1, -z_3),
\end{aligned}
\tag{A.27}
$$

makes the fixed points and associated singularities manifest. The $\mathbb{Z}_2$ associated with $z_5$ acts as

$$
(z_1, z_6) \rightarrow (-z_1, -z_6).
\tag{A.28}
$$

The domain wall geometry can then be obtained by fibering $\mathcal{X}_{E_3}(2\phi + m_\lambda, 2\phi, m - \phi, 0)$ over $x_4 \in \mathbb{R}$ with

$$
m_\lambda = -x_4, \qquad m = \frac{m_c + x_4}{4}, \qquad \phi = \frac{x_4 + |x_4|}{4},
\tag{A.29}
$$

and $m_c$ being any positive real constant, as

$$
\mathcal{M}_{E_3}(m_c) = \left\{ \begin{aligned}
|z_1|^2 - |z_2|^2 + |z_3|^2 - |z_4|^2 &= -x_4 \quad | \quad x_4 \in \mathbb{R} \\
-|z_1|^2 + |z_2|^2 + |z_3|^2 - 3|z_4|^2 + 2|z_6|^2 &= m_c
\end{aligned} \right\} \Big/ U(1)^2 \times \mathbb{Z}_2^2, \tag{A.30}
$$

where the two $U(1)$'s are those in the first and last row of (A.25) and the two $\mathbb{Z}_2$'s are those in (A.27) and (A.28).

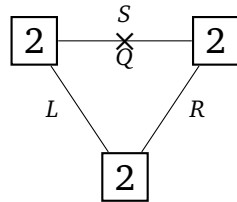

Figure 21: Quiver for the 4d $\mathcal{N} = 1$ theory domain wall theory associated to $E_3$.

As we already mentioned above, when we cross the domain wall at $x_4 = 0$ the geometry on the left side of figure 20, undergoes a flop involving the middle square which is similar to the one we saw in figure 1 for the $E_1$ domain wall.

For every $x_4$ we have $\mathcal{X}_{E_3}(\frac{|x_4| - x_4}{2}, \frac{|x_4| + x_4}{2}, \frac{m_c - |x_4|}{4}, 0)$, which has an $A_1$ singularity over a $\mathbb{P}^1$ whose size varies with $x_4$ and vanishes at $x_4 = 0$. Furthermore, there is a compact curve $C_{14}$ intersecting this $A_1$ and another $A_1$ singularity (which is a subgroup of the flavor symmetry of the $E_3$ theory) at $z_1 = z_6 = 0$. The curve on $\mathcal{X}_{E_3}$ defined by the latter is non-compact, exists for all $x_4$ and never meets the first $A_1$ singularity.

Inside the entire $\mathcal{M}_{E_3}(m_c)$ geometry, the compact $A_1$ of $\mathcal{X}_{E_3}$ gives rise to two non-compact $A_1$ singularities sitting at $z_2 = z_4 = 0$ for $x_4 < 0$ and at $z_1 = z_3 = 0$ for $x_4 > 0$, as shown in figure 9. These correspond to the two $\mathfrak{su}(2)$ flavor symmetries that appear as the boxes on the left and on the right of the 4d $\mathcal{N} = 1$ domain wall field theory, which we discussed in subsection 5.3 and we reproduce for the case $N_F = 2$ in figure 21. As the two $A_1$ singularities meet at $x_4 = 0$, there are massless $\mathfrak{su}(2) \oplus \mathfrak{su}(2)$ bifundamental chirals drawn as an horizontal edges in figure 21. Furthermore, there are two additional chiral fields coming from the two singular points appearing when $C_{14}$ shrinks to zero size in figure 9, which are $\mathfrak{su}(2) \oplus \mathfrak{su}(2)$ bifundamentals drawn as diagonal edges in figure 21, where one $\mathfrak{su}(2)$ is the one on the bottom, and the other is either the left or the right one.

### A.4 Stacking domain walls

Similarly to what we discussed in subsection 4.5 for the $E_1$ domain wall, we can stack two of the $E_3$ domain walls that we just discussed, which annihilate each other leaving just the 5d $SU(2) + 2\textbf{F}$ theory on a line. In field theory this is once again understood as a Seiberg duality

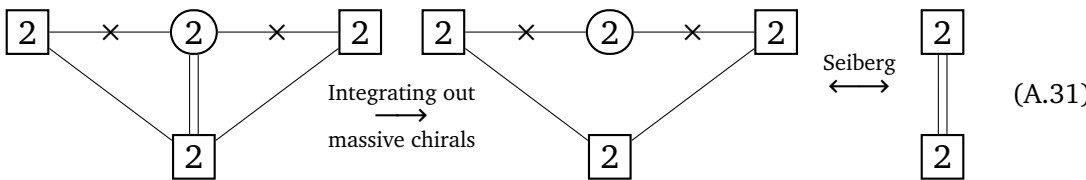

(A.31)

On the left and middle quiver, the $\mathfrak{su}(2)$'s at the two sides are flavor symmetries from the perspective of the 4d domain wall theory while they are gauged in the 5d bulk, the $\mathfrak{su}(2)$ in the middle is a 4d gauge symmetry and the $\mathfrak{su}(2)$ on the bottom is a flavor symmetry both from the 4d and the 5d perspective. For the left quiver there are two cubic superpotential terms, one for each triangle of chirals and a quadratic mass term for the two chirals in the middle. For the middle quiver the massive chirals are integrated out, and we are left with a quartic superpotential that involves the four chirals forming a loop. On the right quiver, we have two $\mathfrak{su}(2) \oplus \mathfrak{su}(2)$ chirals that recombine to form an hyper.

We can understand this geometrically exactly in the same way as we did for the $E_1$ theory, since from the defining equations of $\mathcal{M}_{E_3}(m_c)$ we see that the only equation with a Kähler parameter that has a non-trivial profile along $x_4$ is the first one which is the same as the one that appeared in $\mathcal{M}_{E_1}$ for the domain wall of the $E_1$ theory and in $\mathcal{M}_C$ for the free hyper, see equations. (124)-(5) respectively.

We define the geometry of two stacked domain walls as

$$\mathcal{M}_{2E_3}(m_c) = \left\{ \begin{array}{c} -|z_1|^2 + |z_2|^2 - |z_3|^2 + |z_4|^2 = x_4^2 - a \quad | \quad x_4 \in \mathbb{R} \\ -|z_1|^2 + |z_2|^2 + |z_3|^2 - 3|z_4|^2 + 2|z_6|^2 = m_c \end{array} \right\} \Big/ U(1)^2 \times \mathbb{Z}_2^2. \quad \text{(A.32)}$$

For $a > 0$ there are three $A_1$ singularities, two of which, those for $|x_4| > \sqrt{a}$, sit on a non-compact space whose cross section is a $\mathbb{P}^1$ and hence give rise to the $\mathfrak{su}(2)$'s on the two sides of the quivers on the left and middle of (A.31), while the one for $|x_4| < \sqrt{a}$ sits on a compact $S^3$ and gives rise to the gauge $\mathfrak{su}(2)$ in (A.31). The picture is then similar to the one of figure 5 for $E_1$, but this is further decorated by the curve $C_{14}$ and the non-compact $A_1$ singularity as in figure 9. For $a < 0$ we have a transition to the trivial geometry $\mathcal{X}_{E_3} \times \mathbb{R}$.

In a similar manner to the $E_1$ theory case we can generalize the above to the stacking of $n$ domain walls by fibering $\mathcal{X}_{E_3}(\frac{|m_\lambda|+m_\lambda}{2}, \frac{|m_\lambda|-m_\lambda}{2}, \frac{m_c-|m_\lambda|}{4}, 0)$ over $x_4 \in \mathbb{R}$ with

$$m_\lambda = P_n(x_4), \quad \text{(A.33)}$$

for some real polynomial $P_n(x_4)$ with $n$ roots that in our convention we take to be positive for $x_4 \to -\infty$. For $n$ even we have a transition to $\mathcal{X}_{E_3} \times \mathbb{R}$, i.e. there is no domain wall left, while for $n$ odd the transition is to $\mathcal{M}_{E_3}(m_c)$, i.e. there is one single domain wall. This again matches the field theory expectation, where the same result is achieved by applying $n-1$ times Seiberg duality.

# B  UV duality domain wall between $SU(2)^3$ and $SU(4)_0 + 4\text{F}$

In this appendix we give more details about the UV duality domain wall interpolating between the linear quiver with $k - 1$ $SU(2)$ gauge nodes where adjacent nodes are connect by one bifundamental hyper-multiplet and the $SU(k)_0 + (2k-4)\text{F}$ for the case of $k = 4$. In particular, we compute the geometric prepotential in a particular resolution and match it with the field theory prepotential in the corresponding phase of both of the gauge theories so to find the precise mapping between the geometric and the field theory parameters, which will allow us to then understand which partial resolutions of the toric polygon of the UV SCFT are involved in the construction of the domain wall. We also propose an alternative domain wall in this case compared to the one we discussed in subsection 7.2.

## B.1  Geometric prepotential

We consider the geometry of the rank 3 SCFT that UV completes both the $SU(2)^3$ and the $SU(4)_0 + 4\text{F}$ theories in the specific resolution depicted in figure 22.

The three relations among the divisors

$$\sum_{i=1}^{9} D_i + \sum_{a=1}^{3} S_a \cong 0,$$

$$D_2 + D_3 + D_4 \cong D_6 + D_7 + D_8, \quad \text{(B.1)}$$

$$D_2 + 2D_3 + 3D_4 + 4D_5 + 3D_6 + 2D_7 + D_8 \cong 0,$$

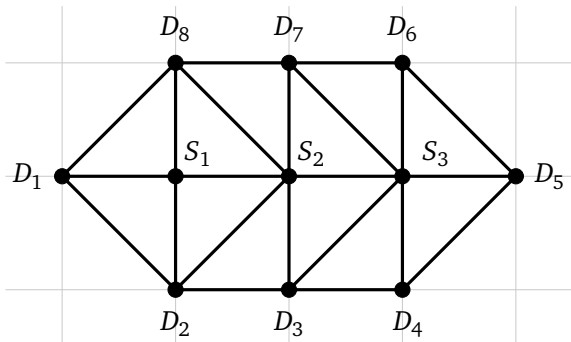

Figure 22: A resolution of the toric polygon of the millipede SCFT that UV completes both the $SU(2)^3$ linear quiver theory and the $SU(4)_0 + 4F$ theory.

can be used to treat as independent, for example, $D_1, D_2, D_3, D_4, D_6, S_1, S_2, S_3$, so we can parameterize the Kähler form form as

$$J = \mu_1 D_2 + \mu_2 D_2 + \mu_3 D_3 + \mu_4 D_4 + \mu_6 D_6 + \nu_1 S_1 + \nu_2 S_2 + \nu_3 S_3 \,. \tag{B.2}$$

For the computation of the geometric prepotential $\mathcal{F}_{\text{geom}} = -\frac{1}{6} J^3$ we need the following relevant non-zero triple intersection numbers

$$
\begin{aligned}
& S_1^2 D_1 = -2\,, && S_1^2 D_2 = -2\,, && S_2^2 D_2 = -1\,, && S_2^2 D_3 = -1\,, && S_3^2 D_3 = -1\,, \\
& S_3^2 D_4 = -1\,, && S_2 D_2^2 = -1\,, && S_2 D_3^2 = -1\,, && S_3 D_3^2 = -1\,, && S_3 D_4^2 = -1\,, \\
& S_1^2 S_2 = -2\,, && S_2^2 S_3 = -2\,, && S_1 D_1 D_2 = 1\,, && S_1 S_2 D_2 = 1\,, && S_2 D_2 D_3 = 1\,, \\
& S_2 S_3 D_3 = 1\,, && S_3 D_3 D_4 = 1\,, && S_1^3 = 8\,, && S_2^3 = 6\,, && S_3^3 = 6\,.
\end{aligned}
\tag{B.3}
$$

We then get

$$
\mathcal{F}_{\text{geom}} = (\mu_1 + \mu_2)\, \nu_1^2 + \frac{\mu_2 + \mu_3}{2}\, \nu_2^2 + \frac{\mu_3 + \mu_4 + \mu_6}{2}\, \nu_3^2 - \frac{4}{3} \nu_1^3 - \nu_2^3 - \nu_3^3 + \nu_1^2 \nu_2 + \nu_2^2 \nu_3 \tag{B.4}
$$
$$
- \mu_2 \nu_1 \nu_2 - \mu_3 \nu_2 \nu_3 - \mu_1 \mu_2 \nu_1 + \left( \frac{\mu_2^2}{2} - \mu_2 \mu_3 + \frac{\mu_3^2}{2} \right) \nu_2 + \left( \frac{\mu_3^2}{2} - \mu_3 \mu_4 + \frac{\mu_4^2}{2} + \frac{\mu_6^2}{2} \right) \nu_3 \,.
$$

We can also compute the curve volumes, which will be useful when we will discuss the domain wall

$$
\begin{aligned}
& \text{Vol}\big(C_{1S_1}\big) = \text{Vol}\big(C_{S_1 S_2}\big) = \mu_2 - 2\nu_1\,, && \text{Vol}\big(C_{S_2 S_3}\big) = \mu_3 - 2\nu_2\,, \\
& \text{Vol}\big(C_{5S_3}\big) = \mu_4 - 2\nu_3 + \mu_6\,, && \text{Vol}\big(C_{2S_1}\big) = \text{Vol}\big(C_{8S_1}\big) = \mu_1 - 2\nu_1 + \nu_2\,, \\
& \text{Vol}\big(C_{3S_2}\big) = \mu_2 - \mu_3 - \nu_2 + \nu_3\,, && \text{Vol}\big(C_{4S_3}\big) = \mu_3 - \mu_4 - \nu_3\,, \\
& \text{Vol}\big(C_{7S_2}\big) = -\nu_2 + \nu_3\,, && \text{Vol}\big(C_{6S_3}\big) = -\mu_6 - \nu_3\,, \\
& \text{Vol}\big(C_{2S_2}\big) = -\mu_2 + \mu_3 + \nu_1 - \nu_2\,, && \text{Vol}\big(C_{8S_2}\big) = \nu_1 - \nu_2\,, \\
& \text{Vol}\big(C_{3S_3}\big) = -\mu_3 + \mu_4 + \nu_2 - \nu_3\,, && \text{Vol}\big(C_{7S_3}\big) = \mu_6 + \nu_2 - \nu_3\,.
\end{aligned}
\tag{B.5}
$$

## B.2 Linear $SU(2)^3$ quiver theory

We now consider the linear quiver theory with three $SU(2)$ gauge nodes where adjacent nodes are connected by a bifundamental hyper. The theory has three $\mathfrak{u}(1)$ instantonic symmetries, whose mass parameters we denote by $h_1, h_2, h_3$, and two $\mathfrak{su}(2)$ symmetries, whose mass parameters we denote by $m_1, m_2$, that act independently on each bifundamental hyper-multiplet

by rotating the two half-hyper-multiplets it is made of. The IMS prepotential of the theory reads

$$\mathcal{F}_{\text{IMS}}^{SU(2)^3} = h_1\phi_1^2 + h_2\phi_2^2 + h_3\phi_3^2 + \frac{4}{3}\left(\phi_1^3 + \phi_2^3 + \phi_3^3\right) \tag{B.6}$$
$$-\frac{1}{12}\left(|\phi_1 + \phi_2 + m_1|^3 + |\phi_1 - \phi_2 + m_1|^3 + |-\phi_1 + \phi_2 + m_1|^3 + |-\phi_1 - \phi_2 + m_1|^3\right)$$
$$-\frac{1}{12}\left(|\phi_2 + \phi_3 + m_2|^3 + |\phi_2 - \phi_3 + m_2|^3 + |-\phi_2 + \phi_3 + m_2|^3 + |-\phi_2 - \phi_3 + m_2|^3\right).$$

The resolution we considered in the previous subsection corresponds to the phase where

$$\begin{aligned} \phi_1 + \phi_2 \pm m_1 > 0, \qquad & -\phi_1 + \phi_2 \pm 1 m_1 > 0, \\ \phi_2 + \phi_3 \pm m_2 > 0, \qquad & -\phi_2 + \phi_3 \pm m_2 > 0, \end{aligned} \tag{B.7}$$

in which the prepotential becomes

$$\mathcal{F}_{\text{IMS}}^{SU(2)^3} = h_1\phi_1^2 + h_2\phi_2^2 + h_3\phi_3^2 + \frac{4}{3}\phi_1^3 + \phi_2^3 + \phi_3^3 - \phi_1^2\phi_2 - \phi_2^2\phi_3 - m_1^2\phi_2 - m_2^2\phi_3. \tag{B.8}$$

In order to facilitate the match of the field theory prepotential (B.8) with the geometric one (B.4), it is useful to make the following Ansatz (based on the results of the $k = 3$ case we saw in subsection 7.1):

$$\begin{aligned} \text{Vol}\left(C_{1S_1}\right) &= \mu_2 - 2\nu_1 = 2\phi_1, \\ \text{Vol}\left(C_{S_2 S_3}\right) &= \mu_3 - 2\nu_2 = 2\phi_2, \\ \text{Vol}\left(C_{5 S_3}\right) &= \mu_4 - 2\nu_3 + \mu_6 = 2\phi_3. \end{aligned} \tag{B.9}$$

This can be solved as

$$\nu_1 = -\phi_1 + \frac{\mu_2}{2}, \qquad \nu_2 = -\phi_2 + \frac{\mu_3}{2}, \qquad \nu_3 = -\phi_3 + \frac{\mu_4 + \mu_6}{2}, \tag{B.10}$$

which we can plug in the geometric prepotential (B.4) so to get

$$\mathcal{F}_{\text{geom}} = \left(\mu_1 - \mu_2 + \frac{\mu_3}{2}\right)\phi_1^2 + \left(\frac{\mu_2}{2} - \mu_3 + \frac{\mu_4}{2} + \frac{\mu_6}{2}\right)\phi_2^2 + \left(\frac{\mu_3}{2} - \mu_4 - \mu_6\right)\phi_3^2$$
$$+ \frac{4}{3}\phi_1^3 + \phi_2^+ \phi_3^3 - \phi_1^2\phi_2 - \phi_2^2\phi_3 - \left(\frac{\mu_3 - \mu_2}{2}\right)^2 \phi_2 - \left(\frac{\mu_3 - \mu_4 + \mu_6}{2}\right)^2 \phi_3. \tag{B.11}$$

The comparison with the field theory prepotential (B.8) is now immediate and leads us to identify

$$\begin{aligned} h_1 &= \mu_1 - \mu_2 + \frac{\mu_3}{2}, \qquad & h_2 &= \frac{\mu_2}{2} - \mu_3 + \frac{\mu_4}{2} + \frac{\mu_6}{2}, \qquad & h_3 &= \frac{\mu_3}{2} - \mu_4 - \mu_6, \\ m_1 &= \frac{\mu_3 - \mu_2}{2}, \qquad & m_2 &= \frac{\mu_3 - \mu_4 + \mu_6}{2}, \end{aligned} \tag{B.12}$$

which combined with (B.10) gives us

$$\begin{aligned} \mu_1 &= h_1 - 2h_2 - h_3 - 4m_1, \\ \mu_2 &= -4h_2 - 2h_3 - 6m_1, \\ \mu_3 &= -4h_2 - 2h_3 - 4m_1, \\ \mu_4 &= -3h_2 - 2h_3 - 3m_1 - m_2, \\ \mu_6 &= h_2 + m_1 + m_2, \\ \nu_1 &= -\phi_1 - 2h_2 - h_3 - 3m_1, \\ \nu_2 &= -\phi_2 - 2h_2 - h_3 - 2m_1, \\ \nu_3 &= -\phi_3 - h_2 - h_3 - m_1. \end{aligned} \tag{B.13}$$

## B.3 $SU(4)_0 + 4F$ theory

We now move to the $SU(4)_0 + 4F$ theory, which has a $\mathfrak{u}(1)$ instantonic symmetry whose mass parameter we denote by $\widetilde{h}_0$ and a $\mathfrak{u}(4)$ flavor symmetry whose mass parameters we denote by $\widetilde{m}_i$ for $i = 1, \cdots, 4$. The IMS prepotential is

$$\mathcal{F}_{\text{IMS}}^{SU(4)_0+4F} = \widetilde{h}_0\left(\widetilde{\phi}_1^2 + \widetilde{\phi}_2^2 + \widetilde{\phi}_3^2 - \widetilde{\phi}_1\widetilde{\phi}_2 - \widetilde{\phi}_2\widetilde{\phi}_3\right) + \frac{4}{3}\left(\widetilde{\phi}_1^3 + \widetilde{\phi}_2^3 + \widetilde{\phi}_3^3\right) - \left(\widetilde{\phi}_1 + \widetilde{\phi}_3\right)\widetilde{\phi}_2^2 \quad \text{(B.14)}$$

$$- \frac{1}{12}\sum_{i=1}^{4}\left(|\widetilde{\phi}_1 + \widetilde{m}_i|^3 + |-\widetilde{\phi}_1 + \widetilde{\phi}_2 + \widetilde{m}_i|^3 + |-\widetilde{\phi}_2 + \widetilde{\phi}_3 + \widetilde{m}_i|^3 + |-\widetilde{\phi}_3 + \widetilde{m}_i|\right).$$

We are interested in the phase with

$$
\begin{aligned}
i = 1, 2: &\quad \widetilde{\phi}_1 + \widetilde{m}_i > 0, \quad -\widetilde{\phi}_1 + \widetilde{\phi}_2 + \widetilde{m}_i > 0, \quad -\widetilde{\phi}_2 + \widetilde{\phi}_3 + \widetilde{m}_i < 0, \quad -\widetilde{\phi}_3 + \widetilde{m}_i < 0, \\
i = 3, 4: &\quad \widetilde{\phi}_1 + \widetilde{m}_i > 0, \quad -\widetilde{\phi}_1 + \widetilde{\phi}_2 + \widetilde{m}_i > 0, \quad -\widetilde{\phi}_2 + \widetilde{\phi}_3 + \widetilde{m}_i > 0, \quad -\widetilde{\phi}_3 + \widetilde{m}_i < 0,
\end{aligned}
\tag{B.15}
$$

where the prepotential becomes

$$
\begin{aligned}
\mathcal{F}_{\text{IMS}}^{SU(4)_0+4F} =& \left(\widetilde{h}_0 - \frac{\widetilde{m}_1 + \widetilde{m}_2 + \widetilde{m}_3 + \widetilde{m}_4}{2}\right)\widetilde{\phi}_1^2 + \left(\widetilde{h}_0 - \frac{\widetilde{m}_3 + \widetilde{m}_4}{2}\right)\widetilde{\phi}_2^2 \\
&+ \left(\widetilde{h}_0 + \frac{\widetilde{m}_1 + \widetilde{m}_2}{2}\right)\widetilde{\phi}_3^2 + \frac{4}{3}\widetilde{\phi}_1^3 + \widetilde{\phi}_2^3 + \widetilde{\phi}_3^3 - \widetilde{\phi}_1^2\widetilde{\phi}_2 - \widetilde{\phi}_2^2\widetilde{\phi}_3 \\
&- \left(\widetilde{h}_0 - \frac{\widetilde{m}_1 + \widetilde{m}_2 + \widetilde{m}_3 + \widetilde{m}_4}{2}\right)\widetilde{\phi}_1\widetilde{\phi}_2 - \left(\widetilde{h}_0 + \frac{\widetilde{m}_1 + \widetilde{m}_2 - \widetilde{m}_3 - \widetilde{m}_4}{2}\right)\widetilde{\phi}_2\widetilde{\phi}_3 \\
&- \frac{\widetilde{m}_1^2 + \widetilde{m}_2^2}{2}\widetilde{\phi}_2 - \frac{\widetilde{m}_3^2 + \widetilde{m}_4^2}{2}\widetilde{\phi}_3.
\end{aligned}
\tag{B.16}
$$

Similarly to what we did in the previous subsection, in order to facilitate the comparison of the field theory prepotential (B.16) of the $SU(4)_0 + 4F$ theory with the geometric prepotential (B.4) we make the following Ansatz, based on similarity with the $k = 3$ case discussed in subsection 7.1:

$$
\begin{aligned}
\text{Vol}\left(C_{2S_1}\right) &= \mu_1 - 2\nu_1 + \nu_2 = 2\widetilde{\phi}_1 - \widetilde{\phi}_2, \\
\text{Vol}\left(C_{3S_2}\right) &= \mu_2 - \mu_3 - \nu_2 + \nu_3 = \widetilde{\phi}_2 - \widetilde{\phi}_3 - \widetilde{m}_2, \\
\text{Vol}\left(C_{4S_3}\right) &= \mu_3 - \mu_4 - \nu_3 = \widetilde{\phi}_3 - \widetilde{m}_4,
\end{aligned}
\tag{B.17}
$$

which can be solved as

$$
\begin{aligned}
\nu_1 &= -\widetilde{\phi}_1 + \frac{\widetilde{m}_2 + \widetilde{m}_4}{2} + \frac{\mu_1 + \mu_2 - \mu_4}{2}, \\
\nu_2 &= -\widetilde{\phi}_2 + \widetilde{m}_2 + \widetilde{m}_4 + \mu_2 - \mu_4, \\
\nu_3 &= -\widetilde{\phi}_3 + \widetilde{m}_4 + \mu_3 - \mu_4.
\end{aligned}
\tag{B.18}
$$

Plugging this back into the geometric prepotential (B.4) allows us to match it more easily with the field theory one (B.16), so that we find the mapping of parameters

$$
\begin{aligned}
\mu_1 &= 4\widetilde{m}_1\,, \\
\mu_2 &= \widetilde{h}_0 + \frac{11\widetilde{m}_1 - \widetilde{m}_2 - \widetilde{m}_3 - \widetilde{m}_4}{2}\,, \\
\mu_3 &= \widetilde{h}_0 + \frac{9\widetilde{m}_1 + \widetilde{m}_2 - \widetilde{m}_3 - \widetilde{m}_4}{2}\,, \\
\mu_4 &= \widetilde{h}_0 + \frac{7\widetilde{m}_1 + \widetilde{m}_2 - \widetilde{m}_3 + \widetilde{m}_4}{2}\,, \\
\mu_6 &= -\widetilde{m}_1 + \widetilde{m}_3\,, \\
\nu_1 &= -\widetilde{\phi}_1 + 3\widetilde{m}_1\,, \\
\nu_2 &= -\widetilde{\phi}_2 + 2\widetilde{m}_1\,, \\
\nu_3 &= -\widetilde{\phi}_3 + \widetilde{m}_1\,.
\end{aligned}
\tag{B.19}
$$

**S-duality map.** By comparing (B.13) and (B.19) we can find the mapping of parameters across the S-duality that relates the $SU(2)^3$ linear quiver and the $SU(4)_0 + 4\boldsymbol{F}$ theory

$$
\begin{aligned}
\widetilde{\phi}_1 &= \phi_1 + \frac{3h_1 + 2h_2 + h_3}{4}\,, & \widetilde{\phi}_2 &= \phi_2 + \frac{h_1 + 2h_2 + h_3}{2}\,, & \widetilde{\phi}_3 &= \phi_3 + \frac{h_1 + 2h_2 + 3h_3}{4}\,, \\
\widetilde{h}_0 &= -h_1 - h_2 - h_3\,, & \widetilde{m}_1 &= \frac{h_1 - h_3}{4} - \frac{h_2}{2} - m_1\,, & \widetilde{m}_2 &= \frac{h_1 - h_3}{4} - \frac{h_2}{2} + m_1\,, \\
\widetilde{m}_3 &= \frac{h_1 - h_3}{4} + \frac{h_2}{2} + m_2\,, & \widetilde{m}_4 &= \frac{h_1 - h_3}{4} + \frac{h_2}{2} - m_2\,.
\end{aligned}
\tag{B.20}
$$

Notice that in particular the last two lines instruct us about the precise mapping of the global symmetries between the two dual theories. Specifically, the instanton mass of the $SU(4)_0 + 4\boldsymbol{F}$ theory corresponds to the diagonal combination of all the instantonic symmetries of the linear quiver theory. The $\mathfrak{u}(1)_B$ baryonic symmetry, defined by expressing the flavor symmetry as $\mathfrak{u}(4) \cong \mathfrak{su}(4) \oplus \mathfrak{u}(1)_B$, corresponds to the off-diagonal combination $\frac{1}{4}(h_1 - h_3)$ of the instantonic symmetries of the first and the last gauge nodes in the quiver. Finally, the $\mathfrak{su}(4)$ flavor symmetry arises as an enhancement of the two $\mathfrak{su}(2)$ symmetries that rotate the two bifundamental hypers in the quiver and the instantonic symmetry of the middle gauge node, which is balanced. This confirms the parameter map across the duality domain wall we proposed in subsection 7.2 for generic $k$.

## B.4 An alternative domain wall between $SU(2)^3$ and $SU(4)_0 + 4\mathbf{F}$

We will now consider a different domain wall set-up compared to the one we discussed in subsection 7.2. This is obtained by setting the parameters of the theories on the two sides of the wall to a different value rather than as in (149), namely

$$
\begin{aligned}
SU(2)^3: \quad &\phi_1 = \phi_2 = \phi_3 = m_1 = m_2 = 0\,, & h_1 = h_2 = h_3 = m_\lambda\,, \\
SU(4)_0 + 4\boldsymbol{F}: \quad &\widetilde{\phi}_1 = \widetilde{\phi}_2 = \widetilde{\phi}_3 = \widetilde{\phi}_4 = 0\,, & \widetilde{m}_1 = \widetilde{m}_2 = -\widetilde{m}_3 = -\widetilde{m}_4 = -\frac{m_\lambda}{2}\,, \\
&\widetilde{h}_0 = -3m_\lambda\,.
\end{aligned}
\tag{B.21}
$$

Notice that this is an allowed specialization since it is compatible with the mapping of the parameters for the global symmetries across the UV duality that we found in (B.20).

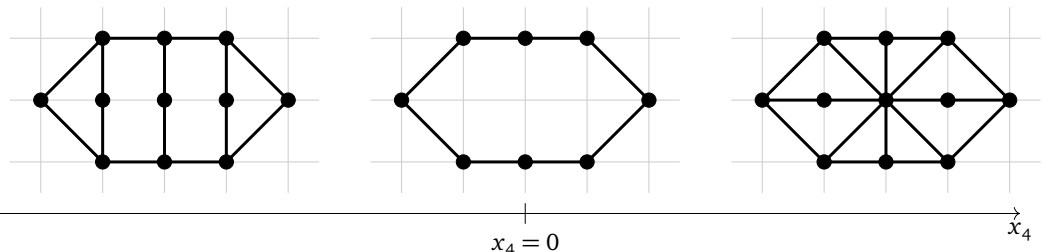

Figure 23: Geometry of the alternative domain wall between the $SU(2)^3$ quiver and the $SU(4)_0 + 4\boldsymbol{F}$ theory.

This implies that the effective gauge couplings are set to

$$
\begin{aligned}
SU(2)^3: \quad & \frac{1}{2\left(g_{\text{eff}}^{(1)}\right)^2} = \cdots = \frac{1}{2\left(g_{\text{eff}}^{(k-1)}\right)^2} = m_\lambda, \\
SU(4)_0 + 4\boldsymbol{F}: \quad & \frac{1}{2\widetilde{g}_{\text{eff}}^2} = -3m_\lambda.
\end{aligned}
\tag{B.22}
$$

Hence, in contrast with the other set-up we considered in subsection 7.2, the theories on both sides of the domain wall are completely Lagrangian: on the left we have the $SU(2)^3$ quiver with massless hypers while on the right we have the $SU(4)_0 + 4\boldsymbol{F}$ theory with a specific mass deformation turned on, which gives the same mass in absolute value to the four hypers but for two of them it is positive while for the other two it is negative.

The domain wall is once again constructed by taking $m_\lambda$ to vary in the $x_4$ direction, going from positive on the left $x_4 < 0$ to negative on the right $x_4 > 0$, so that at the location of the domain wall $m_\lambda = 0$ implying that all the effective gauge couplings of both of the theories on the two sides diverge. This can be achieved as usual by setting $m_\lambda = -x_4$.

In order to understand which partial resolution of the geometry in figure 22 we have on the two sides of the domain wall with this configuration, we just look at the curve volumes and the mapping of the field theory and the geometric parameters that we worked out in the previous subsections. The final result is depicted in figure 23.

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
