# Peer review of "$G_2$-Manifolds from 4d N=1 Theories, Part I: Domain Walls"

_SciPost Physics, doi:SciPost Phys. 17, 102 (2024)_

## Round 1 · Referee Report · Anonymous (Referee 1) · 2024-6-17

Report

This manuscript provides a novel physical procedure for constructing non-compact G2 manifolds by analyzing domain walls in 5D N=1 QFTs, which is systematically implemented in several families of cases. The outcome not only reproduced some non-compact 7d manifolds with explicitly known G2 metrics but also produced many more non-compact 7d manifolds on which G2 metrics are physically justified to exist.

I would recommend publication of this manuscript after the authors considered the following comments/suggestions.

Requested changes

One major comment:

The main objective of this paper is to construct a large family of non-compact G2 manifolds based on physical reasonings. Therefore, the part of ``geometrical checks" is a key component of the paper, since they reproduce specific mathematical results on G2 spaces and provide physics-based conjectures. To emphasize its key role, I recommend that the authors consider repackaging this part into an independent concise appendix, where results are stated purely in the mathematical language. For example, it would be nice to collect individual G2 geometries obtained in each section into that appendix.

The purpose would be to guide a mathematician with little knowledge of string theory to navigate through your paper with minimal effort. For that, it will be helpful to use terms such as "physic-based conjectures/hypothesis", make statements in mathematical terms, and put the minimal amount of necessary physical reasoning in quotation marks.

Minor comments:

  1. On P4, "Utilizing this, allows us" -> maybe "Utilizing this allows us"
  2. On P8, "Note that is equivalently ..." -> maybe "Note that this is equivalently ..."
  3. On P9, "in depth" -> "in-depth"
  4. On P10, "UV-duality-walls" -> "UV-duality walls"
  5. On P37, "Lowering the connection": here it helps to refer to Figure 8 once more.
  6. On P46: "The toric polygon for for ..." -> remove the extra "for"
  7. On P50: "where similarly to the case N=2 with flavors m_\lambda" -> add a comma after "flavors"

Recommendation

Publish (easily meets expectations and criteria for this Journal; among top 50%)

---

## Round 1 · Referee Report · Anonymous (Referee 2) · 2024-7-20

Strengths

1 - The paper connects known field theoretic results in 5d with nice geometric engineering perspectives

2 - The paper has explicit and detailed computations which should be very useful for further investigations

Weaknesses

No weaknesses

Report

In this paper the authors geometrize (find an underlying manifold with $G_2$ holonomy) a class of 4d duality domain wall theories in 5d. The two theories the domain wall interpolates between, are UV dual to each other. Such domain walls were studied field theoretically by Gaiotto and Kim in the past. For general 5d QFTs there is no algorithmic way to derive the theory living on such domain walls. The field theoretic procedure roughly amounts to guessing a theory living on the wall and performing numerous consistancy checks, e. g. studying anomalies. The understanding of such domain walls using geometric engineering in this paper provides a promising pathway into a more algorithmic study of the domain walls by embedding them in the larger string theoretic construction. Beyond the intrinsic interest in such constructions the domain wall theories studied in this paper play an important role in understanding compactifications of 6d SCFTs to 4d. As such progress on understanding them will likely lead to larger impact, e.g. better understanding of 4d supersymmetric physics.

The paper is very well written and easy to follow. It contains detailed computations and interesting results which should be of interest to a broader community interested in supersymmetric quantum field theories in dimensions 4-6. I recommend it to be published in its current form.

Recommendation

Publish (easily meets expectations and criteria for this Journal; among top 50%)

---

## Round 2 · Author Response

We would like to thank the referees for their comments and feedback. In the revised version we addressed the few minor comments addressed by the first referees.

---

## Round 2 · List of Changes

We modified the manuscript following the minor comments of the first referee. These include mainly typos, rephrasings and references.

---

## Editorial Decision

published